# Biofunctionalization of Multiplexed Silicon Photonic Biosensors

**DOI:** 10.3390/bios13010053

**Published:** 2022-12-29

**Authors:** Lauren S. Puumala, Samantha M. Grist, Jennifer M. Morales, Justin R. Bickford, Lukas Chrostowski, Sudip Shekhar, Karen C. Cheung

**Affiliations:** 1School of Biomedical Engineering, University of British Columbia, 2222 Health Sciences Mall, Vancouver, BC V6T 1Z3, Canada; 2Centre for Blood Research, University of British Columbia, 2350 Health Sciences Mall, Vancouver, BC V6T 1Z3, Canada; 3Dream Photonics Inc., Vancouver, BC V6T 0A7, Canada; 4Army Research Laboratory, US Army Combat Capabilities Development Command, 2800 Powder Mill Rd., Adelphi, MD 20783, USA; 5Department of Electrical and Computer Engineering, University of British Columbia, 2332 Main Mall, Vancouver, BC V6T 1Z4, Canada; 6Stewart Blusson Quantum Matter Institute, University of British Columbia, 2355 East Mall, Vancouver, BC V6T 1Z4, Canada

**Keywords:** silicon photonics, evanescent field biosensor, SOI biosensor, biofunctionalization, functionalization, bioreceptor, immobilization chemistry, biopatterning, microfluidics

## Abstract

Silicon photonic (SiP) sensors offer a promising platform for robust and low-cost decentralized diagnostics due to their high scalability, low limit of detection, and ability to integrate multiple sensors for multiplexed analyte detection. Their CMOS-compatible fabrication enables chip-scale miniaturization, high scalability, and low-cost mass production. Sensitive, specific detection with silicon photonic sensors is afforded through biofunctionalization of the sensor surface; consequently, this functionalization chemistry is inextricably linked to sensor performance. In this review, we first highlight the biofunctionalization needs for SiP biosensors, including sensitivity, specificity, cost, shelf-stability, and replicability and establish a set of performance criteria. We then benchmark biofunctionalization strategies for SiP biosensors against these criteria, organizing the review around three key aspects: bioreceptor selection, immobilization strategies, and patterning techniques. First, we evaluate bioreceptors, including antibodies, aptamers, nucleic acid probes, molecularly imprinted polymers, peptides, glycans, and lectins. We then compare adsorption, bioaffinity, and covalent chemistries for immobilizing bioreceptors on SiP surfaces. Finally, we compare biopatterning techniques for spatially controlling and multiplexing the biofunctionalization of SiP sensors, including microcontact printing, pin- and pipette-based spotting, microfluidic patterning in channels, inkjet printing, and microfluidic probes.

## 1. Introduction

Biosensors, which comprise a transducer and biorecognition element, aim to meet increasing demands for medical diagnostics by permitting rapid testing, guiding personalized care, and reducing healthcare costs in decentralized and low-resource settings [1,2,3]. Silicon photonic (SiP) sensors are one class of optical refractometric sensors with promise as sensitive, rapid, and inexpensive transducers for point-of-care (POC) biosensing [4]. Compared to other types of transducers employed for biosensing, such as electrochemical [5], piezoelectric [6], and mechanical (e.g., microcantilever) [7] sensors, some advantages of SiP sensors are their high sensitivity, wide dynamic range, compatibility with label-free operation, mechanical stability, and insensitivity to electromagnetic interferences [8]. SiP devices can be patterned with wafer-scale semiconductor fabrication techniques, allowing for reproducible, inexpensive, and highly scalable production [1,9,10]. These devices consist of nanoscale patterned silicon or silicon nitride structures that can guide and manipulate light, owing to the high refractive index contrast between the structures themselves and the surrounding media [1,11]. In SiP sensors, near-infrared light is confined in silicon or silicon nitride waveguides [1,12]. A portion of the light’s electric field, known as the evanescent field, extends outside the waveguide and interacts with the surrounding medium to create a refractive index-sensitive region (Figure 1a) [1]. A change in the refractive index within this region due to analyte capture on the waveguide surface, for example, perturbs the evanescent field and changes the effective refractive index, *n_eff_*, of the guided optical mode [1,4]. This translates to a shift in the optical phase, and in the case of resonant circuit architectures, leads to a resonance wavelength shift that is proportional to the amount of bound analyte, yielding a quantifiable change in the device’s optical spectrum [1,4,12]. This change is typically read out using benchtop-scale optical inputs (e.g., broadband optical source or tunable laser) and outputs (e.g., spectrum analyzer or photodetector) [12,13,14,15,16].

Interferometers, microring resonators (MRR), and Bragg gratings (Figure 1b) are among the SiP biosensing architectures that have been demonstrated for disease biomarker detection at concentrations down to the pg/mL scale [17,18]. Readers are directed elsewhere [1] for a detailed description of the principles of operation of each of these sensing architectures. Porous silicon sensors, which are fabricated with electrochemically etched crystalline silicon, have also been widely used in Bragg reflector and PhC configurations for biosensing since the late 1990s and are compatible with many of the same functionalization approaches [19]. This review, however, will mainly focus on planar SiP sensors, which permit greater optical confinement and guidance. Dozens of these individually addressable planar SiP sensors can be fabricated on a single millimeter-scale chip [10]. This permits multiplexed sensing, which is the simultaneous detection of multiple analytes from a single sample. Some benefits afforded by multiplexed biosensing are (1) the opportunity to diagnose multiple conditions/diseases from the same sample, (2) more selective and reliable diagnosis of a single condition by using multiple biomarkers to inform decision-making [20,21,22], and (3) the opportunity to include controls and reference sensors (e.g., to control for temperature fluctuations) to improve measurement accuracy [23,24,25,26]. In addition to these benefits afforded by multiplexed functionalization with different bioreceptors, multiple sensors on the same chip with identical functionalization offer the benefit of replicate measurements to improve accuracy and replicability (e.g., serving as technical replicates allowing for exclusion of failed measurements and averaging out the effects of sensor-to-sensor variability and some assay issues) [27].

The process of functionalizing the sensor surface with biorecognition elements (also called bioreceptors) that selectively bind target analytes is essential to accurate SiP biosensing. The performance characteristics of the biosensor, such as sensitivity, reproducibility, and stability, are inextricably linked to the biofunctionalization chemistry [28]. Here, we broadly characterize biofunctionalization in terms of bioreceptor selection, bioreceptor immobilization strategy (attachment to the sensor surface), and biopatterning technique. Designing antifouling surface modifications is also often included in biofunctionalization procedures to prevent non-specific binding. However, this topic has been reviewed in detail elsewhere [29] and will not be a major focus of the current review.

Many different biofunctionalization strategies are available and should be carefully chosen and optimized to suit the application and sensor architecture. In general, the selected bioreceptor should have good selectivity toward the target analyte to ensure low cross-reactivity with non-target molecules in the sample, high affinity toward the target to achieve fast, sensitive detection, good stability to retain consistent binding activity over time, and reproducible production to ensure predictable and replicable sensor performance across batches/lots of reagents [30]. The strategy used to immobilize bioreceptors on the sensor must not damage the sensor surface or the bioreceptors, and it should be compatible with any system-level integration required for the sensor chips (e.g., chip-mounted lasers and detectors, photonic wire bonds, etc.). It should also allow for oriented bioreceptor immobilization to optimize target accessibility and binding activity, permit uniform bioreceptor coverage on the sensor surface to ensure predictable and consistent target binding across all active sensing areas, have good stability to prevent bioreceptor detachment, and be reproducible [29,31]. The patterning strategy refers to the method by which bioreceptors are deposited on specific locations of the sensor surface (Figure 1c). This is required for multiplexed sensing and to confine bioreceptors to active sensing areas, thus preventing target depletion from dilute samples during sensing [32,33,34]. The selected patterning technique should not damage the sensor surface or bioreceptors. It should also have sufficient resolution for the selected application, be multiplexable so multiple different bioreceptors can be handled and deposited on a single substrate, produce uniform patterns with good spot-to-spot reproducibility, be compatible with the immobilization protocol (e.g., patterning under conditions that preserve functional groups on the silicon surface), and have low reagent consumption to conserve costly and precious reagents.

In addition to the general biosensor functionalization needs outlined in the previous paragraph, SiP devices have unique needs that distinguish them from other biosensors. Many immobilization techniques (e.g., covalent crosslinking) and bioreceptor types (e.g., antibodies, aptamers, etc.) [35] are shared across an array of sensing technologies including lateral flow assays [36], electrochemical probes [37], piezoelectric sensors [38] and other optical sensors like SPR [39]. While these sensing technology applications can provide valuable insight to inform functionalization strategies for SiP devices, only some of the findings are relevant because they utilize a variety of surfaces including glass, paper, polymers, specialized membranes (nitrocellulose), quartz, nanomaterials, alloys, metals (gold), and ceramics. Here, we focus specifically on immobilization techniques for silicon, silicon nitride, and other like materials.

Among these other transducer types, SiP sensors likely share the most similarities with SPR sensors, which employ a similar evanescent field-based detection principle. Nevertheless, SiP and SPR sensors exhibit differences in their surface chemistries, evanescent field propagation distances, miniaturizability, and multiplexability, as summarized in Table 1 [4,11,12,40,41,42,43,44,45,46,47,48,49,50,51,52,53]. Due to these differences, SiP devices have unique biofunctionalization needs, providing the main motivation for this review. For example, as SiP surfaces typically consist of 90–220 nm-thick silicon or silicon nitride nanostructures patterned on a silicon dioxide substrate [11,40], while SPR sensors typically have gold surfaces, the efficient thiol self-assembled monolayer-based strategies often used to modify metallic SPR biosensor surfaces are not suitable for SiP devices; instead silane-based chemistries are typically used [31,41]. Another unique consideration is the evanescent field penetration depth. For SiP devices, this is ~40–200 nm, depending on waveguide geometry and polarization (Figure 1a) [4,12]. Consequently, SiP sensors require a very thin biofunctional layer that brings target analytes within ~40–200 nm of the sensor surface. The size of this refractive index-sensitive region must be considered when choosing both the biorecognition element and the immobilization chemistry.

Broadly, the more well-established field of SPR sensing offers a few advantages over SiP sensing. For example, SPR permits the use of simple thiol-based self-assembled monolayer functionalization strategies [29,31]. SPR variants (e.g., SPRi and LSPR) are also compatible with excitation via direct illumination and simple colorimetric readout, which are attractive for portable sensing [42,53,54]. Multimodal SPR-SERS (surface-enhanced Raman scattering) sensing is also possible for highly sensitive and reliable analyte detection [55,56,57], while multi-modal sensing strategies based on SiP still require further research and development [58,59]. Nevertheless, large-scale and low-cost production remains a challenge for widespread use of SPR-based sensors outside of the laboratory environment [12,53].

SiP biosensor chips, themselves, are uniquely suited to reliable point-of-care (POC) use owing to their ease of miniaturization, low cost, and ease of multiplexing [1,12]. POC biosensing not only permits accessible diagnosis in decentralized and resource-limited settings, but also facilitates treatment decision-making in situations like stroke and sepsis where rapid confirmation of clinical findings is required and conventional lab-based assays may be too time consuming [20,60]. Further, wearable sensors that can be interfaced with flexible electronics may permit real-time and noninvasive monitoring of physiologically relevant analytes (e.g., in sweat) [61,62]. However, one major challenge associated with the translation of SiP biosensors to POC applications is that SiP devices are typically operated with expensive benchtop-scale fluidics and optical readout systems [13]. Miniaturized system-level integration is possible in principle, though, and work to integrate SiP sensors with microfluidics, CMOS electronics, and on-chip lasers and detectors via photonic wire bonds is underway to produce low-cost and portable complete-system PCB-mounted sensors [13,63,64,65,66]. Another major challenge with this translation is biofunctionalization.

Given the potential of SiP devices for POC biosensing, a major focus of this work is benchmarking SiP biofunctionalization strategies against needs pertaining to their multiplexed use at the POC. Some of these needs include good environmental and temporal stability to ensure predictable performance after transport and storage at ambient conditions, scalability and manufacturability to permit large-scale deployment, low cost to ensure accessibility, compatibility with easy-to-collect biological samples, such as whole blood, urine, and saliva, and biopatterning resolution on the order of 10 µm to complement the sensor miniaturization afforded by SiP technologies [2]. Reusability is another desirable feature for POC devices that could further reduce sensing costs and improve the accessibility of diagnostic tests in remote and low-resource settings [30]. Chip-level integration of SiP sensors introduces additional biofunctionalization needs. Not only must the biofunctionalization workflow be compatible with the SiP chip architecture, but it also must be compatible with attached optical inputs/outputs and electronics. For example, the immobilization chemistry and patterning technique must not damage electrical or photonic wire bonds, chip-mounted lasers, or PCB materials. Additionally, the immobilized bioreceptors need to be stable through any processing and packaging that needs to be done after immobilization.

To date, numerous existing reviews provide an overview of SiP biosensing technologies, focusing largely on the transduction techniques [1,12,14,19,42,67], with limited discussion about surface biofunctionalization. Others have focused on a single class of bioreceptors for biosensing applications (e.g., antibodies [68,69], nucleic acid probes [70,71], and molecularly imprinted polymers [72]), often including discussion about immobilization chemistries specific to that bioreceptor, and others have focused solely on the comparison of multiple bioreceptor classes for biosensing [30,73]. Several reviews have provided detailed discussion about bioreceptor immobilization chemistries for SiP sensors [31,69] and other biosensing technologies [43,74,75,76]. A number of works have explored different patterning techniques for the preparation of microarrays and the multiplexed functionalization of biosensors [32,77,78]. Finally, some reviews have discussed at least two of the three key aspects of biofunctionalization (bioreceptor selection, bioreceptor immobilization strategy, and biopatterning technique) for SiP [29] and other sensor technologies (e.g., SPR [41,79,80] and electrochemical sensors [80,81]). Distinct from these existing works, the current review (1) focuses on the unique functionalization needs and strategies of multiplexed SiP biosensors, (2) discusses all three key aspects of biofunctionalization (bioreceptor selection, immobilization chemistry, and patterning technique) and how they are interrelated, and (3) includes a review of biofunctionalization strategies that have been previously implemented on SiP biosensors. To our knowledge, our review is the first contribution to comprehensively summarize and categorize the biofunctionalization strategies previously demonstrated for SiP biosensors (from 2005 to present) as well as present a critical analysis of the various existing (demonstrated on SiP) and potential (demonstrated on similar sensor types) strategies towards the goal of meeting the performance criteria most relevant to SiP biosensors.

Here, we benchmark biofunctionalization strategies against the needs outlined in Table 2, with specific focus placed on biosensor design for multiplexed POC use [82,83]. First, we critically discuss several bioreceptor classes as biorecognition elements for SiP biosensors. Examples of SiP biosensors employing these bioreceptors are highlighted, including their demonstrated sensing performance and assay format. Strategies for bioreceptor immobilization on SiP platforms are discussed along with their advantages and limitations, with particular focus on gold standard silane-mediated covalent chemistries. Finally, contact and contact-free techniques for patterning bioreceptors on SiP sensors are identified and their performance characteristics are discussed. This review aims to present a balanced discussion of the tradeoffs of a range of biofunctionalization strategies to help guide those designing SiP biosensors in selecting a biofunctionalization approach that meets the unique needs of their intended application.

## 2. Bioreceptors

In this section, we introduce several classes of bioreceptors that have been used for SiP sensor functionalization and benchmark them against performance criteria outlined in Table 2. A high-level comparison of these bioreceptors is provided in Table 3. We have included subsections for each bioreceptor class to provide details about the opportunities and tradeoffs associated with each of these bioreceptors. For each bioreceptor class, tables summarizing their key advantages and limitations, and categorizing their use in SiP sensor functionalization approaches demonstrated in the previous literature are provided. Because strategies to improve sensitivity, specificity, stability, and other performance metrics are in many cases dependent on the bioreceptor class, within each subsection we have outlined strategies for these types of improvements as well as provided comparisons with other classes where relevant and available. Where appropriate, comparisons between bioreceptor subtypes are also tabulated according to these performance metrics. 

### 2.1. Antibodies

Antibodies (Figure 2) are the most commonly used bioreceptors for diagnostic assays [91,153]. Antibodies are Y-shaped proteins of ~150 kDa in size, which consist of two identical Fab regions (fragment, antigen-binding), and a single Fc region (fragment, crystallizable) [30,33,87]. The Fab regions specifically bind with high affinity to target molecules called antigens via binding sites called epitopes on the antigen surface. Antigens comprise a diverse range of biological molecules including simple sugars, hormones and lipids, complex macromolecules like proteins, nucleic acids, phospholipids and carbohydrates, and even viruses and bacteria [29,33,84]. On the other hand, the Fc region typically interacts with effector molecules and cells in biological systems and may be targeted for antibody immobilization on a solid substrate in biosensing applications [33,87,154]. Millions of antibodies have been validated for tens of thousands of antigen targets, making them a widely-available and flexible bioreceptor option for many different use cases [92,93,94,95]. Antibody production starts by immunizing animals against an antigen to stimulate the production of antigen-specific antibodies by the animals’ B cells [88,155]. Then, the antibodies can be obtained directly from the animal immune-sera. Alternatively, antibody-producing B cells can be immortalized by fusion with hybridoma cells for long-term production.

There are two major classes of antibodies: polyclonal and monoclonal. Polyclonal antibodies are produced as heterogeneous mixtures from animal serum and individual antibodies in a serum sample may bind to various epitopes on a single antigen [87]. Polyclonal antibodies exhibit significant batch-to-batch variability, partly owing to their animal origin [156]. Antibody quality can vary from animal-to-animal and even throughout an individual animal’s lifetime [156]. Conversely, monoclonal antibodies are produced from immortalized cell lines, are homogeneous in nature, and bind to a single epitope on the target antigen surface [88,156]. Monoclonal antibodies offer excellent specificity and reduced cross-reactivity and variability compared to their polyclonal counterparts; as a result, monoclonal antibodies have been widely used in diagnostic assay applications [86,87,88,90]. More recently, molecular engineering has also been used to generate shorter antibody variants including Fabs, single chain variable fragments, and single domain antibodies that can be produced more easily in vitro and used for applications that solely require epitope binding [29,75,157]. A comparison of polyclonal antibodies, monoclonal antibodies, and Fab fragments as bioreceptors for SiP biosensors is provided in Table 4.

Numerous SiP biosensing platforms using antibodies as bioreceptors have been reported in the literature. Conventional ELISAs are typically done in sandwich or competitive assay formats, requiring labeled secondary antibodies or labeled analyte molecules, respectively [160]. SiP platforms, however, permit label-free assays [1]. In the label-free format, binding of a target analyte to surface-bound antibodies is directly monitored, offering the advantages of real-time detection and simple sample preparation [14,161]. Nevertheless, sandwich formats using an unlabeled secondary antibody [18] or labeled antibody combined with subsequent enzymatic amplification [17,162] or protein-based multilayer signal enhancement [163] have been used to achieve more sensitive and specific detection for low-concentration and low-molecular weight analytes. To tether the capture antibodies to the sensor, these antibody-based SiP platforms typically rely on randomly oriented covalent immobilization strategies that target abundant amine or carboxyl groups on the antibody surface [75]. However, other covalent and non-covalent immobilization strategies have also been used [75]. 

SiP biosensors using antibodies as bioreceptors (Table 5) have been proposed for the biomarker-based diagnosis of cancer [17,18,22,161,163], cardiac disorders [164,165], inflammation [166], and viral infection [167], in addition to the detection of toxins [25,168], viral particles [169,170,171], and bacteria [172]. Such antibody-based SiP platforms have achieved LoDs as low as the pg/mL range using enzymatically or layer-by-layer-enhanced sandwich assay formats [17,163]. Other antibody-based SiP platforms have achieved label-free analyte detection with LoDs in the low-ng/mL range [161,169]. While most of the aforementioned examples employ whole polyclonal or monoclonal antibodies, Chalyan et al. [25] functionalized thiolated silicon oxynitride microring resonators with Fab fragments obtained from protease digestion of polyclonal antibodies for the detection of a carcinogenic mycotoxin, Aflatoxin M1, with a LoD of ~5 nM. The functionalization strategy used in this work targeted sulfhydryl (–SH) groups present on the Fab surface that were liberated from splitting the intact antibody; since these sulfhydryl groups are located opposite to the antigen-binding sites, this strategy ensures highly oriented bioreceptor immobilization, making it an attractive alternative to amine- and carboxyl-targeting strategies [75,173]. Shia and Bailey [168] functionalized silicon microring resonators with recombinantly derived single domain antibodies for the detection of ricin, a lethal protein toxin. The single domain antibodies exhibited improved specificity and lower cross-reactivity compared to a commercial polyclonal anti-ricin antibody.

Despite their excellent sensitivity and specificity, antibody-based biosensors present notable challenges regarding POC sensing. Namely, antibody discovery is achieved by months-long in vivo screening processes, which are expensive and laborious [89]. Antibody production largely relies on mammalian cell lines, which means that these bioreceptors are costly and require highly trained personnel to produce, precluding their use in highly scalable and low-cost sensors [2,97,98,99,157]. Moreover, among antibody vendors, there is a lack of consistency in the context-specific validation and reporting of antibody specificity and reproducibility for different applications [92,156,178]. The use of animals and cell colonies in antibody production makes these bioreceptors susceptible to sample contamination [89]. This means that choosing successful antibodies for biosensors is often an expensive and time-consuming task involving troubleshooting and returning failed antibodies to suppliers [156,178]. Antibodies are also susceptible to denaturation and require carefully controlled storage conditions, which may be difficult to maintain in POC settings [24,91]. Further, antibody immobilization on a solid substrate is known to reduce antibody binding activity, making the optimization of immobilization strategies using mild chemistries a particular challenge in the design of highly sensitive biosensors [75]. The key advantages and limitations of antibodies as bioreceptors are highlighted in Table 6. Given the limitations of antibodies discussed here, several classes of synthetic affinity reagents have been developed as alternatives to antibodies and have been demonstrated as bioreceptors on SiP platforms [2].

### 2.2. Aptamers

Aptamers, which have been referred to as “synthetic antibodies”, are short, single-stranded DNA or RNA molecules that are systematically selected to bind to a given target molecule (Figure 3) [87,89]. These single-stranded oligonucleotides fold into unique sequence-specific three-dimensional structures that bind to targets with high specificity and affinity via non-covalent effects, including electrostatic interactions, van der Waals, and hydrogen bonding [89,100]. Aptamers are generated using an in vitro process called SELEX (systematic evolution of ligands by exponential enrichment), which allows for the selection of unique target-binding DNA or RNA molecules from a large library (Figure 3c) [100]. The SELEX process begins with a library of around 10^15^ single-stranded oligonucleotides, each containing a different random sequence of 20–60 nucleotides, flanked by fixed sequences on the 3′ and 5′ ends [89,100]. This library is amplified by the polymerase chain reaction (PCR), then strand-separated to yield ssDNA or transcribed to yield RNA, depending on whether a DNA or RNA aptamer is desired [100,102,179]. These amplified products are then incubated with target molecules and target-bound DNA or RNA are separated from unbound sequences, followed by elution of the bound species. The amplification and target-binding stages of this process are repeated with the enriched pool of target-binding sequences. The process is repeated for a total of 8–20 cycles during which competitive binding causes high-affinity binding sequences to outcompete lower-affinity ones, eventually yielding a pool dominated by sequences with the strongest affinity to the target [100,101,102,179]. An additional negative selection step can also be included in the SELEX process to reduce cross-reactivity of aptamers to structurally similar targets, thus enhancing selectivity [102]. The selected oligonucleotides can subsequently be sequenced and synthesized for analysis and use [100]. The resulting aptamers can achieve comparable, or even better, affinity to their targets when compared to monoclonal antibodies, with typical dissociation constants (K_D_) in the low nanomolar to picomolar range [85,100,101].

Since their discovery three decades ago, aptamers have been generated against inorganic ions, metabolites, dyes, drugs, amino acids, peptides, proteins, cells, and even tissues [89,100,101,105]. Because the production of antibodies relies on the immune response, antibodies can only be generated for immunogenic and non-toxic targets [89,100]. Conversely, the in vitro SELEX process theoretically allows for the generation of aptamers against any target. Further, given the small size of aptamers (5–30 kDa) compared to antibodies (150–180 kDa), aptamers can be designed against small molecule targets that are inaccessible to antibodies [89]. In evanescent field-based sensing applications, the smaller size of aptamers can allow for greater surface immobilization density and can bring captured analytes closer to the sensor surface, potentially improving sensitivity [113,114]. The selection environment (e.g., buffer type, ionic strength, pH, temperature, etc.) during aptamer generation can also be tailored to the binding conditions required for the intended use case [89,100,180]. This is contrasted to antibodies which are limited to target recognition under physiological conditions. 

Other advantages conferred to aptamers by the nature of the SELEX discovery process include fast discovery and low batch-to-batch variability [89]. While antibody discovery requires upward of 6 months, the SELEX process can be completed in a matter of days if high-throughput automated methods are used [89,102]. Additionally, since antibody synthesis relies on animals or cell cultures, batch-to-batch variability can be high; this variability is avoided in aptamer samples because they are generated via chemical synthesis procedures with a low risk of contamination [89]. Aptamers also exhibit better environmental stability, especially thermal stability, and long shelf lives compared to antibodies [89,105]. Namely, aptamers are resistant to high temperatures up to 95 °C and cycles of denaturation and renaturation, while they can also be lyophilized and stored at room temperature [89]. This makes aptamers attractive bioreceptors for point of care devices and opens opportunities for surface regeneration and reusable sensors [96,105]. Finally, aptamer discovery and manufacture are generally lower cost than for antibodies. For example, CamBio offers custom aptamer discovery down to USD 5000 per target [181]. After the aptamer has been selected and sequenced, it can be manufactured at low cost using common oligonucleotide synthesis techniques. For example, Aptagen offers aptamer manufacture at USD 1–4 per milligram for microgram-scale synthesis, US $300 per gram for milligram-scale synthesis, and USD 50 per gram for gram-scale synthesis, while IDT offers DNA oligonucleotide synthesis at CAD 1.40–2.40 per base for 1 µmol quantities for sequences of 5–100 bases in length [89,182]. However, the manufacture of RNA sequences, especially those exceeding 60 bases can be more costly. For example, Bio-Synthesis, Inc. manufactures RNA sequences of 10–30 bases in length for USD 14.50–50 per milligram for 50–1000 mg-scale synthesis, while IDT manufactures RNA sequences of 5–60 bases in length for CAD 24.00 per base at 1 µmol quantities and RNA sequences of 60–120 bases in length for CAD 23.00 per base at 80 nmol quantities [89,106,182]. Table 7 provides a high-level comparison of DNA and RNA aptamers for SiP biosensing.

Table 8 summarizes aptamer-functionalized SiP biosensors that have been demonstrated in the literature. In all of these aptamer-based SiP sensors, label-free sensing formats were used. Park et al. [24] demonstrated IgE and thrombin detection on an aptamer-functionalized silicon microring resonator and demonstrated reproducible surface regenerations for up to 10 cycles after IgE and thrombin binding using a NaOH solution. Byeon and Bailey [174] compared thrombin binding on aptamer-functionalized silicon microring resonators to antibody-functionalized resonators and demonstrated aptamer-functionalized surface regeneration using proteinase K. The authors found that the aptamer had a lower affinity toward thrombin (K_D_ = 8.2 nM) compared to the antibody (K_D_ = 3.3 nM), suggesting a poorer limit of detection for sensing applications relying on steady-state binding. However, the aptamer-functionalized sensors demonstrated faster thrombin-binding kinetics, which could produce a theoretically lower LoD for the aptamer-based sensor in applications that leverage binding kinetics measurements to generate a calibration curve (e.g., by linearly fitting the initial slope of the binding kinetics curve to quantify analyte concentration [161,184]). Christenson et al. [164] presented a comparative study in which aptamer- and antibody-functionalized Photonic Crystal-Total Internal Reflection biosensors were investigated for the detection of cardiac troponin I. The aptamer- and antibody-functionalized sensors achieved detection limits of 0.1 ng/mL and 0.01 ng/mL, respectively. While the aptamer-functionalized sensor demonstrated poorer sensitivity, both sensors achieved clinically relevant limits of detection, and the aptamer sensor was lower cost and did not require refrigeration during storage. Chalyan et al. [25] compared the performance of aptamer- and Fab-functionalized silicon oxynitride microring resonator biosensors for the detection of Aflatoxin M1. A limit of detection of 5 nM was reported for both the aptamer- and Fab-functionalized sensors, though the Fab-functionalized sensor was deemed preferable due to its superior reproducibility. Both Chalyan et al. [25] and Guider et al. [185] reported effective sensor regeneration after Aflatoxin M1 binding using glycine solutions. 

While aptamers offer notable advantages over antibodies in the context of POC diagnostics, they still face challenges such as degradation in biological fluids, low SELEX success rates, lower availability, and highly variable costs. Firstly, aptamers, especially RNA aptamers, are susceptible to nuclease degradation in biological fluids [100,102]. For example, in human serum, the half-life of an unmodified aptamer is about one minute [180]. This limits the use of unmodified aptamers as bioreceptors in diagnostic devices using blood or serum samples. RNA aptamers are also more susceptible to hydrolysis than DNA aptamers at pH >6 [183]. However, chemical modifications, such as the incorporation of 2′-fluoro or 2′-amino-modified nucleotides, are often introduced to aptamers either at the beginning of SELEX or during chemical synthesis to improve their resistance to nuclease degradation [89,186]. These types of modifications can increase an aptamer’s half-life in biological fluids to multiple days [180], but modifications introduced during and after SELEX can add complexity to the SELEX process or change the folding structure and binding properties of the aptamer, respectively [89]. As such, careful optimization is required to achieve effective nuclease resistance without compromising binding performance. 

Next, the success rate of SELEX aptamer generation is lower than in vivo antibody generation, likely due to the lower structural diversity of nucleotides compared to amino acids and the small size of aptamers [101,180]. This increases the time and resources required to optimize aptamers for new targets. However, this <30% SELEX success rate could be improved through the use of specialized SELEX technology variants, personalized protocols, optimized oligonucleotide libraries, and quality control measures [180,187]. The target-binding performance of an aptamer depends on its structural conformation, which can be influenced by pH, ionic strength, and temperature [180]. Therefore, to ensure predictable binding, aptamer selection must be carried out in buffer systems similar to those used in the final application. However, this may also mean that an aptamer that performs well in solutions of a purified target in buffer may not perform as well in complex biological samples. Lastly, aptamers lack the type of extensive commercial infrastructure and investment seen in the antibody market and usually must be custom-synthesized by a handful of companies [89]. A summary of the key advantages and limitations of aptamers as bioreceptors is provided in Table 9.

### 2.3. Nucleic Acid Probes (Hybridization-Based Sensing)

Short, single-stranded nucleic acid probes have been widely used for the detection of nucleic acid targets via hybridization-based SiP sensing (Figure 4) [80,107,188]. Both ssDNA and RNA sequences can be immobilized on a biosensor surface, where they bind complementary nucleic acid target sequences through hydrogen bond formation, yielding DNA-DNA, DNA-RNA, or RNA-RNA duplexes [33,70,189]. Such biosensors are often called genosensors [81]. Compared to aptamers, which can be designed to bind many different types of target molecules, nucleic acid probes can only bind other nucleic acids [30]. Additionally, the function of nucleic acid probes depends primarily on their nucleotide sequence, not on their three dimensional structure: once the target gene sequence is known, the complementary probe can be designed directly [30]. This means that nucleic acid probes can be designed against a new target very quickly compared to antibodies and aptamers. Short nucleic acid probes of 100 nucleotides or less can be synthesized using well-characterized phosphoramidite chemistry [103,104,111,112]. This synthetic method of nucleic acid synthesis is highly reproducible, allows for the incorporation of functional groups like thiols and amines to aid in probe immobilization on solid substrates, and is typically low-cost [81,111,112,190]. Another key advantage of nucleic acid probe-based biosensors is that they can be thermally or chemically regenerated with good reproducibility between sensing cycles [80]. 

In addition to conventional ssDNA and RNA probes, synthetic nucleic acid analogues with functional chemical modifications to improve binding performance and biostability have recently been explored for biosensing applications. These include peptide nucleic acids (PNAs), locked nucleic acids (LNAs), and morpholinos [23,30,73,81,110,115,116,117,191,192]. PNAs (Figure 4d) are synthetic DNA mimics that can hybridize to complementary DNA and RNA, but have a backbone consisting of *N*-(2-aminoethyl)-glycine units linked by peptide bonds, rather than the sugar-phosphate backbone usually found in DNA [81]. Unlike natural nucleic acids, PNAs are uncharged, giving them improved hybridization stability [73]. Their hybridization stability is also impacted to a greater extent by single base mismatches than DNA-DNA hybridization, making PNAs more selective than DNA probes and a good choice for detecting single nucleotide polymorphisms [193]. PNAs also exhibit ionic insensitivity and improved pH, thermal, and enzymatic stability [73]. LNAs (Figure 4e) are another class of synthetic DNA mimics in which the ribose is locked in the 3′-endo conformation, resulting in reduced conformational flexibility, improved biostability, and enhanced binding affinity toward the target sequence [30,81,101]. Morpholinos (Figure 4f) are synthetic nucleic acid analogues in which the sugar-phosphate backbone is replaced by alternating morpholine rings, connected by phosphoramidite groups [110]. Morpholinos are uncharged and possess many of the same characteristics as PNAs, but morpholinos exhibit improved solubility, poorer stability at low pH, and improved flexibility of synthesis regarding sequence length, offering the opportunity to bind longer DNA and RNA target sequences, compared to PNAs [108]. Table 10 provides a comparison between these nucleic acid subtypes and benchmarks them against functionalization performance criteria for SiP biosensing.

Numerous SiP sensing platforms have been demonstrated in the literature using nucleic acids or nucleic acid analogues as biorecognition elements for the detection of ssDNA and RNA biomarkers with applications in the detection of cancer [23,192,194,195,196,197] and bacteria [198,199] (Table 11). Often, a label-free assay format is used on these sensing platforms. For example, Sepúlveda et al. [200] demonstrated label-free detection of short ssDNA targets down to 300 pM using a silicon nitride Mach-Zehnder interferometer sensor functionalized with ssDNA probes, while Shin et al. [197] demonstrated specific and label-free detection of longer ssDNA targets (>100 nucleotides) on ssDNA-functionalized silicon microring resonators down to 400 fmol, which corresponds to 16 µL of a 25 nM sample. A silicon nitride slot waveguide Mach-Zehnder interferometer functionalized with methylated ssDNA probes was demonstrated by Liu et al. [192] to quantify the methylation density of a DNA-based cancer biomarker at sample concentrations down to 1 fmol/µL or 1 nM. Nucleic acid-functionalized SiP sensors have also been used for microRNA detection, as demonstrated by Qavi and Bailey [194], who used a ssDNA-functionalized silicon MRR sensor for the rapid and label-free quantification of microRNAs. In this work, the authors reported a limit of detection of 150 fmol, which represented the minimum quantity of microRNA that could be reasonably detected in solution with the reported biosensor. Based on the supporting information provided for this work, this detection limit corresponded to a 75 µL analysis volume of 2 nM microRNA. Synthetic nucleic acid analogues have been demonstrated as receptors and targets for SiP sensors. Yousuf et al. [110] recently demonstrated the detection of short ssDNA targets on morpholino-functionalized suspended silicon microrings down to 250 pM, while Hu et al. [201] demonstrated PNA detection using ssDNA-functionalized planar SiP sensors.

In contrast to these label-free methods, Qavi et al. [109] amplified the detection of microRNA on a ssDNA-functionalized silicon microring resonator sensor using S9.6 anti-DNA:RNA antibodies. The S9.6 antibody selectively binds to DNA-RNA heteroduplexes and was shown here to effectively amplify the signal after microRNA hybridization, achieving a limit of detection of 350 amol, corresponding to 35 µL of a 10 pM microRNA sample. This was a 3-fold improvement compared to label-free microRNA detection on the same sensor. This work also demonstrated preliminary results demonstrating that LNA probes could be used to capture the microRNA targets, followed by successful, albeit slightly less effective, amplification with the S9.6 antibody. Kindt and Bailey [196] improved the limit of detection of a ssDNA-functionalized silicon microring resonator sensor for the detection of mRNA using streptavidin-coated beads. This bead-based amplification improved the sensor’s limit of detection to 512 amol, compared to 32 fmol without bead-based amplification. 

To date, most nucleic acid hybridization-based biosensors have been demonstrated for the detection of short target sequences due to the tendency of longer sequences to fold and obtain secondary structures [70,198]. These secondary structures significantly slow down binding kinetics, thus increasing sensing times. This challenge can be mitigated by pre-treating the targets via thermal denaturation, fragmentation, or the use of short nucleic acid chaperones which disrupt the nucleic acid target’s secondary structure [196,198]. In one work [198], the folded structures of long transfer-messenger RNA (tmRNA) targets were modified using one of the three following strategies prior to detection: (1) chemical fragmentation, (2) thermal denaturation, or (3) thermal denaturation in the presence of chaperone probes. Subsequently, the treated tmRNA targets were detected in a label-free format on ssDNA-functionalized silicon microring resonators. Chemical fragmentation was found to be the most effective RNA pre-treatment strategy for increasing the binding kinetics and magnitude of the sensor response. In another work [196], short DNA chaperone molecules were used to disrupt the secondary structure of full length mRNA transcripts prior to detection on ssDNA-functionalized silicon microring resonators. This effectively improved the sensing assay’s binding kinetics.

Indeed, the greatest limitation of nucleic acid-based bioreceptors is their limited applicability: they are only suitable for applications requiring nucleic acid targets [30]. Further, nucleic acid targets usually require significant sample preparation prior to detection [188]. For DNA targets, the sample usually must undergo fragmentation to ensure that the target sequence is accessible to the capture probes, followed by denaturation to yield single-stranded sequences. Depending on the abundance of the target, it may also require amplification through PCR or isothermal strategies prior to detection [188,195,199]. For RNA targets, sample preparation may be simpler, but still typically requires a fragmentation step [188]. Finally, DNA and RNA carry an inherent negative charge, making them susceptible to non-specific binding due to electrostatic interactions with non-target molecules [30]. This also poses challenges regarding nucleic acid probe immobilization. For example, nucleic acid probes are repelled by an unmodified SiP sensor’s negatively charged native oxide surface, which means that the SiP surface must be modified with a cationic film should passive adsorption be used for probe immobilization [204]. When covalent immobilization strategies are used, this negative charge increases steric hindrance between adjacent nucleic acid probes, which affects the maximum density of probes that can be immobilized on the sensor surface and the number of available binding sites for targets, potentially limiting sensor sensitivity [201]. This effect, however, can be reduced by employing in situ synthesis of nucleic acid probes on the SiP surface. Hu et al. [201] demonstrated a greater than 5-fold increase in ssDNA probe surface coverage and a greater than 5-fold increase in detection sensitivity for SiP microring resonators and photonic crystal sensors when functionalized via in situ probe synthesis, compared to the covalent immobilization of full ssDNA sequences. Conversely, if the immobilization strategy is optimized and the density of immobilized nucleic probes on the surface becomes too high, hybridization of targets to the surface-bound probes is hindered by steric crowding and electrostatic repulsion, also limiting sensor sensitivity [71]. As such, careful tuning of the spacing between immobilized probes is required for optimal performance. Some of these limitations can be mitigated by the use of uncharged synthetic DNA analogues including PNAs or morpholinos [30]. For example, in a study investigating DNA- and PNA-functionalized electrochemical sensors for the capture of DNA targets, the PNA-functionalized sensors exhibited stronger target capture and demonstrated optimal sensing performance at higher probe surface density than the DNA-functionalized sensors, likely due to reduced steric and electrostatic effects [205]. This contributed, in part, to a greater sensitivity for the PNA-functionalized sensor, which had a very wide dynamic range from pM to µM and a LoD that was 370 times lower than that achieved when using DNA probes. However, the lack of electrostatic repulsion between uncharged DNA analogues can lead to local clustering on the sensor surface, creating a heterogeneous layer of these uncharged probes, thus hindering the reproducibility of the functionalization strategy [193,206]. A summary of the key advantages and limitations of nucleic acid probes for SiP biosensing is provided in Table 12.

### 2.4. Molecularly Imprinted Polymers (MIPs)

Molecularly Imprinted Polymers (MIPs) are a type of label-free synthetic receptor for binding a broad spectrum of analytes from small molecules and viruses to larger proteins and cell membrane structures (Figure 5) [72]. The first imprinted polymers were developed in the early 1990s and demonstrated the ability to change impedance in response to target binding. Later, more developed MIP films exhibited changes in refractive index upon binding, making them ideal for optical sensors.

Several strategies for MIP preparation on SiP platforms and representative surfaces have been summarized in Table 13. MIPs are created via template assisted synthesis where an analyte is cured within a polymer making a 3D impression in the form of a binding pocket (Figure 6a) [119,208]. There are two main methods of MIP polymerization or “templating” for optical sensors: solution based (Figure 6b) or surface stamping (Figure 6c) [119,209]. In solution-based MIPs, a target, or template, is solvated in organic solvents with precursors, initiators, and monomers [72]. Smaller molecules are primarily used directly as a template, whereas larger targets (proteins, peptides, etc.) use a smaller binding epitope for imprinting. These formulations are specific to the template and form complexes of reversible covalent or noncovalent interactions with the template’s chemical structure. Next the solution is deposited on a surface and cured by ultraviolet (UV) or thermal polymerization. Solution-based MIPs can be templated onto many shapes such as coatings, thin films, and nanoparticles [210]. This is advantageous since they can conform to many different fiber and waveguide topologies. MIP films can be grown on a variety of photonic sensor designs, dipped on optical fibers, or developed in solution on microspheres [211]. Following MIP synthesis, the template molecules must be extracted, which is often achieved by washing or soaking in solution [211,212,213,214,215] or by plasma-treatment [211], though physically assisted solvent extraction (e.g., microwave- or ultrasound-assisted extraction) and extraction using supercritical or subcritical fluids have also been used [216]. This produces a distribution of exposed binding site geometries due to the template’s random orientation on the polymer surface (Figure 6b). Surface stamping using support molds was the first method of casting [209,217]. Template molecules are crosslinked to a surface mold and pressed onto the polymer surface over the sensor prior to curing. Removal of the mold leaves imprinted binding sites stamped on the surface of the polymer. This method produces more regular pockets in comparison to solution-based MIPs due to the added control over the depth of imprinted binding sites and the opportunity to control template orientation on the surface mold (Figure 6c) [218].

These methods produce specific binding pockets on the polymer surface that match the three-dimensional molecular structure of the template. Targets primarily bind via hydrogen bonding, electrostatic interactions, and Van der Waals forces. Reversible covalent bonding is less common since it is dependent on the template’s molecular structure, available specialized monomers, and more complex synthesis [220]. Direct adsorption of analytes into the binding pockets produces a change in refractive index or electrochemical (impedance) signal that can be read out by optical and amperometric sensors, respectively.

MIPs are considered an alternative to antibodies since they are highly sensitive, reversible and have both chemical and mechanical stability. They are synthetic making them robust, scalable, low-cost, and shelf-stable [118]. They have been shown to be stable over months in a large temperature range (up to 150 °C) with over 50 adsorption/desorption cycles in organic solvents, acids, and bases [221]. Divinylbenzene MIP bases are twice as robust (up to 100 cycles) in comparison to methacrylate- or acrylamide-based polymers over a larger pH range. Although MIPs are an excellent synthetic method of producing a non-refrigerated product with a long shelf life, there are several limitations to the technology. Currently, synthesis is developed for one target at a time and requires computational studies to downselect polymer precursors and benchtop chemistry to optimize formulation. Computational studies include quantum mechanics/molecular mechanics (QM/MM) calculations (ab initio, molecular dynamics, etc.) between possible precursors and the template molecule [222]. These calculations determine which reagents interact with the chemical structure of the template molecule. Then, MIPs are formulated based on the set and ratios of precursors are empirically tested. The final MIP formulation is selected to maximize sensitivity and specificity, based on these empirical data.

MIPs have limited specificity in complex solutions due to the imprinted nature of the polymers, which include an array of heterogeneous binding pocket orientations [30]. Smaller or like molecules can fill the binding pockets, producing a background signal or affecting the MIP’s affinity toward its target [223]. Formulations thus need to be thoroughly optimized for the template (as described above) and tested against non-imprinted polymers (NIPs) [224]. NIPs are the same composition as the MIPs, only formulated without the template. They are used as a control to determine the sensitivity of the MIPs against nonspecific adsorption. Further studies testing MIPs in real bioanalyte samples are essential to validate their specificity [30]. A summary of the advantages and limitations of MIPs as bioreceptors is provided in Table 14.

The use of refractive index sensing with MIPs in silicon photonics (Table 15) is limited, although they have been well demonstrated with SPR-based sensors [209]. MIPs can be drop-cast, spray-coated, spin-coated and inkjet printed on the sensor surface. Chen et al. [212] demonstrated thermally polymerized, drop-cast ultrathin film MIPs on a passive SOI microring resonator sensor for testosterone. This method is highly sensitive for sensing ultralow concentrations with a sensitivity of 4.803 nm/(ng·mL). First the template solution is premixed to promote self-assembly between the template and monomers specific to its chemical structure. This produces a pre-polymerized layer surrounding the template in solution that is further complexed with the addition of carboxyl-terminated monomers. This matrix is then drop-cast on the sensor’s surface and thermally treated for 12 h. The combination of the pre-polymerized matrix and dilute solution results in an ultrathin assembled monolayer of MIPs on the surface with a limit of detection of 48.7 pg/mL. Multiple cycles of MIP regeneration (using a 1:1 acetic acid-ethanol rinse) and sensing with a solution of 1 ng/mL testosterone were tested on this platform to assess reproducibility. There was a drift in the sensor response and corresponding decrease in sensitivity as the number of regenerations increased, which the authors attributed to damage to the MIP during testing. Selectivity was also assessed by introducing the small molecule toxin, microcystin-LR, to the sensor, which produced a negligible response.

Photopolymerization can be achieved all at once by direct UV polymerization or in stages by pre-polymerizing in a dilute crosslinking solution followed by the addition of a UV initiator for a final cure. Xie et al. [219] used this process with cascaded microring resonators for sensing progesterone. They used an SU-8 cladding and a slightly larger ring diameter to match the free spectral range of the reference ring to the MIP-coated sensing ring. The MIP is prepared by pre-polymerizing acetic and methacrylic acid with progesterone for 3 h followed by adding UV crosslinkers in a specialized tank for UV curing. This produces a thin self-assembled film on the sensor surface. Their results showed a limit of detection of 83.5 fg/mL which is approximately 3 orders of magnitude lower than enzyme-linked immunosorbent assays (ELISA). The sensor shows good selectivity to progesterone with little to no response with testosterone and the NIP. Eisner et al. [213] used MIP sol-gels to compare airbrush versus electrospray ionization deposition techniques. These sol-gels are formed by hydrolysis and polycondensation of a colloidal liquid into a gel at low temperatures. The colloid includes metal oxides, salts, or alkoxides suspended in solvents. This ceramic-based MIP was designed for the detection of trinitrotoluene (TNT) vapor and was coated on passive silicon racetrack resonators with thicknesses of 500–700 nm to minimize resonant wavelength shift artifacts due to changes in the bulk refractive index surrounding the MIP. The results showed a ~10× increase in response and sensitivity in the electrospray MIP in comparison to airbrushing. The MIP-coated sensors showed a nonspecific response to other nitro-based explosives (2,4-dinitrotoluene (DNT) and 1,3-dinitrobenzene (DNB)); however, the device’s sensitivity was about an order of magnitude greater for TNT than for DNT and DNB.

Hydrogel-based MIP thin films are less successful since they expand and contract based on water content and salinity, producing unwanted effects. Reddy et al. [225] sensed hemoglobin on silicon oxynitride waveguides for dual polarization interferometry. The gels initially increased in thickness and mass upon injection of a control solution, but the response was transient suggesting adsorption and desorption of the control on the hydrogel surface. In contrast, the target, hemoglobin, produced a continuous signal and remained selective in solutions containing <1% pooled bovine serum.

### 2.5. Peptides and Protein-Catalyzed Capture Agents

Synthetic and native peptides are an attractive method of capture for chemical and biological targets in SiP due to their small size in comparison to antibodies, aptamers, and other larger components (Figure 7) [226,227,228]. Peptides are differentiated from proteins by their size (2–70 amino acids) and flexible structure. There are two main types of peptides for attachment: native and synthetic [120]. Native peptides are small binding epitopes or ligands found in nature that selectively bind to a specific site on the target of interest. They are primarily recombinant and produced by cloning the peptide in an organism. The peptide sequence is inserted into a plasmid, expressed in bacteria, insect or mammalian cells and purified for processing [229].

Synthetic peptides are chemically synthesized using solid phase peptide synthesis (SPPS) or solution-based synthesis (SPS) [230]. Synthetic peptides are made using D-amino acids instead of the more naturally occurring L-amino acids seen in native peptides. D- and L-amino acids are enantiomers, or the same amino acid sequence with a mirror image structure. This change in configuration makes D-amino acids less susceptible to enzyme degradation without changing their biological function. SPS was the first synthesis method, developed in 1901, where a chain of amino acids is grown one residue at a time in solution [231]. SPPS followed in 1963 and uses a solid support for anchoring the peptide chain that enables washing steps between the addition of successive amino acids. Both methods start from a primary amino acid using selective protecting groups (FMOC, BOC) where successive amino acids are added in a step-by-step fashion to form a chain [232]. Generally, SPPS is the most common method since it is a well-established commercially available process and contributed to the Nobel Prize in Chemistry in 1984 [231]. Its use of a support and wash cycles results in a higher production of correctly formed peptides, removes reaction byproducts, as well as decreases the tendency of aggregation and incomplete reactions. However, SPS is still used since the lack of a support enables more challenging structures (cyclic), nonstandard components, and a larger array of coupling conditions (acidic, oxidative) [233].

Protein-catalyzed capture (PCC) agents are specialized, short (20 amino acid), synthetic peptides optimized to capture a target of interest [234]. They are considered “synthetic antibodies” due to their comparable high specificity and affinity for a target without the temperature sensitivity or stability issues common in enzymes, aptamers, and antibodies [235]. PCCs are highly selective since they are computationally designed based on the binding sites of proteins and other targets. Screening of chemical peptide libraries, such as one-bead one-compound (OBOC), identifies peptide components with high specificity and selectivity to the target of interest [236,237]. Due to this design, their affinity can be tailored to the specific dynamic range needed for sensing. Agnew et al. [234] evaluated the epitope binding sites and affinity of PCCs to those of monoclonal antibodies of the same target using principal component analysis. Their analysis covered 14 different protein targets as well as considered their physicochemical properties and molecular binding interactions. The results showed that PCCs are able to match and surpass antibody affinities with the majority of the binding driven by electrostatic interactions and hydrogen bonding.

In the literature, peptides and PCCs have been demonstrated as bioreceptors against antibodies [238], cancer cells [239], viral proteins [91], and streptavidin [240] on SiP platforms (Table 16). Angelopoulou et al. compared recombinant SARS-CoV-2 spike protein peptide on silicon nitride MZI sensors to conventional ELISA assays [238]. Silicon nitride MZI sensors were crosslinked via glutaraldehyde to the spike peptide against SARS-CoV-2 in a manner that selectively attached the peptides to only the silicon nitride waveguides and not the surrounding silicon dioxide. The reference was blocked with bovine serum albumin as a control for non-specific binding. The label-free peptide MZI showed a 80 ng/mL limit of detection and correlated with the ELISA results of 37 diluted serum samples. The addition of an antibody as a label improved the limit of detection to 20 ng/mL.

Martucci et al. [239] used idiotype peptides to determine the surface capture efficiency of tumor cells on silicon surfaces. Idiotype peptides are ligands from the binding site of receptors on the surface of immune cells that bind to antigens on the surface of lymphoma cells. They are specific to a subset of B-cells and can specifically identify lymphoma cells. The authors functionalized the surface of porous silicon microcavities by submerging in a 5% amino-terminated silane solution, crosslinking with a double N-succinimidyl terminated linker to crosslink to a primary amine on the peptide. The authors moved away from crosslinking antibodies to silicon surfaces since antibodies are known to have problems assembling monolayers in the same orientation due to their large size and multiple crosslinking sites [241,242]. Their results showed that the idiotype peptide covered 85% of the sensor’s surface with a uniform, oriented layer and had a detection efficiency of 8.5 × 10^−3^ cells/µm^2^.

PCCs are starting to become a more well-known method for biological sensing using silicon photonics due to their good stability and long shelf life. They are temperature stable, showing little to no change in affinity after heating to 90 °C, and resistance against protease degradation [91,243]. Layouni et al. [91] showed a PCC specific to Chikungunya virus E2 protein on porous silicon microcavities and with positive detection in response to 1 µM E2 viral protein. In addition, their results showed no statistical significance in sensor response between previously heated (90 °C, 1 h) and unheated PCCs. This stability was further confirmed by PCCs for vascular endothelial growth factor maintaining 81% of their affinity after 1 h using standard ELISA assays. Another work with porous silicon microcavities for PCC sensing of streptavidin showed detection of 5 µM streptavidin using PCCs immobilized via click chemistry crosslinking [240]. A summary of the advantages and limitations of peptides and PCCs as bioreceptors is provided in Table 17.

### 2.6. Glycans and Lectins

Both glycans and lectins have been employed as biosensor recognition elements on SiP devices (Figure 8). Glycans are carbohydrates which are covalently conjugated to proteins (glycoproteins) and lipids (glycolipids) [122,244]. In biological systems, glycoconjugates are typically found on cell surfaces, in the extracellular matrix, or in cellular secretions, and participate in intermolecular and cell–cell recognition events. Glycans consist of monosaccharides linked together in linear or branched structures by glycosidic bonds [244]. The diversity of their constituent monosaccharide residues and the position and configuration of their glycosidic bonds give glycans significant structural variability [128,244]. Lectins are non-immune proteins that recognize and bind glycoconjugates and non-conjugated glycans via carbohydrate recognition domains (CRD) [121,122,134]. Specific lectin-glycan binding is affinity-based and facilitated by hydrogen bonding, metal coordination, van der Waals and hydrophobic interactions [121]. The CRDs of lectins may target monosaccharide residues or they may show poor affinity toward monosaccharides and, instead, preferentially bind oligosaccharides based on their glycosidic linkages [121,122,244]. The affinity of individual CRD-glycan interactions are weak, with dissociation constants in the micromolar to millimolar range [121,122]. Multivalent binding between lectins and glycans, however, allows for higher-avidity interactions, with dissociation constants that are multiple orders of magnitude lower [122,123]. Namely, some lectins possess multiple CRDs that bind to multiple monosaccharide residues on a polysaccharide or to multiple proximal carbohydrates immobilized on a densely-coated solid substrate [121,122,123]; moreover, lectins can recognize homogeneous carbohydrate-coated surfaces or mixed glycan patches. Conversely, in the case of lectins with only one CRD, higher-avidity binding may be achieved by the clustering of many lectin molecules [122]. While many lectins have been identified and their glycan-binding characteristics have been characterized, these only encompass a small fraction of the diverse set of glycans that are found in nature [123]. Compared to proteins and nucleic acids, the functional study of glycans lags far behind [129].

Glycans can be immobilized easily on biosensor surfaces in an oriented manner; for example, the terminal amine group of a glycan derivative can be targeted for site-directed covalent amine coupling to a surface [244]. In comparison, lectins possess more complex structures, making oriented immobilization more challenging.

Homogeneous glycan samples for biosensing applications cannot be synthesized easily in large quantities using biological systems, making chemical and chemoenzymatic synthesis the preferred routes of production for structurally defined glycans and glycoconjugates [127,245]. Multi-milligram quantities of polysaccharides up to 50 mers in length can be rapidly and reproducibly synthesized and optionally conjugated to nonglycan entities, like proteins, to yield glycoconjugates [127,128]. Nevertheless, chemical glycan synthesis is in its infancy and is inherently more challenging than oligonucleotide and oligopeptide synthesis because glycans are often highly branched and their biosynthesis is not template-driven [129]. Chemical glycan synthesis requires the modification of one monosaccharide hydroxyl group at a time in the presence of many others and the careful control of glycosidic linkage positions [127]. Currently, the synthesis of complex and highly branched glycan structures remains a major challenge [129].

Lectins may be purified from various organisms, though yields, especially for animal-derived lectins, are often too low for practical use [130]. Consequently, recombinant techniques are usually required for the production of lectins in multi-gram quantities [130]. Notably, anti-carbohydrate antibodies can be generated for glycan capture, but, due to the poor immunogenicity of carbohydrates, these antibodies typically have poor affinities toward their targets and limited versatility, making lectins preferable for carbohydrate detection [121]. In comparison to antibodies, the cost of lectin production is also lower. However, similarly to antibodies, the commercial synthesis of lectins is cell-based, and samples may vary in purity, properties, availability, and activity within and between vendors [121]. An overall comparison of glycans and lectins as bioreceptors for SiP biosensors is detailed in Table 18.

Glycan-coupled SiP biosensors can be used for lectin capture and have applications in toxin [132] and virus [126] detection. For example, Ghasemi et al. [132] covalently immobilized GM1 ganglioside glycans on the surface of a TM mode silicon nitride microring resonator sensor for label-free detection of Cholera Toxin subunit B. The authors reported an absolute limit of detection of 400 ag, which corresponds to a surface coverage of 8 pg/mm^2^. Shang et al. [126] used an organophosphonate strategy to tether glycans and glycoproteins to silicon microring resonators for label-free detection of various lectins and norovirus-like particles. The authors reported a limit of detection of 250 ng/mL for the norovirus-like particles. The functionalized sensors also demonstrated excellent stability, retaining strong binding performance after one month of storage at ambient conditions and after multiple cycles of surface regenerations with high-salt and high- and low-pH solutions. Indeed, the good chemical stability of glycans, even at ambient and dry conditions for prolonged periods of time, is an attractive characteristic of glycan-conjugated biosensors [124,125]. Other publications have demonstrated glycan- and glycoconjugate-functionalized SiP sensors for the label-free detection of common lectins, with limits of detection down to the ng/mL range [133,246].

Given that various diseases, such as cancer, autoimmune diseases, infections, and chronic inflammatory diseases are associated with glycan aberrations, glycans are valuable disease biomarkers [121,129]. Lectin-coupled biosensors have, therefore, been proposed for glycan biomarker-based disease diagnosis [121,129]. While lectin-coupled SiP sensors have seldom been reported in the literature, Yaghoubi et al. [131] reported a lectin-coupled porous silicon sensor using reflectometric interference Fourier transform spectroscopy for label-free detection of bacteria. The authors functionalized sensors with three different lectins, concanavalin A (Con A), wheat germ agglutinin (WGA), and ulex europaeus agglutinin (UEA), and found that the Con A- and WGA-coupled sensors demonstrated the greatest binding affinities for *E. coli* and *S. aureus*, respectively and demonstrated limits of detection of approximately 10^3^ cells/mL. Table 19 provides a summary of SiP biosensors demonstrated in the literature that use glycans or lectins as bioreceptors.

The greatest limitation of biosensors using glycan-lectin binding is their specificity. Unlike antibodies, lectins often bind to more than one glycostructure and demonstrate broader specificity, thus requiring extensive selectivity and cross-reactivity characterization prior to use [121,123]. The poorer selectivity of glycan-lectin interactions complicates their detection in complex biological samples and makes it difficult to detect small aberrations in the target structure [121,244]. Moreover, the avidity of glycan-lectin interactions is highly variable and depends not only on the structure of the biomolecules, but also on their multivalency and packing density on the sensor surface [122]. While glycans offer simple oriented conjugation to sensor surfaces and improved stability compared to antibodies, the discovery and production of biologically relevant glycans, especially complex and highly branched ones, is limited by current structural characterization and synthesis techniques [129]. Commercially available glycans are very expensive, at roughly CAD 200–1200/10 µg [135], while custom glycan synthesis is also costly [247]. On the other hand, lectins can be characterized and produced using mature and cost-effective techniques, but these proteins suffer from the same batch and vendor variability and pH-, temperature-, and buffer-sensitivity issues as antibodies [121,136]. While lectin regeneration is possible, it is likely to result in activity loss [121]. Table 20 highlights the key advantages and limitations of glycans and lectins as bioreceptors for SIP biosensors. 

### 2.7. Other

#### 2.7.1. High Contrast Cleavage Detection (i.e., CRISPR Cleavage Detection)

Recently, high contrast cleavage detection (HCCD) has been proposed as a detection mechanism for optical biosensing employing CRISPR-associated proteins as the biorecognition elements [137,138,139]. This is a clustered regularly interspaced short palindromic repeats (CRISPR)-based biosensing approach that can be used for sensitive detection of nucleic acid (DNA or RNA) targets. CRISPR systems contain CRISPR-associated (Cas) proteins, which possess endonuclease activity to cleave targets via guide RNA [140,248]. Most reported CRISPR based biosensors use Cas9, Cas12, or Cas13 effectors, which demonstrate different cleavage activities [249]. Namely, CRISPR-Cas9 cleaves target dsDNA based on guidance from single guide RNA [249]. CRISPR-Cas12 captures target DNA that is complementary to its guide RNA, activating non-specific collateral cleavage (or trans-cleavage) of nearby ssDNA [250]. Similarly, CRISPR-Cas13 captures target RNA that is complementary to its guide RNA, activating non-specific collateral cleavage of nearby ssRNA [250]. In the HCCD technique, Cas12 or Cas13 effectors can be used [138].

Most SiP biosensors rely on affinity-based detection, whereby low-index bioanalytes are captured on the sensor surface upon introduction of the target analyte. The HCCD method, however, adopts a different architecture and relies on the removal of high-index contrast reporters from the sensor surface upon introduction of the target analyte (Figure 9). In HCCD, the sensor surface is first decorated with high-index contrast reporters, such as silicon nanoparticles, gold nanoparticles or quantum dots, tethered to the surface by single-stranded oligonucleotides [137,138,139,141,142,251]. Then, the analyte is combined with CRISPR-Cas12 or CRISPR-Cas13, which have guide RNA complementary to the target [137]. Once activated, these CRISPR-Cas complexes cleave the reporters from the surface, leading to a change in the local refractive index that can be transduced by the SiP device.

The first experimental implementation of this method was reported in 2021 by Layouni et al. [137] on a porous silicon interferometer platform. This was a proof-of-concept study in which the sensor surface was decorated with nucleic-acid-conjugated quantum dot reporters, then exposed to a DNase solution, which cleaved reporters from the surface. While this work did not report specific analyte detection, it demonstrated the ability to detect a large shift in the sensor’s reflectance peak upon enzyme-mediated removal of reporters from the porous silicon surface. This work paved the way for another preliminary study in which Liu et al. [139] demonstrated the detection of SARS-CoV-2 target DNA on a silicon microring resonator chip using HCCD (Table 21). The authors reported a ~8 nm blue shift in the resonance wavelength upon cleavage of gold nanoparticle reporters from the sensor surface by CRISPR-Cas12a activated by a 1 nM sample of target DNA in buffer solution. To our knowledge, SiP sensors using HCCD have yet to be demonstrated for the detection of nucleic acid targets in complex biological samples like whole blood, serum, and plasma. Chung et al. [251] proposed an inverse-designed waveguide-based integrated silicon photonic biosensor for HCCD-mediated sensing. However, this biosensing architecture has not yet been demonstrated experimentally.

The HCCD technique touts several advantages compared to traditional hybridization-based nucleic acid sensing. On a SiP sensor using hybridization-based sensing, signal generation relies on the small difference in refractive index between the sample buffer and the target nucleic acids. Typically, to achieve a detectable signal, the nucleic acid sample needs to be PCR amplified prior to detection or a secondary amplification molecule must be used after hybridization [138]. In HCCD, the refractive index contrast between the high-index reporters and background fluid is greater, leading to a greater signal change upon reporter removal compared to the binding of unlabeled targets [137,138]. Each activated CRISPR-Cas complex can perform up to 10^4^ non-specific probe cleavages after activation, leading to multiplicative signal amplification, thus enhancing sensitivity [138,251]. Further, since HCCD relies on the removal of reporters from the surface, the SiP sensor experiences a blue shift in resonant frequency for a positive result; this is in contrast to affinity-based sensing in which a positive result causes a red shift. This means that HCCD is less susceptible to false positives caused by non-specific adsorption of biomolecules to the sensor surface [137]. Another beneficial feature of the HCCD method is that it derives its specificity from the CRISPR-Cas12 or -Cas13 complexes, which are activated in a highly specific manner by their nucleic acid targets. Since specificity is conferred by the CRISPR-Cas complexes rather than biomolecules immobilized on the sensor surface, there is an opportunity to develop universal reporter-functionalized SiP sensors which can be used with application-specific CRISPR-Cas reagents, thus reducing the costs of sensor development and production [138].

While the sensitivity of this detection strategy is bolstered by the collateral cleavage of the activated CRISPR-Cas complexes, this non-specific cleavage also makes multiplexing challenging. To the best of our knowledge, multiplexed nucleic acid detection based on HCCD has not yet been demonstrated. Another limitation of HCCD is that the irreversible cleavage of reporters from the SiP surface prevents facile regeneration of the functionalized sensor for repeated use. Further, HCCD is only suitable for the detection of nucleic acid targets, limiting its versatility. Finally, while the nucleic-acid-based reporter-modified surface has improved storage stability compared to antibody-functionalized surfaces, Cas enzyme activity is sensitive to storage conditions, complicating POC use [33]. This could potentially be addressed by lyophilizing the assay reagents [145]. Overall, while HCCD remains in its infancy and is yet to be validated for detection in complex media, this method addresses some of the limitations of hybridization-based nucleic acid detection schemes and offers potential as a highly sensitive and specific strategy for SiP sensing. Table 22 highlights the key advantages and limitations of HCCD.

#### 2.7.2. CRISPR-dCas9-Mediated Sensing

CRISPR-associated proteins have also been used as a biorecognition element for signal amplification in silicon photonic sensors, in combination with nucleic acid probes. In 2018, Koo et al. [143] proposed a CRISPR-dCas9-mediated SiP biosensor for highly specific and sensitive detection of pathogenic DNA and RNA fragments for the diagnosis of tick-borne diseases. Broadly, this sensing method relies on twofold signal enhancement. Firstly, recombinase polymerase amplification (RPA) is used to amplify nucleic acid targets. RPA is a rapid enzyme-mediated DNA amplification technique that can be completed isothermally at mild temperatures [147,252]. This isothermal strategy obviates the need for power-intensive thermal cycling, which is required for conventional DNA amplification via PCR [147]. As such, RPA has been identified as an attractive alternative for POC use [145]. Additionally, reverse transcriptase (RT) can be added to the RPA reagents to facilitate isothermal amplification of RNA targets and reverse transcription of cDNA from RNA [143,147]. Secondly, nuclease-deactivated Cas9 (dCas9) is used in this sensing method as a labeling molecule. Like its active form, dCas9 binds to target dsDNA based on guidance from a target-specific single guide RNA (sgRNA) sequence [143,249]. Unlike its active form, dCas9 cannot cleave target sequences.

Koo et al. demonstrated this sensing method on SiP microring resonator sensors for the detection of pathogenic DNA and RNA sequences for scrub typhus (ST) and severe fever with thrombocytopenia syndrome (SFTS), respectively (Table 23) [143]. The sensor surface was first functionalized with single-stranded nucleic acid probes, complementary to the target sequences (Figure 10a) [143]. RPA or RT-RPA reagents were prepared, then added to the pathogenic DNA or RNA samples, along with dCas9 effectors and sgRNA. This mixture was incubated on the sensor chip in acrylic wells at 38 °C (for DNA targets) or 43 °C (for RNA targets). During this on-chip incubation, three key events took place: (1) the target DNA or RNA was amplified via RPA or RT-RPA, respectively, (2) amplified targets bound to complementary probes immobilized on the sensor surface (Figure 10b) and (3) dCas9 effectors bound to the hybridized targets to increase the refractive index change associated with each bound target (Figure 10c).

The authors reported the detection of pathogenic DNA for ST with a detection limit of 0.54 aM and the detection of pathogenic RNA for SFTS with a detection limit of 0.63 aM [143]. The platform effectively discriminated between ST and SFTS in clinical blood serum samples in just 20 min. Indeed, this platform allows for exceptional sensitivity as a result of the aforementioned twofold signal enhancement. Further, specificity is ensured in three ways. Firstly, the nucleic acid probes immobilized on the sensor surface facilitate selective hybridization of complementary targets. Secondly, RPA or RT-RPA nucleic acid amplification is guided by primers to selectively amplify target sequences in the sample [252]. Thirdly, dCas9 solely binds to double-stranded target sequences based on sgRNA guidance, so dCas9 signal enhancement can only occur after targets have hybridized to complementary probes on the sensor surface. As such, this is a promising method for applications requiring highly sensitive detection of nucleic acid targets in complex samples.

To our knowledge, this is the only example of CRISPR-dCas9-mediated biosensing on a SiP platform in the literature. Because this sensing method uses dCas9, which does not demonstrate collateral cleavage, it may offer more straight-forward multiplexing compared to the HCCD technique, but at the cost of increased assay complexity [248,253]. Multiplexing may be possible if multiple microrings on a single chip are functionalized with different target-specific nucleic acid probes in a spatially defined manner, and multiple target-specific RPA primers and dCas9/sgRNA complexes are used [254].

Regarding costs, the short synthetic nucleic acid probes and CRISPR-Cas reagents required for this detection method can be produced at moderate cost, but the RPA reagents are more expensive [144,145,248,249,255]. For example, a single CRISPR-based diagnostics reaction involving RPA pre-amplification costs an estimated USD 0.61–5.00 in a laboratory setting, with RPA reagents making up the majority of this price [145,146,147]. Nevertheless, given the microlitre-scale reagent and sample volume requirements of SiP-based assays, these costs are unlikely to be prohibitive for POC use.

In this detection method, the sensor surface is prepared similarly to conventional nucleic acid hybridization-based biosensors, as described in Section 2.3, which allows for superior sensor stability compared to antibody-functionalized devices. However, one key limitation of this method is the requirement for many different assay reagents, including RPA enzymes and primers, dCas9 enzymes, and sgRNA. This increases the complexity of the assay preparation and requires environmentally controlled storage of the assay reagents, especially the enzymes, making POC use less feasible. However, lyophilization of environmentally sensitive reagents for transport and storage before use is a potential solution to this challenge [145]. The use of such a platform in a POC setting is further complicated by the need to implement careful thermal control over the RPA reaction. Finally, as is the case with classic nucleic acid hybridization-based biosensors, this platform only allows for the detection of nucleic acid targets, limiting its breadth of applications. A summary of the advantages and limitations of CRISPR-dCas9-mediated sensing as a biodetection technique for SiP biosensors is provided in Table 24.

#### 2.7.3. Lipid Nanodiscs

Lipid nanodisc-functionalized SiP sensors have been proposed to study signaling and interactions at cell membranes [148,149,150]. Lipid nanodiscs are 8–16 nm scale discoidal lipid bilayers, held together and made soluble by two encircling amphipathic protein belts, called membrane scaffold proteins (Figure 11) [150,151]. These nanodisc structures recapitulate the native cell membrane environment and allow for the precise control of lipid composition. This permits the study of biochemical processes that occur at cell membranes, and which require specific lipid compositions for full functionality [256]. Lipid nanodiscs also solubilize and stabilize membrane proteins, which typically demonstrate loss of activity and function outside of the phospholipid membrane environment [151]. Given that membrane proteins are involved in vital regulatory cell functions and are often the target of therapeutic drugs, lipid nanodiscs are a valuable tool for studying cell membrane interactions involving these proteins. Compared to other structures, such as liposomes and detergent-stabilized micelles, which are used to mimic the cell membrane environment, nanodiscs offer improved consistency, monodispersity, production yield, and control over lipid and protein composition [150,151].

SiP sensors are an appealing platform on which to investigate interactions between lipid nanodiscs and other biomolecules. The multiplexability of SiP sensors permits high-throughput screening of cell membrane interactions. Further, membrane proteins are challenging to produce and typically have low yields, making the low reagent volume requirements of SiP sensors particularly attractive [148]. Finally, nanodiscs physisorb directly onto silicon dioxide, permitting their facile immobilization onto the native oxide surfaces of silicon and silicon nitride waveguides [150].

In the literature, lipid nanodisc-functionalized silicon microring resonator sensors have been used to probe interactions between soluble proteins and lipids, glycolipids, and membrane proteins embedded in nanodiscs (Table 25) [148,149,150]. In a study by Sloan et al. [150], lipid nanodiscs prepared with varying compositions of the phospholipids, 1-palmitoyl-2-oleoyl-sn-glycero-3-phosphocholine (POPC) and 1-palitoyl-2-oleoyl-sn-glycero-3-[phospho-L-serine] (POPS), were used to probe the binding of annexin V, a lipid-binding protein. Nanodiscs prepared with glycolipids (1,2-dimyristoyl-sn-glycero-3-phosphocholine/monosialotetrahexosyl ganglioside, GM1), biotinylated lipids (*N*-(biotinoyl)-1,2-dipalmitoyl-sn-glycer-3-phoaphoethanolamine, biotin-DPPE), and enzymes (cytochrome P450 3A4, CYP3A4) were also used to probe binding interactions with cholera toxin subunit B, streptavidin, and anti-CYP3A4, respectively. A 4-plex assay was prepared by microfluidically patterning the sensor chip with POPS, GM1, biotin-DPPE, and CYP3A4 nanodiscs, then exposing the whole sensor surface to annexin V, CTB, streptavidin, and anti-CYP3A4 solutions in sequence. This multiplexed assay demonstrated effective binding with minimal cross-reactivity for each specific protein-nanodisc combination.

In another work by Muehl et al. [149], a SiP microring resonator platform was used to investigate interactions between four different blood clotting proteins (pro-thrombin, factor X, activated factor VII, and activated protein C) and lipid nanodiscs prepared with seven different binary lipid combinations of phosphatidylcholine (PC), phosphatidylserine (PS), and phosphatidic acid (PA). A 7-plex sensor was demonstrated using these seven nanodisc preparations to obtain dissociation constants for binding between the coagulation proteins and lipid surfaces. All of the coagulation proteins studied in this work bind in a Ca^2+^ manner, so the nanodisc-functionalized surfaces were regenerated with good replicability using a Ca^2+^-free buffer after protein binding. In a subsequent work, Medfisch et al. [148] used a SiP microring resonator platform to study the binding interactions of seven different protein clotting factors (prothrombin, activated factor VII, factor IX, factor X, activated protein C, protein S, and protein Z) and lipid nanodiscs prepared with nine different phospholipid compositions involving PS, phosphatidylethanolamine (PE), and PC. The effect of PE-PS lipid synergy on the membrane binding of clotting factors was investigated. Again, surface regeneration after binding events was achieved using Ca^2+^-free buffer.

So far, SiP sensors functionalized with lipid nanodiscs have demonstrated value in the study of binding interactions at cell membranes. This is in contrast with other classes of bioreceptors discussed in this review, which have primarily been proposed for toxin and pathogen detection and/or diagnostic applications. The nanodisc-protein interactions demonstrated by Muehl et al. [149] and Medfisch et al. [148] have limited specificity, with all of the investigated clotting proteins binding, albeit to different extents, to the lipid nanodisc-functionalized surfaces. Hence, these nanodisc-functionalized sensors are likely unsuitable for selective discrimination between multiple targets. The incorporation of embedded membrane proteins or glycolipids into the nanodiscs, however, may offer more selective detection of soluble proteins, as demonstrated by Sloan et al. [150]. Lipid nanodiscs are typically custom-synthesized in the laboratory setting, allowing for the precise control of lipid composition and membrane protein content; while this leverages the flexibility of lipid nanodiscs, it limits their accessibility for assay-development and widespread use [151]. Overall, nanodisc-functionalized SiP sensors offer an excellent opportunity for high-throughput laboratory-based cell membrane interaction studies, but their potential in POC diagnostics may be limited. A summary of the advantages and limitations of lipid nanodiscs as bioreceptors for SiP biosensors is provided in Table 26.

### 2.8. Summary and Future Directions

Given the myriad of potential applications for SiP biosensors and the complex trade-offs of each bioreceptor class, there is no simple formula for selecting an optimal bioreceptor as each class has its own set of advantages and limitations that must be balanced with the needs of the application. For studies specifically probing carbohydrate-protein or cell membrane interactions, the choice is simple, with glycans/lectins or lipid nanodiscs typically being the most appropriate options, respectively. For other applications, the choice of bioreceptor can initially be narrowed down based on compatibility with the target of interest (see Table 3). Beyond this, the specific functionalization needs for the application of interest must be identified and used to guide further bioreceptor selection. For example, for non-nucleic acid targets, one must choose between antibodies, aptamers, MIPs, PCCs, and peptides. For non-POC applications where stability, regenerability, and cost are less important, monoclonal antibodies may be a suitable option due to their widespread availability and good binding affinity and selectivity. For POC applications, antibodies may not be suitable, and the choice between aptamers, MIPs, PCCs, and peptides will likely depend on the availability of pre-designed and validated products for the target of interest, or access to the relevant expertise and resources to design a custom bioreceptor for the target of interest. Trade-offs between affinity, selectivity, and stability should also be considered as relevant to the desired application. For nucleic acid targets, nucleic acid probes may be the best option for applications where assay simplicity, cost, stability, and/or multiplexing are the most important considerations. The opportunity to choose between different nucleic acid analogues (e.g., DNA, RNA, PNA, LNA, morpholinos) and chemical modifications can be used to tailor the stability and affinity of the nucleic acid probes for the application of interest. Applications requiring exceptional sensitivity and selectivity may benefit from the use of the more complex and early-stage HCCD or CRISPR-dCas9-mediated sensing strategies.

Regarding future directions, further research and development are required to improve the availability of pre-designed synthetic antibody analogues (e.g., aptamers, MIPs, PCCs) against various biomarkers. The availability of successful MIP formulations may be enhanced by increased use of computational methods. Such computational methods can aid in the development and optimization of MIP formulations for targets of interest and reduce experimental effort by guiding researchers toward promising systems [119]. Future SiP biosensing studies should focus on biomarker detection in complex biological fluids to quantify bioreceptor selectivity and to ensure reliable detection performance when using clinically relevant samples. For instance, the validation of aptamers for target detection in complex biological samples is essential for their translation to real-world sensing applications due to the sensitivity of their three-dimensional conformation and binding affinity to the ionic strength and pH of the sample [180]. HCCD and CRISPR-dCas9-mediated sensing are in their infancy and future studies should focus on validating these strategies for sensing in complex biological samples. Moreover, future work should focus on multiplexing these CRISPR-based methods to enable simultaneous detection of multiple targets.

## 3. Bioreceptor Immobilization Strategies

The surface of unmodified SiP sensors consists of a native silicon dioxide layer, which grows on silicon and silicon nitride upon exposure to air and moisture [31,257]. This oxide surface is hydrophilic in character [258,259] and has a negative surface charge density above pH 3.9 [260]. Strategies for immobilizing bioreceptors on SiP devices generally rely on non-covalent interactions between bioreceptors and the native oxide surface or target surface silanol groups for covalent attachment. In this section we discuss bioreceptor immobilization strategies for SiP biofunctionalization, focusing on passive adsorption, bioaffinity binding, and covalent immobilization (Figure 12). We discuss methods relevant to antibody, aptamer, nucleic acid probe, peptide, PCC, glycan, lectin, and lipid nanodisc immobilization and present tables categorizing bioreceptor immobilization strategies that have been used in functionalization approaches in the previous literature. It should be noted that strategies discussed in the following subsections generally are not relevant to MIP-based bioreceptors, which are immobilized on SiP surfaces during synthesis via casting and/or in situ polymerization; as such, MIPs are not discussed in detail here. Table 27 provides a summary of bioreceptor immobilization strategies that have been employed on SiP devices in the literature and benchmarks these strategies against biofunctionalization needs for SiP biosensors.

### 3.1. Passive Adsorption

Adsorption (Figure 12a) is the fastest and simplest method by which bioreceptors can be immobilized on a biosensor surface [29,68,69,80]. Adsorption-based bioreceptor immobilization has been widely used, especially in preliminary demonstrations of novel sensing architectures [31]. Bioreceptors may adsorb to a bare or modified SiP surface due to electrostatic, hydrophobic, polar-polar or Van der Waals interactions, or some combination of these non-covalent interactions [29,262]. Nevertheless, covalent and affinity-based strategies are typically preferred to adsorption-based immobilization.

One major disadvantage of adsorption is that it provides little control over the orientation of immobilized bioreceptors [31,69,204,263,274]. This may render binding sites unavailable for target capture, reducing the target binding capacity and, therefore, the sensitivity of the sensor. This random orientation, combined with intermolecular interactions, may also lead to poor bioreceptor loading density on the sensor surface [263]. Adsorption-based immobilization may lead to reduced bioreceptor activity due to folding or denaturation. This is especially relevant for protein-based bioreceptors, like antibodies, which are known to denature when adsorbed to surfaces, potentially changing the structure of their Fab fragments and diminishing their antigen-binding capacity [29,69,262,263]. Further, adsorbed bioreceptors are susceptible to desorption, leading to poor surface stability [31,69,261]. This is particularly relevant when the sensor is operated in flow conditions or when surface regeneration involving the release of targets from the sensor for multiple cycles of reproducible binding is desired. For example, Jönsson et al. [261] demonstrated that antibodies physisorbed onto chemically modified silicon dioxide surfaces were unstable toward changes in the surrounding medium, demonstrating significant desorption upon exposure to low pH, low surface tension, detergent, urea, and high ionic strength solutions. Finally, surfaces allowing for strong adsorption of bioreceptors may also be amenable to the adsorption of other biomolecules present in a complex biological sample, such as blood, leading to non-specific adsorption and high background signals [33]. Similarly, if other proteins possessing higher adsorption affinities to the sensor surface are present in the fluid, the bioreceptors may leach off the sensor [75]. This, in turn, compromises the selectivity of the sensor.

Despite the numerous limitations of adsorption-based functionalization, SiP sensors functionalized with lipid nanodiscs have demonstrated good stability and selectivity using adsorptive immobilization [148,149,150]. Indeed, lipid nanodiscs are especially amenable to adsorption-based immobilization because, like lipid bilayers, they are known to adsorb well to silicon dioxide surfaces [275,276]. This allows for simple nanodisc immobilization without the need to chemically modify the SiP surface or the nanodiscs. These nanodisc-functionalized sensors were regenerated after target binding with good reproducibility and no appreciable nanodisc desorption using Ca^2+^-free buffer, indicating stable immobilization [148,149]. These sensors were also used for multiplexed detection of soluble proteins with minimal non-specific binding [150]. However, the native silicon dioxide surface of SiP sensors is negatively charged at physiological pHs, such as the buffered systems employed in these nanodisc studies, meaning that nanodiscs with lipid compositions containing a high percentage of anionic lipids show a lower affinity for the sensor surface, leading to poorer surface coverage [148,149]. Fortunately, this reduced affinity is predictable and could be counteracted, at least in part, by using higher spotting concentrations [149].

### 3.2. Bioaffinity-Based Immobilization

Bioaffinity-based receptor immobilization involves the creation of multiple non-covalent interactions between a bioreceptor and biomolecule(s) acting as linker to the substrate [41]. The sum of many weak interactions yields a strong link between the bioreceptor and the surface. Two of the most common bioaffinity-based receptor immobilization strategies used for SiP sensor functionalization involve antibody-binding proteins and the biotin-avidin system, both of which can achieve oriented bioreceptor immobilization [29,69,74,75].

Antibody-binding proteins, including Protein A, Protein G, Protein A/G and Protein L, have been widely used for the oriented capture of antibodies on biosensor surfaces [29,75,263,277]. Protein A is derived from *Staphylococcus aureus*, while Protein G is derived from *Streptococcus* species, and Protein L is derived from *Peptostreptococcus magnus* [75,277]. Both Proteins A and G reversibly bind the Fc region of antibodies, binding a maximum of two antibodies at a time, and have variable antibody-binding affinities that depend on the immunoglobulin (Ig) subclass and the species of origin [75]. Protein A can capture mammalian IgGs with dissociation constants as low as the 1–10 nM range, while Protein G can achieve slightly higher affinity capture of mouse and human IgGs with dissociation constants as low at the 0.1–1 nM range [278,279]. The oriented capture of antibodies by these proteins ensures that the antibody’s Fab fragments are accessible for antigen capture, significantly enhancing the functionalized surface’s antigen binding activity [277,278]. For example, Ikeda et al. [278] demonstrated a 4- to 5-fold increase in antigen binding capacity for antibodies immobilized on silicon wafers using Protein A, compared to antibodies immobilized via physisorption alone. This was attributed to the improved steric accessibility of the antigen binding sites of the well-oriented Protein A-immobilized antibodies. In addition to its Fc-binding regions, native Protein G has additional sites for albumin and cell surface binding; however, recombinant Protein G, containing only Fc-binding domains, has been produced using *E. coli* to prevent this nonspecific binding [74,75]. Protein A/G is a recombinant protein that contains the Fc-binding domains from both Protein A and G [74]. Similarly to Proteins A and G, Protein L also binds antibodies in an oriented manner, but instead of binding to the Fc region, Protein L binds to antibodies’ κ-light chains outside of the antigen-binding site with dissociation constants as low as ~10 nM [277,279]. As a result, Protein L can bind any class of antibody, in addition to Fab fragments, which lack an Fc region, though its binding affinity is species-specific [277,280]. Indeed, a significant challenge associated with antibody-binding protein-directed bioreceptor immobilization is this Ig subclass and/or species-based variation in antibody-binding affinity; further this technique cannot be used to immobilize any bioreceptors aside from antibodies.

While Proteins A, G, and L allow for optimal orientation of immobilized capture antibodies, oriented immobilization of these antibody-binding proteins remains a challenge [75]. Fortunately, these antibody-binding proteins have several high affinity binding sites for antibodies, making their orientation on sensor surfaces less critical [41]. In the literature, antibody-functionalized SiP microring resonator sensors have been prepared using Protein A physisorbed on the sensor surface (Figure 12c) [1,264]. It has been reported that Protein A adsorbs onto silicon surfaces in a two-step process to yield a ~3–4 nm-thick adlayer [259,264]. First, a monolayer of Protein A is rapidly adsorbed on the surface; this first monolayer is denatured due to very strong non-covalent binding to the surface. Next, a second and third monolayer of Protein A are slowly adsorbed on the surface; these layers consist of non-denatured proteins which retain their ability to effectively bind the Fc region of antibodies. This strategy of passive Protein A adsorption followed by oriented antibody capture was used on sub-wavelength grating SiP microring resonators by Flueckiger et al. [264] and Luan et al. [1] to immobilize anti-streptavidin for model streptavidin-binding assays.

Others have tagged antibody-binding proteins with small molecules or other proteins to achieve higher-affinity binding to SiP sensor substrates. For example, Ikeda et al. [278] constructed a fusion of Protein A and bacterial ribosomal protein L2, which is termed “Si-tag” and binds strongly to silicon dioxide surfaces [281]. The authors demonstrated that the fusion protein was strongly immobilized on silicon dioxide surfaces in an oriented manner with a dissociation constant of 0.31 nM. The Si-tagged protein A also strongly bound mouse IgGs with a dissociation constant of 3.8 nM. The fusion protein immobilized 30–70% more IgG compared to physisorption of IgGs on bare silicon dioxide surface. Further, the fusion protein-immobilized IgGs demonstrated a 4- to 5-fold increase in antigen binding performance compared to the physisorbed IgGs. This functionalization strategy was subsequently demonstrated on a SiP microring resonator platform [282]. Christenson et al. [164] leveraged the strong bioaffinity interaction between biotin and streptavidin to immobilize recombinant Protein G on silicon photonic crystal-total internal reflection sensors. In this work, the sensor surface was modified with silane-PEG-biotin molecules, followed by streptavidin, then biotinylated recombinant Protein G. Antibodies were immobilized on this surface and used to detect cardiac troponin I. Covalent immobilization of antibody-binding proteins to silicon-based substrates (Figure 12d) may also be facilitated via methods such as surface modification with silane and a crosslinker, followed by Protein A or G attachment, as demonstrated by Anderson et al. [283] or via click chemistry, as demonstrated by Seo et al. [277] on glass substrates.

Interactions between antibodies and antibody-binding proteins are reversible and can be disrupted by variations in pH [29,75]. This limits biosensor stability and complicates sensor regeneration because antigens cannot be easily eluted from the sensor surface without also removing the capture antibodies. In this way, sensor regeneration is possible, but requires that both the antigen and capture antibody be eluted from the antibody-binding protein-functionalized surface, followed by reapplication of the capture antibody for another round of detection [75,263,283]. For example, Seo et al. [277] covalently bound Protein A onto glass slides, followed by the immobilization of receptor antibodies (rabbit anti-goat IgGs). These antibody-functionalized slides were used to capture target antibodies (goat anti-human IgGs) and were then treated with a low pH glycine-HCl buffer to remove the receptor and target antibodies. After this wash step, only the covalently bound Protein A remained. The surfaces were then successfully regenerated for a second round of binding by reapplying the receptor antibodies. Similarly, Anderson et al. [283] covalently bound Protein A to silicon dioxide optical fibers, followed by the immobilization of capture antibodies (rabbit anti-goat IgGs). Then, the functionalized fibers were used to capture fluorescently labeled targets (Cy5.5-goat IgG). The surfaces were regenerated using a pH 2.5 glycine-HCl, 2% acetic acid solution, followed by re-application of the capture antibody. Four cycles of regeneration were performed successfully with no appreciable reduction in Protein A’s Fc-binding capacity. However, the authors also reported unsuccessful regeneration of Protein A and G for an assay detecting plague F1 antigen, showing that regeneration of antibody-binding proteins may depend on the selected capture antibody and antigen. Here, the necessary reapplication of the receptor antibody also increases the cost and complexity of sensor reuse compared to strategies in which the functionalized surface can be regenerated solely by the removal of the target.

Another common bioaffinity interaction coupling method used in biosensor functionalization is based on the biotin-avidin/streptavidin complex, whereby the sensor surface is coated with avidin or streptavidin and used to immobilized biotinylated receptors (Figure 12e) [41]. Biotin is a small vitamin and avidin is a glycoprotein found in egg whites, which contains four biotin binding sites [265]. The biotin-avidin interaction is one of the highest affinity non-covalent interactions known in biology, with a dissociation constant on the order of 10^−15^ M [75,80,265]. This nearly irreversible non-covalent interaction is extremely resistant to variations in temperature, buffer salt, pH, and the presence of denaturants and detergents [74,265]. Streptavidin is a biotin-binding protein, derived from *Streptomyces avidinii*, which shows similar biotin-binding activity to avidin [265]. Streptavidin, however, has a pI of 5, while avidin has a pI of 10.5; as such, streptavidin is less susceptible to nonspecific interactions at physiologic pH, often making it the preferred choice [80,265].

The high-affinity nature of the biotin-streptavidin interaction means that biosensor regeneration via target removal can be achieved without disrupting the link between the receptor and the surface [284]. This means that the sensor can be used for multiple cycles of target binding without reapplying receptors. For example, Choi et al. [285] functionalized silicon nitride chips for reflectometric interference spectroscopy by covalently linking biotin to the surface, followed by avidin, and biotinylated concanavalin A. This lectin-coupled chip was used to reproducibly capture ovalbumin, a glycoprotein, over multiple binding cycles by regenerating the surface with a 10 mM glycine-HCl (pH 1.5) solution, which removed captured glycoproteins, while leaving the lectin-functionalized surface intact. In another work [284], SPR surfaces were functionalized with a biotin analogue, desthiobiotin, followed by streptavidin, and biotinylated IgGs. The authors reported that the functionalized surface was stable throughout multiple cycles of regeneration with solutions commonly used for target removal from bioreceptors, including HCl, Na_2_CO_3_, glycine buffer, and SDS solutions. Lü et al. [286] functionalized optical fiber probes by covalently linking streptavidin to the exposed silicon dioxide core, followed by the oriented immobilization of 5′-biotinylated DNA probes. These surfaces were used to bind complementary DNA targets, followed by thermal regeneration via washing for 2 min in hybridization buffer at 70 °C, or chemical regeneration via washing in 4 M urea solution. The surfaces demonstrated no appreciable loss in hybridization ability over six cycles of thermal or chemical regeneration. Efforts have also been made to break biotin-streptavidin interactions for complete surface regeneration whereby receptors are completely removed from the surface [266,284]. This has been achieved using a pH 7 chemical buffer solution [266], sequential rinsing with free biotin, guanidinium thiocyanate, pepsin, and sodium dodecyl sulfate [284], and washing with water at 70 °C [287]. These strategies require the reapplication of biotinylated receptors and sometimes streptavidin/avidin between binding cycles, but also open the possibility for sensors to be reused with different bioreceptors for each cycle.

The biotin-avidin/streptavidin-based immobilization strategy is more flexible than antibody-binding proteins, in that many different classes of receptors can be tagged with biotin and immobilized on avidin/streptavidin-coated sensor surfaces. On SiP platforms, this biotin-avidin/streptavidin bioaffinity functionalization strategy has been used to immobilize antibodies [288,289] and nucleic acid probes for both hybridization sensing [203] and CRISPR-Cas-modulated high contrast cleavage detection [137,139]. Similarly, it has been used to immobilize lectins on a silicon nitride sensor using reflectometric interference spectroscopy as the transduction technique [285]. This strategy can achieve unoriented or oriented receptor immobilization. For antibodies, amine, carboxyl, sulfhydryl, and carbohydrate groups can all be targeted for biotinylation, depending on the choice of biotin derivative; this can lead to unoriented antibody capture in the case of amine and carboxyl targeting or oriented capture in the case of sulfhydryl or carbohydrate targeting [29,69,265]. Optimally oriented nucleic acid probe immobilization has been achieved through biotinylation at terminal groups [137,139,286].

### 3.3. Covalent Immobilization

Covalent strategies are the gold standard for bioreceptor immobilization on SiP biosensors. Covalent immobilization (Figure 12b) is versatile, robust, and can be used to tether many different types of bioreceptors to SiP surfaces, yielding irreversibly bound functional layers [29,31]. This irreversible immobilization is beneficial for stable sensor performance under flow conditions and across multiple cycles of regeneration. Covalent methods may yield a higher density of immobilized bioreceptors compared to physisorption and bioaffinity techniques, which may, in turn, increase sensitivity [263]. Designing and optimizing a suitable covalent immobilization chemistry, however, can complicate assay development and preparation. Surface pre-treatments, reagents, and reaction conditions must be carefully chosen to yield reproducible and homogeneously thin surface modifications, while avoiding damage to the biosensor surface and bioreceptors [31]. For the design of POC sensors, further considerations may include selecting a scalable chemistry and designing a workflow that is suitable for SiP chips integrated with electronics and optical inputs and outputs.

#### 3.3.1. Silane-Mediated Immobilization

Most covalent immobilization strategies for SiP sensors involve silanization with organosilanes. Organosilane-based methods have been used for antibody, aptamer, nucleic acid probe, glycan, and lectin immobilization on SiP devices. Silanes consist of a silicon atom bonded to four other constituents [290]. Organosilanes include silane reactive groups and at least one functional organic group. The silane reactive groups covalently couple to the sensor’s native oxide surface by forming siloxane linkages with surface hydroxyls (Figure 13) [31,290]. A surface pre-treatment step is typically performed prior to silane deposition to remove organic contaminants and increase the number of surface hydroxyl groups available for silane grafting (Figure 13a) [31,74]. This pre-treatment step, which often involves oxidation via piranha, UV radiation and ozone, or plasma treatment, is essential to improving the silane grafting density and reproducibility of silanization. The silanization reaction can be performed using solution- or vapor-phase processes, with solution-phase processes being more widely used on SiP devices. However, no consensus on optimal reagents or reaction conditions exists, with significant variations in solvent choice, reagent concentrations, reaction time, and reaction temperature existing in the literature. After the silane is attached to the sensor surface, the silane’s organic groups can react with other organic molecules to facilitate bioreceptor attachment. While it is possible to directly attach bioreceptors to the organosilane surface [25,172,185,291], it is more common to attach bioreceptors using a bifunctional crosslinker that is highly reactive toward both the silane and the bioreceptors, as the most commonly used organosilanes lack sufficient reactivity toward bioreceptors [29].

When attaching the bioreceptor, native reactive groups or non-native reactive groups introduced during synthesis are targeted for immobilization. These may include amine, carboxyl, thiol, or carbohydrate groups. The choice of targeted functional group affects the orientation of the immobilized bioreceptor. Antibodies, for example, possess native amines in their lysine residues and native carboxyls in their aspartate and glutamate residues [69]. These residues are abundant on the antibody surface, so targeting amines or carboxyls leads to unoriented antibody immobilization. Conversely, thiol groups present in cysteine residues of the hinge region can be targeted for site-directed antibody immobilization [69,75]. However, creating reactive thiol groups to target requires reduction of the hinge disulfide bonds, which may lead to undesired reduction of other disulfide bonds, potentially reducing the antibody’s activity toward its target [75]. Native carbohydrate moieties present in the Fc region of antibodies can also be targeted for oriented capture [69,75]. Synthetic bioreceptors, including aptamers, nucleic acid probes, and glycans can be immobilized on silanized SiP surfaces by targeting terminal amine or thiol groups introduced during synthesis; this allows for oriented immobilization.

The most commonly used silanes for SiP functionalization are aminosilanes, particularly 3-aminopropyltriethoxysilane (APTES) (Figure 13). Aminosilanes contain organic groups that terminate in a primary amine, which can be targeted by amine-reactive crosslinkers for bioreceptor conjugation [290]. In order to initiate the reaction between the silane reactive groups of APTES and the hydroxyl groups present on the SiP surface, APTES must be hydrolyzed by moisture or water (Figure 13b) [74,290]. In the literature, APTES silanization of SiP sensors has been performed in anhydrous solvents such as toluene [201,239,292], acetone [17,195,293,294], and ethanol [132,202]. In these reactions, APTES hydrolysis is initiated by trace amounts of moisture present in the solvent [74]. APTES silanization has also been performed on SiP sensors using aqueous reaction solutions that contain a small quantity of water (e.g., ~5%) to catalyze APTES hydrolysis, combined with an organic solvent, typically ethanol [23,24,161,166,192,194,197,199,295]. These aqueous reactions are simpler than anhydrous ones, as they typically do not require drying the solvent or carrying out the reaction in a rigorously controlled inert atmosphere and/or under reflux [290,296]. However, in aqueous solutions, APTES is susceptible to copolymerization in the liquid phase prior to attachment to the solid substrate [258,268,297]. This can lead to the formation of thick and uneven films containing large silane aggregates (Figure 13d). Consequently, using an anhydrous solvent or maintaining low water content (~0.1%) in the reaction solution may yield thinner and more uniform silane layers [261,297,298]. Aside from solvent choice, an APTES concentration of 1–5% is typically used for solution-phase deposition [166,195,292], while reaction times vary significantly from several minutes [166,195] to overnight [202]. An alternative approach is vapor-phase silanization, which has been used to create uniform monolayer aminosilane films on silicon substrates [261,267,268,298]. In vapor-phase techniques, APTES is hydrolyzed by atmospheric moisture [74]. Compared to solution-phase reactions, vapor-phase aminosilane deposition has been reported as more reproducible, less sensitive to reagent purity and atmospheric conditions, and less likely to deposit polymeric silane particles [261,267,268,269]. Further, vapor-phase silanization may be more suitable than solution-phase methods when functionalizing SiP chips integrated with chip-mounted electronics and optical inputs/outputs, as vapor-phase processes do not require solvents that may degrade PCB or photonic wire bond materials. The final step of APTES silanization is typically a curing step at elevated temperature, which aids in the removal of moisture and the formation of siloxane bonds between the silane and surface [192,197,202,290].

Once the SiP surface has been modified with an aminosilane, bioreceptors are covalently linked to the surface via functional linkers such as glutaraldehyde (GA), bis(sulfosuccinimidyl)suberate (BS^3^), or 1-ethyl-3-[3-(dimethylamino)propyl]-carbodiimide/N-hydroxysuccinimide (EDC/NHS) [74]. GA and BS^3^ are homobifunctional linkers, which crosslink amine groups on the silanized substrate to amine groups on the bioreceptor. GA contains aldehyde groups which form imine bonds with amines via the formation of Schiff bases [31,74]. GA has been used to link antibodies [170], amine-terminated aptamers [24], amine-terminated DNA probes [23,197,199], and amine-terminated morpholinos [110] to aminosilane-modified SiP sensors.

GA linking has also been combined with SiP surface modification strategies whereby hydrofluoric acid (HF) is used to produce primary amines on silicon nitride waveguide surfaces [177]. These HF crosslinking approaches are particularly attractive for use with silicon nitride waveguides since they can be designed so that the amines are only produced on the nitride and not on the surrounding oxide [299]. This method uses basic cleaning methods followed by a HF dip to produce primary amines on the waveguide surface without the need of an additional aminosilane surface coating step. Next the sensor is immersed in a 2.5% GA crosslinker solution and washed. Bañuls et al. [299] developed this process to increase and localize biotarget capture to waveguide surfaces. The authors hypothesized that oxide comprised 98% of their sensor surface area with only 2% of the surface belonging to the silicon nitride sensing waveguides. This suggested that non-selective bioreceptor immobilization would lead to the majority of the target being captured by bioreceptors immobilized outside the sensing region. To show selective attachment to silicon nitride, slot waveguide ring resonator biosensors were modified with BSA and anti-BSA using the HF/GA procedure. Their results showed a detection limit of 28 pg/mm^2^ for anti-BSA antibody immobilization on the surface and 16 pg/mm^2^ for BSA. A similar procedure was used by Angelopoulou et al. [238] who modified MZI sensors with HF and GA, then spotted mouse IgG on individual sensors using an inkjet printer for multiplexing, followed by incubation with fluorescently labeled goat anti-mouse IgG antibodies and washing steps. The authors tested this direct attachment method in comparison to physical adsorption of the bioreceptors on amine-terminated silane (APTES) coated waveguides. The silane protocol yielded fluorescently tagged antibodies attached to both the waveguides and the surrounding oxide, whereas the HF procedure only functionalized the silicon nitride waveguides (Figure 14). Next, both sensors were spotted with a peptide, Receptor Binding Domain (RBD) of SARS-CoV-2 Spike 1 protein, and a BSA blocking protein on the sensing and reference waveguides, respectively. The HF method produced well-coated waveguides with the response of the reference sensor showing little change compared to the baseline signal upon exposure to anti-RBD antibodies. In comparison, the APTES modified reference sensor response could be clearly distinguished from the baseline signal. This suggests that BSA did not fully coat the APTES coated waveguides.

BS^3^ consists of sulfo-NHS esters at either end of an 8-carbon spacer arm [300]. The NHS esters react with primary amines to form stable amide bonds. BS^3^ has been used to conjugate antibodies [17,166], amine-terminated DNA probes [195], and peptides [239] to APTES-modified SiP sensors. When applied to antibody immobilization, GA and BS^3^ target native amine functional groups that are abundant on the antibody surface, leading to random antibody orientation. Moreover, as these immobilization strategies target functional groups that are abundant on the antibody surface, they may result in the formation of multiple bonds between the antibody and the surface [74]. This may lead to conformational changes of the antibody and render binding sites inaccessible for target capture. As such, spacer molecules or hydrophilic polymers can be incorporated into the linking chemistry to reduce steric hindrance and the risk of bioreceptor denaturation. The hydrophilic polymer, oligo(ethylene glycol), which can be used for this purpose, has also shown antifouling properties with short chains (≤7 repeats), which create a less ordered surface and decrease non-specific adsorption [301]. When applied to bioreceptors modified with terminal amine groups, such as 5′ amine-modified aptamers or nucleic acid probes, linking strategies using GA and BS^3^ permit site-directed immobilization. Another notable limitation of homobifunctional crosslinkers like these is that they may form bridged structures where both reactive ends are linked to the substrate, limiting the number of binding sites available for bioreceptor attachment and thus reducing bioreceptor density (Figure 13e) [302]. This can be avoided with heterobifunctional crosslinkers. EDC/NHS is a heterobifunctional crosslinker combination using carbodiimide chemistry, which links carboxyl groups on the bioreceptor to amine groups on the silanized substrate via the formation of stable amide bonds [74,176]. This linker chemistry has been used to covalently attach antibodies to APTES-modified MRRs [166] and silicon photonic crystals [165]. Since this strategy targets abundant carboxyl groups, which are also abundant on antibody surfaces, it results in unoriented antibody immobilization and may cause conformational changes, as described above. A similar carbodiimide chemistry was used by Peserico et al. [202] in which an APTES-modified MRR chip was carboxylated with succinic anhydride, then EDC was used to covalently link 5′ amine-modified DNA probes to the carboxyl-presenting surface, this time in an oriented manner.

Despite their popularity, GA, BS^3^, and EDC/NHS linker chemistries pose reproducibility challenges. GA polymerizes in aqueous solutions and the extent and nature of this polymerization depends on the age of the solution and can be difficult to control and reproduce [300]. BS^3^ and EDC/NHS linker chemistries both involve NHS ester groups which rapidly hydrolyze in aqueous solutions [31,74]. This rapid hydrolysis competes with biomolecule conjugation and is highly sensitive to reaction conditions, hindering reproducibility and limiting the yield of the conjugation reaction.

Bioreceptor conjugation using SoluLink chemistry is another silane-based strategy which offers good reproducibility and has been extensively used on SiP devices, namely the commercial Genalyte MRR platform [31,295]. In the literature, this chemistry has been used to covalently immobilize antibodies [18,22,161,168,169,174,295], 5′ amine-modified aptamers [174], and 5′ amine-modified DNA probes [109,194,196,198,303] on MRRs. Using this strategy, the bioreceptor is reacted with succinimidyl-4-formylbenzamide (S-4FB), which targets primary amines via succinimide coupling. The substrate surface is either modified with an aminosilane, followed by reaction with 6-hydrazinonicotinamide (S-HyNic) [161,194], or the bare SiP surface is directly reacted with HyNic-silane [18,22,109,168,169,174,196,198,295,303]. The 4FB-conjugated bioreceptors are introduced to the HyNic-modified surface, leading to bioreceptor immobilization through hydrazone bond formation. This reaction proceeds slowly, but aniline can be used as a catalyst to increase the rate of reaction, improve bioreceptor loading on the substrate, and allow for lower reagent consumption [295]. Despite its good reproducibility, chemically modifying bioreceptors with 4FB prior to immobilization adds time and complexity to this technique. More recent demonstrations on the Genalyte platform have instead used APTES silanization and BS^3^ to immobilize unmodified amine-containing bioreceptors for simple and flexible assay design [17,27,195,304,305].

Others have used 3-mercaptopropyltrimethoxysilane (MPTMS) to install thiol groups on SiP sensor surfaces to mediate bioreceptor immobilization. Thiolated bioreceptors can be directly conjugated to MPTMS-modified surfaces without an intermediate crosslinker through the formation of disulfide bonds [31]. For example, Chalyan et al. [25] directly immobilized Fab fragments on a MPTMS-modified SiP sensor. In this work, the Fab fragments were generated from protease digestion of polyclonal antibodies, followed by the reduction of hinge disulfide bonds to generate reactive thiol groups [25,69]. A similar strategy omitting the protease digestion step can also be used for site-directed antibody capture on MPTMS-modified surfaces [306]. However, covalent immobilization via thiol-bearing cysteine residues, which are usually internal to the antibody structure, and the unintentional reduction of non-target disulfide bonds may disrupt antibody conformation and binding affinity [69,75]. In addition to antibodies, this thiol-directed covalent strategy has been used for nucleic acid probe immobilization. Sepúlveda et al. [200] modified silicon nitride Mach-Zehnder interferometer sensors with MPTMS, followed by covalent and oriented immobilization of 5′ thiol-modified ssDNA probes.

Bioreceptors that lack reactive thiols can also be conjugated to MPTMS-modified surfaces using maleimide linkers. For example, Xu et al. [175] covalently immobilized antibodies on a MPTMS-modified planar silicon nitride optical waveguide interferometric biosensor using m-maleimidobenzoyl-N-hydroxysuccinimide ester as a thiol-to-amine crosslinker. Ghasemi et al. [133] covalently attached amine-derivatized glycans to MPTMS-modified silicon nitride MRRs using a SM(PEG)12 linker. SM(PRG)12 contains a polyethylene glycol (PEG) chain terminated by NHS ester and maleimide reactive groups. As such, it acted as a heterobifunctional linker between the thiolated surface and amine-derivatized glycans, while the PEG chain prevented nonspecific interactions between non-target molecules and the sensor surface.

3-Glycidoxypropyltrimethoxysilane (GPTMS) is another silane that can mediate direct covalent immobilization of bioreceptors on SiP sensors. GPTMS installs epoxy groups on silicon surfaces, which are reactive toward amine, thiol, or hydroxyl groups [31,290]. Ramachandran et al. [172] conjugated monoclonal antibodies and 5′ amine-modified ssDNA probes to GPTMS-modified glass (Hydex) MRRs. Using this strategy, the bioreceptors were covalently linked to the surface via amine reactive groups, resulting in unoriented and oriented antibody and ssDNA probe capture, respectively. Chalyan et al. [25] and Guider et al. [185] covalently immobilized amine-terminated aptamers on GPTMS-modified silicon oxynitride MRRs in an oriented manner.

#### 3.3.2. Organophosphonate-Mediated Immobilization

Organophosphonate chemistry presents a promising alternative to silane chemistry. Compared to silanes, phosphonate films can achieve greater monolayer density, surface coverage, and stability, and have a lower tendency to form multilayered structures [270,307]. Shang et al. [126] demonstrated covalent immobilization of amine-bearing glycan and glycoprotein bioreceptors on silicon MRRs using an organophosphonate surface coating and an amine-vinyl sulfone linker (Figure 15). After treating the surface with piranha solution to increase the number of available surface hydroxyl groups for organophosphonate grafting, the sensor surface was coated with a monolayer of 11-hydroxyundecylphosphonic acid (UDPA). This was achieved using the “T-BAG” method whereby UDPA is adsorbed onto the substrate, then heated to 120–140 °C to activate the formation of covalent linkages [126,307]. After the sensors were modified with UDPA, divinyl sulfone (DVS) was used to link the hydroxyl-terminated organophosphonate film to the amine-bearing bioreceptors [126]. In this work, the MRRs demonstrated excellent stability and reproducibility across multiple cycles of chemical regeneration and long-term storage at ambient conditions. A similar strategy was used to functionalize the surface of silicon nanowires with cysteine-modified PNA oligonucleotides [270]. Here, 3-maleimidopropionic-acid-N-hydroxysuccinimidester was used instead of DVS as a heterobifunctional linker to attach the thiol-containing PNA oligonucleotides to the UDPA-modified nanowires. 

#### 3.3.3. Click Chemistry

Click chemistry is a widely used crosslinking technique for simple, fast, and selective attachments with high efficiency. This method is attractive for biorecognition components since it uses physiological reaction conditions (neutral pH, buffered solution). Briefly, click chemistry involves linking molecules via heteroatom links (C–X–C) [271]. There are three main click procedures based on Cu(I) catalyzed azide-alkyne, strain promoted azide-alkyne, and tetrazine-alkene ligation reactions. The reaction is simple, more efficient than EDC/NHS chemistry, selective to only click reagents, has many commercially available modular components, and is not sensitive to oxygen or water [271]. 

This method has been used to immobilize ssDNA probes [59] and PCCs [91,240] on the surfaces of silicon-based optical biosensors. Juan-Colás et al. [59] demonstrated a novel silicon electrophotonic biosensor consisting of silicon MRRs fabricated with a thin n-doped layer at their surface to combine high-Q-factor photonic ring resonance with electrochemical sensing (Figure 16). In this work, the MRRs were covalently functionalized with ssDNA probes using the popular copper-catalyzed azide-alkyne click reaction. Firstly, two electrophotonic MRRs fabricated on a single chip were modified by electrografting azidoaniline or ethynylaniline onto the rings to install azide or alkyne groups, respectively (Figure 16a). The two electrophotonic MRRs were individually addressable, allowing for site-directed electrografting of azide groups on one ring and alkyne groups on the other. Next, the copper-catalyzed azide-alkyne click reaction was performed to conjugate azide-modified ssDNA probes to the alkyne-modified ring and alkyne-modified ssDNA probes to the azide-modified ring (Figure 16b,c). This unique strategy permits high-density multiplexed functionalization with submicrometer- to micrometer-scale precision, though it is not suitable for traditional SiP sensors that lack electrochemical control. Click chemistry was also used by Cao et al. [240] and Layouni et al. [91] to covalently link PCCs to porous silicon surfaces. In these works, the surfaces were modified with alkyne moieties by thermal hydrolyzation with 1,8-nonadiyne, followed by copper-catalyzed azide alkyne cycloaddition to attach azide-modified PCCs. This method requires removal of the substrate’s native oxide layer by exposure to HF prior to hydrolyzation. Consequently, this method may not be suitable for SiP devices patterned with extremely fragile silicon structures like sub-wavelength gratings, which may be partially etched or delaminated upon exposure to HF. Overall, some of the key advantages of click chemistry compared to silane-mediated strategies are its insensitivity to oxygen and water and its chemoselectivity, which prevents side reactions with other bioreceptor functional groups and preserves bioreceptor activity [29,91,308]. However, a limitation is the requirement for prior surface and bioreceptor modification with functional tags, like azide and alkyne groups, which adds complexity to the functionalization process [29,234].

#### 3.3.4. UV-Crosslinking

Direct UV-crosslinking of nucleic-acid-based bioreceptors has been demonstrated on planar glass and silicon dioxide wafers. This is a simple and inexpensive method that could be extended to SiP biosensors. Gudnason et al. [273] linked poly(T)10-poly(C)10-tagged ssDNA probes to unmodified glass surfaces using UV light irradiation. The immobilized probes demonstrated similar hybridization efficiency when compared to ssDNA probes immobilized on an amino-silane surface via traditional chemical crosslinking. The UV-linked probes showed no appreciable decrease in hybridization performance after incubation in water at 100 °C for 20 min, demonstrating strong thermal stability. In this work, the hybridization assay was performed in PerfectHyb Plus buffer to obviate the need for a surface blocking step. A similar strategy was used by Chen et al. [272] to covalently link thrombin-binding DNA aptamers with poly(T)20 tails to unmodified glass and silicon dioxide wafer surfaces using UV irradiation, while maintaining strong target affinity. Note that in this work, thrombin binding was performed in the presence of BSA and Tween-20 surfactant to reduce non-specific binding of the target to unmodified regions of the substrates. This UV-linking strategy is both simple and rapid because it requires no prior chemical modification of the substrate. Additionally, the nucleic acid-based bioreceptors do not require chemical modifications with reactive functional groups, lowering synthesis costs. However, to our knowledge, this strategy has not yet been demonstrated on patterned SiP sensor surfaces.

Table 28, Table 29, Table 30, Table 31, Table 32, Table 33, Table 34 and Table 35 outline strategies that have been demonstrated on SiP sensors and representative surfaces for the immobilization of antibodies (Table 28), aptamers (Table 29), nucleic acid probes (Table 30), peptides and PCCs (Table 31), glycans and lectins (Table 32), HCCD reporters (Table 33), CRISPR-dCas9-mediated sensing probes (Table 34), and lipid nanodiscs (Table 35) in the previous literature.

### 3.4. Summary and Future Directions

We have discussed adsorption, bioaffinity, and covalent strategies for immobilizing bioreceptors on SiP surfaces. While adsorption-based strategies offer excellent simplicity, their poor stability and lack of control over bioreceptor orientation limit their suitability for SiP biosensing applications. However, novel polymeric coating materials, such as PAcrAm™ and AziGrip4™ from SuSoS AG, are available and replicably self-assemble as stable monolayers on silicon substrates by adsorption from solution [309,310,311]. These polymeric coatings have customizable functional binding groups and allow for covalent and electrostatic capture of bioreceptors on the adsorbed coating [309,310,311]. This may create the opportunity for bioreceptor immobilization with similar simplicity to passive adsorption, but with improved stability and more controllable bioreceptor orientation, making this a potentially valuable future research direction. To the best of our knowledge, such functionalization techniques have not yet been demonstrated on SiP platforms.

Bioaffinity and covalent strategies typically offer improved stability and control over bioreceptor orientation compared to adsorption, but at the cost of increased complexity [41]. Bioaffinity strategies involving antibody-binding proteins permit controlled antibody orientation, but have limited stability compared to biotin-based and covalent methods [265,278,279]. Covalent strategies, especially those using silanization, have been widely used on SiP platforms, as they can permit very stable and tailorable bioreceptor immobilization [29,31]. When designing a covalent immobilization protocol, surface pre-treatment must be carefully considered to ensure that the sensor surface is free of organic contaminants prior to applying the immobilization chemistry, and to activate surface functional groups (e.g., hydroxyls) that will be targeted by the immobilization chemistry [31,74]. Such pre-treatments improve grafting density on the sensor surface, while also improving the reproducibility of the immobilization protocol [74]. Pre-treatment approaches that have been used in SiP bioreceptor immobilization protocols, such as piranha, UV radiation and ozone, plasma, and HF treatments, have been comprehensively summarized in Table 28, Table 29, Table 30, Table 31, Table 32, Table 33, Table 34 and Table 35. Future work should focus on optimizing standardized silanization protocols that can be used for highly replicable, scalable, and robust surface modifications with limited silane aggregation. In parallel with future work focusing on the system-level integration of SiP sensors for POC use, immobilization chemistries that are compatible with these integrated sensor architectures should be designed and tested. For example, translating solution-phase surface modification protocols to vapor-phase ones may reduce the risk of damage to the sensing system during functionalization, while improving scalability, reproducibility, and film uniformity [261,267,268,269]. Immobilization strategies using UV-crosslinking of bioreceptors directly to unmodified surfaces should also be explored on SiP sensors as a potentially simple, low cost, and scalable immobilization technique [272,273].

In designing immobilization protocols, potential steric crowding effects should also be considered in the context of bioreceptor immobilization and target capture. For example, crowding of bioaffinity linkers on the sensor surface may hinder subsequent bioreceptor immobilization [289]. These steric effects can be counteracted by using a higher bioreceptor concentration in the immobilization protocol or by using long linking molecules to increase the distance between the sensor surface and bioreceptors, providing more flexibility for the receptors to optimize steric crowding. When using these longer linking molecules, however, the potential sensitivity trade-offs associated with moving the binding reaction farther away from the sensor surface should also be considered. Immobilization approaches using these longer linking molecules may be most suitable for SiP architectures with greater evanescent field penetration depths (e.g., those based on ultra-thin [40] or sub-wavelength grating [16,46,264] waveguides). Similarly, dense receptor packing on the sensor surface may not always enhance target binding. Steric hindrance effects due to target molecule binding can reduce the rate of the forward binding reaction for neighboring receptors and affect the dynamic range of the sensor [312]. Thus, these steric effects should be accounted for when optimizing bioreceptor immobilization protocols.

## 4. Patterning Techniques

In this section, we introduce several patterning techniques that can be used for SiP sensor functionalization and benchmark them against the critical patterning performance criteria relevant to SiP biosensing, as outlined in Table 2. A high-level comparison of these patterning techniques is provided in Table 36. The subsequent subsections provide further details about each patterning technique, outline their opportunities and limitations for multiplexed SiP biofunctionalization, and highlight demonstrations from the previous literature in which these patterning techniques have been used to deposit bioreceptors on SiP biosensors. For each patterning technique, tables categorizing these demonstrations from the previous literature are provided.

### 4.1. Microcontact Printing

Microcontact printing (µCP), also called microstamping, is a soft lithography method whereby geometrically defined 2D patterns of biomolecules are transferred to a substrate using an elastomeric stamp (Figure 17a) [320,321]. This technique has been used to prepare patterns of bioreceptors like antibodies [32,322], DNA [323,324,325], MIPs [326], and carbohydrates [327] on solid substrates. 

The first step of µCP is fabricating the elastomeric stamp. Polydimethylsiloxane (PDMS) is the most popular stamp material for µCP because it is easy to mold, flexible, chemically inert, and impermeable to biomolecules like proteins [32,320,328]. In µCP, the geometry of the stamp is defined by casting it in a master mold, prepared by photolithography or micromachining [320,321]. Once the stamp has been cast, it is “inked” with the bioreceptor solution to be deposited on the substrate. The ink adheres to the stamp via passive adsorption, which can be tuned by modifying the stamp’s surface wettability with plasma or ozone treatment [32,328]. The inked stamp can be dried prior to stamping or used wet [329]. Next, the stamp is contacted with the substrate under a load, which can be achieved robotically or using a micropositioner to ensure precise alignment. The stamp is removed, leaving behind a 2D pattern of bioreceptors. The transfer of ink from the stamp to the substrate depends on the differential wettability between the stamp and substrate; in particular, the substrate must have greater wettability and, therefore, greater affinity toward the ink compared to the stamp [32,316].

A notable advantage of µCP is its excellent resolution. Patterns with critical dimensions down to 0.1–0.5 µm can be achieved [32,77]. This resolution is more than sufficient for patterning biomolecules on SiP surfaces, where the patterned sensing structures, like MRRs, typically have dimensions on the order of 10 µm. Some other advantages of µCP include its procedural simplicity, low cost, and good reproducibility [77,321,328,329]. PDMS stamps are robust and can be reused many times without significant loss of performance, but they are also sufficiently inexpensive and easy to fabricate that they can be treated as disposable when sample contamination is a concern [77,321]. Compared to printing techniques that address one spot on a substrate surface at a time, µCP is high-throughput, as a complex 2D pattern can be printed with only a single inking and application step [329].

While µCP is suitable for efficiently creating complex 2D patterns of a single bioreceptor, it is poorly suited to creating multiplexed arrays with many different bioreceptors [32]. Multiple cycles of inking and printing and careful stamp alignment would be required to print multiple bioreceptors, making this a time-consuming and cumbersome process. Another challenge is that bioreceptor immobilization strategies often include surface modifications, like silanization, which increase surface hydrophobicity prior to bioreceptor attachment [258]. This can reduce the differential wettability between the stamp and substrate, which may, in turn, reduce the efficiency of bioreceptor transfer to the substrate. Materials like PDMS can also transfer unwanted materials like residual uncured oligomers to the regions of the chip that they contact during stamping, potentially contaminating the surface and complicating bioreceptor patterning and subsequent assay steps [330,331,332]. Other limitations of µCP include a potential reduction in bioreceptor binding activity due to drying [322,329], patterning accuracy issues due to PDMS deformation under loads and swelling in the presence of some solvents [328], the requirement for cleanroom facility access to fabricate stamp master molds [77,329], and potential damage to the fragile sensor surface resulting from direct contact with the stamp.

To date, µCP has not been widely used to pattern SiP biosensors, though Peserico et al. [202] used a “tip-mold microcontact printing” technique to functionalize silicon nitride MRRs with ssDNA probes in a spatially defined manner (Table 37, Figure 17b,c). Instead of using a traditional stamp, a PDMS µCP probe was prepared by casting a thin layer of PDMS over the tip of a 125 µm-diameter optical fiber. The probe tip was treated with hydrochloric acid and hydrogen peroxide to enhance its hydrophilicity, then inked in a solution of amine-modified ssDNA. Using a micrometric positioner, the inked probe tip was contacted with the MRR of interest for 45 min in a humidified environment. This allowed sufficient time for the probes to covalently link to the amine-reactive sensor surface, which had previously been modified with APTES and a succinic anhydride/EDC linker. The authors reported that the printed ssDNA probes retained good hybridization efficiency toward their targets. Overall, a resolution of 100 µm was reported for this µCP method, which was suitable for the 200 µm-diameter MRRs used. While this variant of µCP could be used for multiplexed functionalization if parallelized with multiple tip-mold probes or multiple cycles of inking and printing, such a process would be cumbersome, time-consuming, and generally unsuitable for high-throughput biosensor preparation. 

### 4.2. Pin and Pipette Spotting

Nano- and micropipettes filled with a bioreceptor solution can be used in contact mode to deposit small drops of reagent on a substrate by capillarity [33]. Manual spotting of bioreceptor solutions with a micropipette, potentially accompanied by a microscope or stereoscope for improved positional accuracy, is a simple and low-cost technique for spatially controlling the deposition of different bioreceptor solutions on specified regions of a SiP chip in the research setting. However, this low-resolution technique has limited reproducibility, accuracy, and throughput. This technique could be adapted to a high throughput multiplexed dispense format using a pipetting robot [333]. Commercially available pipetting robots, however, typically have minimum dispense volumes of 200–500 nL [334,335,336], which is approximately three orders of magnitude greater than the dispense volumes achievable with pin- and inkjet-based dispensing. Thus, this strategy would still be limited by poor resolution.

Pin-based spotting or pin printing is a similar technique whereby a robotically controlled pin is loaded with the printing solution, then tapped on the sensor surface to deposit picoliter- to nanoliter-scale droplets (Figure 18) [77,329]. Pin printing has been widely used for the preparation of DNA microarrays, and commercial arrayed pin printers are available for this purpose [33]. This technique offers low sample consumption and good resolution, with minimum spot sizes in the range of 1–100 µm, depending on the pin geometry [33,77,313].

Variations of pin printing include contact printing with solid, split and quill pins (Figure 18c–e) [77,329,337]. Solid pins are usually fabricated from micromachined stainless steel, tungsten, or titanium, and have convex, flat, or concave tips. They are loaded by dipping the pin tip in a reservoir filled with the bioreceptor solution and must be reloaded every few spots [77]. Commercially available solid microarraying pins available from Arrayit Corporation can print spots down to ~90–100 µm in diameter [315]. Solid pins are suitable for printing viscous liquids. This is valuable for protein solutions which are often prepared with viscous additives like glycerol, concentrated sugars, or high molecular weight polymers [77,329]. However, the requirement for frequent pin reloading makes solid pin printing very time-consuming. This limitation is addressed by split and quill pin designs, which permit serial printing of many spots from a single load. Split pins are fabricated with a 10–100 µm-diameter microchannel that is filled by capillary action during sample loading [77]. During printing, the split pin must be impacted on the substrate to overcome surface tension and eject picoliter- to nanoliter-scale droplets [77,337]. Quill pins have a similar design to split pins, but with a larger fluid reservoir [338]. Consequently, they can print hundreds of spots from a single load. Unlike split pins, quill pins only require a small tapping force to eject sample droplets onto the substrate [329,337]. Commercially available split and quill microarraying pins can achieve spot volumes down to ~350 pL and spot sizes down to ~37.5 µm in diameter [339,340,341]. Split and quill pins are best suited to low-viscosity solutions because they are susceptible to clogging with viscous liquids, which hinders spot reproducibility [77,329].

Split and quill pins can be micromachined from metal, but they have also been fabricated from silicon using standard microfabrication techniques that offer lower cost and smaller pin dimensions for improved resolution [33,77]. The BioForce Nano eNabler (Bioforce Nanosciences, Virginia Beach, VA, USA) is a commercial automated pin-based printer which uses a microfabricated silicon cantilever, called a Surface Patterning Tool (SPT), to deposit 1–60 µm droplets with 20 nm positional accuracy in the x, y, and z directions [313,314]. The SPT cantilever includes an integrated microfluidic network consisting of a reservoir to hold 0.5 µL of sample and a microchannel through which the sample flows to the tip via capillary action [313]. The droplet size is controlled by the contact time and contact force of the cantilever tip with the surface [313].

Pin-based functionalization of biosensors can be multiplexed by replacing or washing the printing needle when switching solutions [77]. In general, solid pins are easier to clean than split or quill pins, which usually require ultrasonication (for metal pins) or heating with a propane torch (for silicon pins) to thoroughly remove contamination [337]. Regarding its suitability for patterning SiP sensors, pin printing is inherently a contact technique that may damage fragile SiP structures [329]. A major challenge associated with pin printing is that optimizing spot size and reproducibility is a highly multifactorial problem [338]. Namely, the printing performance is highly dependent on the fluid properties, surface wettability, pin geometry, surface contact force, robotic controls, and environmental conditions [338]. Temperature and humidity control are typically required to slow evaporation of the sample, lower the risk of pin clogging, facilitate covalent bioreceptor immobilization on the surface, and preserve bioreceptor activity [329]. Further, spot reproducibility may deteriorate over time as a pin deforms from repeated contact with the substrate or as a split or quill pin’s reservoir is depleted [77]. All of these considerations must be accounted for when designing a protocol for reliable SiP biosensor functionalization.

In the literature pipette and pin spotting have been widely used to pattern bioreceptors on SiP biosensors (Table 38). Several works have used spotting with a micropipette to pattern SiP sensors with 0.1–10 µL-scale droplets of antibodies [17,166,168,170], ssDNA probes [109,194,195,196,198,303], and lipid nanodiscs [148]. These strategies have been used to create 2- [198] to 9-plex [148] multiplexed biosensors. Several other works have employed the BioForce Nano eNabler to pattern SiP sensors with bovine serum albumin [313], ssDNA [163], glycans [132,133], and lipid nanodiscs [149]. These works have reported the successful preparation of 2- [133] to 8-plex [163] biosensors. Angelopoulou et al. [238] spot printed antibodies and peptides on a silicon nitride MZI sensor chip with a contact printing arrayer using solid (375 μm tip, 12 nL per spot) and quill (62.5 μm tip, 0.5 nL per spot) pins. The spotting design required multiple overlapping spots to coat the waveguides with the solid pins taking 7 times as long to print despite depositing more liquid per spot compared to the quill pins. The authors found no significant difference in the sensor response between the solid versus quill pin tips.

### 4.3. Microfluidic Patterning in Channels 

Microfluidic patterning in channels is a soft lithography technique whereby a gasket fabricated with microchannels, also called a microfluidic network (µFN), is reversibly bonded to a solid substrate and the bioreceptor solution is drawn through the microchannels (Figure 19) [33,77,321,342]. Bioreceptors are, therefore, patterned on the substrate according to the channel geometry. The µFN is usually made of molded PDMS, though laser-cut Mylar gaskets have also been used [161]. Biopatterning with µFNs was first demonstrated in 1997 by Delamarche et al. [342] for the deposition of biomolecules on solid substrates. In this work, immunoglobulins were patterned on gold, glass, and polystyrene with submicron resolution using PDMS µFNs. The channels were rendered hydrophilic with oxygen plasma and filled by capillarity to deposit the biomolecules.

µFNs using capillary flow, like those used by Delamarche et al. [342], can achieve micron-scale pattern resolution, as the microfluidic channels can be prepared with micron-scale cross section dimensions [316]. Microfluidic patterning in µFNs can also be performed using pressure-driven flow, but this requires larger channels with cross section dimensions on the order of 10 µm due to high hydraulic resistance [316,317]. Indeed, this yields poorer pattern resolution than capillary flow. However, pressure-driven flow permits the easy exchange of patterning fluids. For example, sequential surface modification steps, including crosslinker attachment, bioreceptor immobilization, rinsing, and post-processing with blocking molecules to prevent non-specific binding, can all be performed in the µFN without surface drying or removing the flow cell [32,329]. Another valuable feature of this technique is that sensing elements designed to operate in liquid media can be probed throughout the patterning process for real-time biofunctionalization monitoring [313]. Beyond biopatterning, µFNs are often used to facilitate miniaturized, simultaneous, and highly localized multi-step binding assays on functionalized sensors [321,342].

Multiplexing is typically achieved using µFNs with multiple parallel channels. Different bioreceptor solutions can be simultaneously drawn through the individually addressable channels, creating a one-dimensional array. However, this method is not well-suited to creating discrete two-dimensional patterns of bioreceptors, which would require complicated multilayer fluidics with three-dimensional flow paths [32,316]. Further, any change to the SiP sensor layout would require a redesign of the µFN [83]. Therefore, microfluidic patterning in channels has less multiplexing flexibility than pin printing, inkjet printing, and microfluidic probe-based patterning, which can localize chemical processes to arbitrary locations on a substrate [32]. Another limitation of µFNs is that reagent consumption can be high, depending on the microchannel volume, and bioreceptor molecules can be lost to microchannel walls due to nonspecific adsorption [83,316]. This is particularly undesirable when using costly bioreceptors. Similarly to µCP, materials like PDMS, which are used to fabricate the µFN, can leach uncured oligomers, which may contaminate the sensor surface and complicate functionalization and subsequent assay steps [330,331,332]. However, a major advantage of this technique compared to pin and inkjet printing is that bioreceptors are maintained in a controlled liquid environment throughout the patterning process. This ensures good uniformity of the biofunctionalized regions and prevents activity loss of environmentally sensitive bioreceptors due to drying [329,342]. Other advantages of this technique are that it is low cost, exceptionally simple, and unlikely to damage SiP sensing elements.

In the literature, µFNs are a popular choice for patterning SiP devices with bioreceptors (Table 39). µFNs have been used to pattern SiP sensors with antibodies [18,22,161,162,169,174,295], aptamers [174], ssDNA [174], lipid nanodiscs [150], and BSA [174,313]. This technique has been used to confine bioreceptors to select sensing structures on a single SiP device, while leaving other structures bare to control for nonspecific binding, temperature, and instrument drift [18,161]. It has also been used to compare different bioreceptor immobilization strategies. For example, Byeon et al. [295] used a 2-channel microfluidic gasket to compare bioreceptor immobilization in the presence and absence of a chemical catalyst, while González-Guerrero et al. [313] used two microfluidic channels to compare covalent and adsorption-based bioreceptor immobilization on a single sensor. Finally, µFNs have been used to create multiplexed MRR sensors with different microrings functionalized with different bioreceptors [22,150,162,169,174].

### 4.4. Inkjet Printing

In contrast to the contact-based deposition systems discussed in Section 4.1, Section 4.2 and Section 4.3, which can expose the silicon waveguides and other structures to damage, non-contact inkjet systems use piezoelectric actuation for deposition without touching the sensor surface. Non-contact inkjet based printer systems were developed in the late 1990’s where off-the-shelf desktop inkjet printers were repurposed to dispense controllable volumes of reagents in the ~80 pL range [343]. Initial development using home-built ink-jetting exposed the inkjet solution to heat resulting in a loss of functionality by denaturation or decomposition of biomolecules.

Piezoelectrically actuated non-contact inkjet devices have come to the forefront for localized reagent deposition by leveraging the control provided by a piezoelectrically actuated glass capillary that is capable of depositing droplets that are on the order of one pL to a few hundred pL in size [246,318], as illustrated in Figure 20a. These systems have x-y spatial accuracies ~15–20 μm while dispensing highly accurate volumes of assay reagents without any heat source affecting the samples [344,345]. The capillary tips hover above the etched resonator area while a voltage source piezoelectrically compresses a collar surrounding the nozzle to create pressure waves within the fluid that result in expulsion of <1 nL droplets onto the sensor surface.

While the high spatial and volumetric controllability of piezoelectrically actuated inkjet systems is desirable, the disadvantages must be mitigated which vary for each assay solution. Nonspecific adsorption of proteins onto the borosilicate glass capillary can cause protein loss when depositing <1 nL of low concentration (<20 μg/mL) protein solutions. Delehanty and Lingler found that both the ionic strength of the printing buffer and presence of a carrier protein greatly affected the amount of biotinylated Cy5-labeled IgG that adsorbed to the capillary surface, thus influencing the amount of IgG that was dispensed from the capillary [346]. The authors’ results showed an inverse relationship between the ionic strength of the buffer (PBS) and amount of IgG protein dispensed from the capillary, which was attributed to the nonspecific adsorption of proteins to borosilicate glass. Moreover, they found that with the addition of a carrier protein (BSA), the ionic strength effect could be completely mitigated while increasing the total concentration of IgG that was dispensed up to 44-fold.

On-board cameras and positioning software allow for spot printing to be carried out in a systematic fashion by fiducial mark recognition whereby ~300 pL drops of SS-A antigen at 200 μg/ml can be spot printed using a sciFLEXARRAYER S5 (Scienion, AG, Berlin, Germany) on 128 rings with PDC-70 nozzle [293]. Kirk et al. [246] illustrated the high throughput capabilities and low assay reagent consumption by printing 10 array chips with 6 microrings per chip in 9 s, consuming a total < 25 nL of reagent. In addition to functionalizing SiP chips with multiple bioreceptors for multiplexed analyte detection, inkjet printing can be leveraged to include reference sensors. As previously discussed, the functionality of a microring resonator is based on its resonance wavelength shift and is often measured with respect to a nearby reference resonator. Positioning the reference microring resonator nearby the sensing resonator helps to eliminate shift stemming from thermal gradients across the chip. Other sources contribute to anomalous background wavelength shift, such as non-specific binding, which may be useful to control for using reference rings. Therefore, while some reference rings may remain buried under an oxide cladding, it may be beneficial in some applications to also include protein coated reference resonators. Cognetti and Miller [167] fabricated a ring resonator set as shown in Figure 20b. One ring was functionalized with anti-SARS-CoV-2 RBD + SARS-CoV-2 RBD (a-S1 + S2) and another was functionalized with 0.1% BSA as a control for non-specific binding, illustrated by the blue and red dots, respectively. The piezoelectric inkjet process allowed for controlled deposition of the assay reagents as isolated elements and showed the relative (BSA ring-subtracted) wavelength shift in response to the SARS-CoV-2 spike protein [167].

Other types of detection mechanisms have been demonstrated through inkjet printing of assay reagents. For instance, Laplatine et al. [319] used a Scienion sciFELXARRAYER S12 to deposit an array of 64 different peptides in buffer on MZIs (spot size of ~150 μm). The MZI array was used to measure volatile organic chemicals (VOCs) with limits in the ppm range as the basis for a silicon olfactory sensor. Ness et al. [318] used a FUJIFILM Dimatix DMP-2831 materials piezoelectric inkjet printer (FUJIFILM Dimatix, Inc., Santa Clara, CA, USA) with 1 pL dispensing DMC-11601 cartridges to deposit ~30 μm diameter spots by optimizing the functional fluid to have a higher viscosity and lower surface tension which was achieved by the addition of glycerol and a surfactant, respectively. A Dimatix materials printer was also used to deposit a functional biotin-modified polymer and porous hydrogel on MZIs, whereby the functional polymer was able to sense the specific binding of protein streptavidin and the benzophenone dextran (benzo-dextran) porous hydrogel was shown to hinder the non-specific binding of BSA on the sensor surface [347]. Table 40 summarizes several demonstrations of inkjet-based bioreceptor deposition on SiP sensors.

### 4.5. Microfluidic Probes

Microfluidic probes (µFPs) (Figure 21), which were first demonstrated in 2005 by Juncker et al. [348], combine the features of microfluidics and scanning probes to deliver biomolecules to surfaces. µFPs are classified as “open space microfluidics”, as they confine nanoliter volumes of processing liquids on substrates without solid-walled microchannels [83,317]. This is achieved through hydrodynamic flow confinement (HFC) of the processing liquid, which is made possible by the microscale dimensions of the system and the resulting laminar flow regime [32,348].

The tip of the µFP may be fabricated from silicon [348], silicon and glass [317], or PDMS [349]. It consists of coplanar injection and aspiration microapertures and is placed 10–200 µm above the substrate [32]. An immersion liquid fills the gap between the probe tip and substrate, while processing liquid is injected from the injection aperture and collected by the aspiration aperture. The processing liquid is confined above and below by the probe tip and substrate, while it is confined laterally by hydrodynamic boundaries formed by the immersion liquid [32]. In a simple µFP configuration, the flow rate of the injected fluid, Q_I_, must be lower than the flow rate of aspirated fluid, Q_A_, to maintain flow confinement [348]. The ratio Q_A_/Q_I_ can be varied, along with the distance between the probe tip and surface, to tune the shape and size of the region where the processing fluid contacts the surface [348]. Typically, this impingement area has a teardrop shape, but an alternative radial probe tip design can be used to create a circular impingement area [32]. The µFP is mobile and can scan over a substrate to create complex patterns; depending on the direction and speed of travel relative to the microfluidic flow, continuous shapes or discrete spots can be patterned [349]. Spot sizes as small as 10 × 10 µm^2^ are possible [83].

µFP-based bioreceptor patterning has not yet been demonstrated on SiP biosensors, but it may be a promising technique for future application. Firstly, µFPs are suitable for multiplexed patterning, as processing fluids can be rapidly switched using an external valve system [32]. Further, the probe can follow an arbitrary scan path, allowing for flexible and customized patterning of sensors with non-standard layouts [348]. Given that this is a non-contact technique, it is unlikely to damage fragile SiP surfaces. Unlike inkjet and pin printing, µFPs pattern surfaces in a liquid environment, which prevents uncontrolled wetting and drying effects, thus improving spot uniformity and homogeneity, while preventing aggregation or denaturation of printed biomolecules [317,348]. For example, Autebert et al. [83] demonstrated less than 6% variation in spot homogeneity for an array of 170 spots of IgG printed on polystyrene. While a simple µFP configuration typically requires large volumes of processing fluids, a 10-fold decrease in µFP reagent consumption has been achieved using hierarchical flow confinement and recirculation, making it comparable to pin and inkjet printing (e.g., 1.6 µL to print 170 spots of IgG, each with a 50 × 100 µm^2^ footprint).

The main challenges of this patterning technique are its low throughput and limited commercial availability [32]. Using a simple µFP configuration, only one spot can be addressed at a time, each requiring a residence time defined by the kinetics of the bioreceptor’s immobilization reaction. Multiple spots could be patterned simultaneously using probe tips with microfluidic channel bifurcations to increase throughput, but this would only be suitable for SiP sensors with highly standardized layouts, as aperture spacing would need to match the spacing of SiP sensing structures [83]. The accessibility of this technique is limited, as commercial µFP-based patterning systems are not yet available. Another potential challenge is that, when applied to SiP sensor surfaces, this technique may suffer from perturbations in hydrodynamic flow confinement due to the three-dimensional topography introduced by the patterned silicon structures [317]. This, in turn, may result in reduced spot homogeneity.

### 4.6. Summary and Future Directions

Here, we have discussed several strategies for preparing patterns of bioreceptors on SiP sensor surfaces for multiplexed detection. In general, non-contact patterning techniques are attractive for SiP sensor biofunctionalization, as they prevent damage to the sensor surface and integrated optical and electronic components. Of the strategies discussed here, inkjet printing is a promising strategy for biopatterning multiplexed SiP sensors. Inkjet printing is a flexible, high throughput, low-waste, and multiplexable non-contact patterning strategy that can achieve sufficient resolution for the functionalization of most SiP devices [329]. However, printing protocols (e.g., actuation waveform design, environmental controls, additives to bioreceptor “ink”, etc.) must be optimized for replicable deposition of uniform spots. Future studies using inkjet-based biopatterning of SiP sensors should quantify and optimize inter- and intra-spot uniformity, along with inter-spot and run-to-run replicability to validate reliable performance of this patterning technique. µFP is another flexible non-contact patterning technique, which can achieve improved spot uniformity and replicability compared to other printing methods, and may be a promising option for SiP sensors [317,348]. Nevertheless, this technique must still be validated for bioreceptor patterning on SiP surfaces.

## 5. Critical Comparative Analysis of Solutions and Discussion of the Interplay between the Three Aspects of Biofunctionalization

This review has provided a detailed overview of strategies that have been or can be used to functionalize SiP biosensors in terms of bioreceptor selection, immobilization chemistry, and patterning strategy. We have benchmarked potential strategies for each of these three aspects of biofunctionalization against a set of performance criteria relevant to SiP sensing. In addition to assessing the tradeoffs of individual solutions in the context of the anticipated biosensor use case, the compatibilities and incompatibilities between solutions to each of the three aspects of biofunctionalization are an essential consideration. Moreover, the interplay between bioreceptors, immobilization chemistries, and patterning techniques can affect what is considered suitable performance for a given biofunctionalization need. For example, when using a patterning technique with very low reagent consumption, bioreceptors with a greater cost per milligram may still permit very low reagent cost per sensor. This underscores the importance of considering these three aspects of biofunctionalization in concert.

The first step in designing a biofunctionalization protocol once the application of the biosensor is defined and the target(s) known, is bioreceptor selection. As discussed in Section 2, different bioreceptors are suitable for different targets. For many targets (proteins, small molecules, viruses, bacteria, etc.) antibodies, aptamers, MIPs and PCCs may be suitable. Despite being very cost-effective and stable, currently available MIPs cannot achieve sufficient binding affinity and/or selectivity to achieve detection at clinically relevant levels for many targets. Of the other three options, antibodies are the most readily available and well-characterized, but their poor stability and high cost limit their suitability for POC use. Moreover, in our group’s experience, batch-to-batch variability has been a notable roadblock in the design of replicable biosensing assays using antibodies.

Synthetic antibody analogs like aptamers and PCCs, which can achieve similar affinity and specificity to antibodies, are appealing and versatile options for POC sensors. Currently, a significant roadblock in the widespread adoption of aptamers and PCCs for biosensing applications is the relatively limited availability of pre-designed products for ready use against a diverse range of targets, though this challenge can be mitigated in coming years with further research and development [30,89,234]. Additionally, aptamers often require careful sample preparation (buffering, filtering, or tight temperature control) to avoid their folding or denaturing prematurely during use. Robust aptamer formulations need to be screened with these factors in mind, given each sensing device’s use case. Regardless, their low cost, good stability, and highly reproducible and scalable production are important advantages of aptamers and PCCs for POC biosensing.

For nucleic acid targets, nucleic acid probes (hybridization-based detection), HCCD, and CRISPR-dCas9-mediated detection may be suitable bioreceptor options. In applications requiring highly multiplexed nucleic acid sensing, simple hybridization-based detection with nucleic acid probes offers the greatest flexibility and assay simplicity. When exceptionally high sensitivity and selectivity are required for very low-concentration targets, HCCD or CRISPR-dCas9-mediated detection may be preferable. It should be recognized; however, these are very early- stage approaches with limited precedent for use on SiP platforms and have yet to be validated for sensing in complex samples.

Lastly, glycans, lectins, and lipid nanodiscs are valuable for the study of carbohydrate-protein and cell membrane interactions, respectively, but their often-poor affinity and selectivity limit their applications beyond such studies.

In addition to these considerations, the immobilization chemistries that are compatible with each type of bioreceptor should be kept in mind during bioreceptor selection, with particular attention paid to compatibility of the immobilization chemistries with other steps of biosensor fabrication and integration with sample fluid delivery. Broadly, passive adsorption of bioreceptors leads to poor stability of the functionalized surface and diminishes the bioreceptor’s binding activity. One exception is lipid nanodiscs, which adsorb well to silicon dioxide surfaces to yield reproducible, regenerable, and stable functional layers [148,149,275,276]. For other bioreceptors, passive adsorption is not recommended, aside from in preliminary sensor validation experiments where simplicity and rapid assay design are priorities. Nevertheless, novel polymeric coating materials (e.g., PAcrAm™ and AziGrip4™ from SuSoS AG) may permit stabler and more oriented bioreceptor immobilization with similar simplicity to passive adsorption techniques, potentially comprising a valuable future research direction [309,310,311].

Among the various covalent and bioaffinity-based immobilization strategies explored in this review, different immobilization methods can produce very different results depending on the bioreceptor. For example, many covalent methods can readily achieve predictable and oriented aptamer and nucleic acid probe immobilization by targeting terminal functional groups incorporated into these bioreceptors during synthesis; this ensures good binding site availability for target capture. Conversely, when used for antibody immobilization, these covalent strategies typically target native functional groups that are abundant on the antibody surface, leading to random antibody orientation and reduced target-binding capacity.

When antibody binding capacity must be optimized, bioaffinity-based strategies using antibody-binding proteins, like Protein A, may be a preferable choice, though these strategies involve tradeoffs in terms of stability, regenerability, and cost. It should also be noted that Protein A-based antibody immobilization may compromise the specificity of immunoassays using amplification with a secondary antibody [350]. In our experience, using Protein A in sandwich immunoassays was correlated with a considerable non-specific signal during the secondary antibody amplification step, which was not observed in immunoassays prepared using simple passive adsorption of the detection antibody on the SiP sensor surface [351]. This non-specific signal may be related to the unwanted capture of secondary antibodies by unoccupied Fc-binding sites on the Protein A-coated sensor surface (potentially due to incomplete functionalization with capture antibody or due to unbinding of capture antibody during the course of the experiment) [350]. One solution may be to choose secondary antibodies that do not bind well to Protein A, though this may be challenging, as Protein A and Protein G bind well with antibodies from many common host species (cow, goat, mouse, rabbit, and sheep) that are used in immunoassays [350]. Protein L, which does not bind with cow, goat and sheep antibodies and binds weakly to rabbit antibodies, may offer greater flexibility in the choice of secondary antibody, potentially making it a preferable antibody-binding protein for sandwich assays [350]. Silane-based covalent strategies have also been successfully used to immobilize capture antibodies on SiP sensors for assays using amplification with a secondary antibody [17,18,162,163]. This highlights that the assay format (label-free/labeled) and its synergy with the biofunctionalization strategy should be carefully considered.

Next, the selection of a patterning technique should take into consideration factors such as the bioreceptor cost, fluid properties of the bioreceptor solution, and changes in sensor surface hydrophilicity caused by the immobilization chemistry. For instance, patterning in microfluidic channels is a simple and popular choice in SiP biosensor functionalization protocols, but it typically has high reagent consumption. As an example, in our group’s previous work, we deposited 20 µg/mL solutions of capture antibodies on SiP sensors via microfluidic channels using pressure-driven flow at 30 µL/min for 45 min [351]. This consumed a total of 27 µg of antibody, which costs roughly CAD $135, assuming an antibody cost of ~CAD 500/100 µg. Further, bioreceptors may be lost to adsorption on the channel walls during patterning, and this inefficient reagent use is particularly undesirable for costly bioreceptors, such as antibodies. In assays using in-flow patterning followed by sample introduction using the same µFN, targets in the sample may bind to bioreceptors coating the channel walls and non-sensing regions of the SiP chip. This can deplete target molecules from the sample more rapidly than if the sensing regions, alone, were functionalized. Consequently, this may worsen the limit of detection [352]. Offline patterning of bioreceptors to ensure that they are only localized to the sensing regions is, therefore, particularly beneficial for detecting precious targets at very low concentrations.

The fluid properties of the bioreceptor solution can also dictate the success of a patterning technique. In particular, µCP, pin printing, and inkjet printing strategies are sensitive to the viscosity and surface tension of the bioreceptor solution. Required additions to bioreceptor solutions, such as glycerol to slow evaporation, must be accounted for when optimizing the patterning protocol. Surface modifications used for different bioreceptor immobilization chemistries affect the hydrophilicity of the sensor surface, which, in turn, influences the efficacy and resolution of the patterning strategy [338]. For example, silanization decreases the hydrophilicity of the SiP sensor surface [267]. In the context of µCP, this may inhibit the transfer of bioreceptor “ink” from the stamp to the sensor surface. On the other hand, this decreased surface hydrophilicity will decrease the spreading of droplets of aqueous bioreceptor solutions. This may improve the resolution of patterning techniques such as pin and inkjet printing.

While not a focus of this review, antifouling strategies must typically be integrated with biofunctionalization protocols in order to prevent non-specific adsorption of sample matrix components to the sensor surface [29]. Antifouling strategies can be included in covalent bioreceptor immobilization protocols through the use of linkers that include polyethylene glycol (PEG) chains (e.g., SM(PEG)12 [133], BS(PEG)9 [132]), which increase the hydrophilicity of the surface coating to reduce non-specific protein adsorption [29]. Other approaches include coating the surface via passive adsorption with bovine serum albumin [264,353] or commercial blockers, such as StartingBlock [109,163,196], BlockAid, and StabilCoat [167], after bioreceptor immobilization. It is important to consider how to best fit antifouling strategies into biofunctionalization workflows. For further details about antifouling strategies for SiP biosensors, readers are directed to ref. [29].

Lastly, the entire biofunctionalization procedure must be considered in the context of the overall sensor system design. While some functionalization strategies may be suitable for the SiP sensor chip itself, they may not be suitable for systems including chip-mounted electronic/photonic inputs and outputs, which can be used to translate this technology to a commercial POC platform (Figure 22) [12,13,66,354]. For example, immobilization chemistries requiring solution-phase reactions may be unsuitable for sensor designs including photonic wire bonds that connect optical inputs and outputs to the on-chip waveguides. Solvents or other chemicals used in functionalization may damage or swell the photonic wire bond or low-index photonic wire bond cladding materials, resulting in damage to the fine optical connection [355,356]. In this case, immobilization chemistries employing vapor-phase surface modifications or direct crosslinking of bioreceptors (e.g., UV crosslinking of nucleic acids or aptamers) to the unmodified surface may be preferable. Similarly, plasma or UV/ozone treatment are likely more suitable surface pre-treatment techniques than immersion in piranha solution for integrated SiP systems. For these systems, the surface topography and locations of chip-mounted components should inform the selection and design of the patterning strategy. In general, non-contact patterning techniques (e.g., inkjet printing) can permit flexible bioreceptor pattern design, while preventing damage to the system, making them preferable to techniques that require contact between the patterning tool and surface.

In summary, it is important to consider the interplay between the three constituents of biofunctionalization as well as the silicon photonic device, fluidics, and detection assay when designing a biofunctionalization strategy. The examples above highlight the importance of considering and addressing the relationships between different bioreceptors, immobilization strategies, and patterning techniques and their suitability for different assay formats and integrated sensing architectures. This discussion aims to bring attention to the importance of considering and addressing these relationships in order to design successful biofunctionalization protocols for SiP biosensors.

## 6. Conclusions

When combined with carefully designed biofunctionalization strategies, SiP sensors have the potential to permit accurate and information-rich decentralized diagnostic testing for a diverse range of clinical applications. We have identified and evaluated different strategies for SiP sensor biofunctionalization in terms of bioreceptor selection, immobilization strategy, and patterning technique. Different solutions for each aspect of biofunctionalization have been benchmarked against a set of critical performance criteria relevant to multiplexed SiP biosensing and examples from the literature have been discussed and categorized. In addition to providing critical discussion about solutions for each aspect of biofunctionalization, we have also identified the interplay between these three aspects to help inform the design of SiP functionalization protocols and have highlighted additional functionalization process constraints relevant to SiP system integration for POC biosensing. 

Broadly, several classes of synthetic bioreceptors (e.g., aptamers, PCCs, nucleic acids) offer excellent potential for multiplexed POC biosensing, as they can achieve high affinity and specificity, and offer scalable and cost-effective production, good stability, and regenerability. However, the availability of ready-to-use reagents remains a roadblock for the use of synthetic antibody analogs. In terms of immobilization strategies, covalent methods offer stable, scalable, and highly tailorable bioreceptor immobilization, but their success often depends highly on the reaction conditions and bioreceptor type, underscoring the potential value of developing standardized and reliable reaction protocols that are optimized for SiP surfaces. Regarding patterning, pin and inkjet-based printing are popular techniques that offer good flexibility and resolution, while inkjet printing has the additional advantages of exceptionally high throughput and being a non-contact method that will not damage the SiP surface or integrated electronic/photonic structures. µFP-based patterning is another attractive potential solution for flexible bioreceptor patterning that may achieve improved spot uniformity, though this technique has yet to be tested on SiP platforms. Overall, this review serves as a detailed overview of the biofunctionalization options available and previously tested on SiP platforms. This can help guide the design of new functionalization protocols, which must also be individually tailored for the specific target analyte(s), assay format, system architecture, and intended operating environment.

## Figures and Tables

**Figure 1 biosensors-13-00053-f001:**
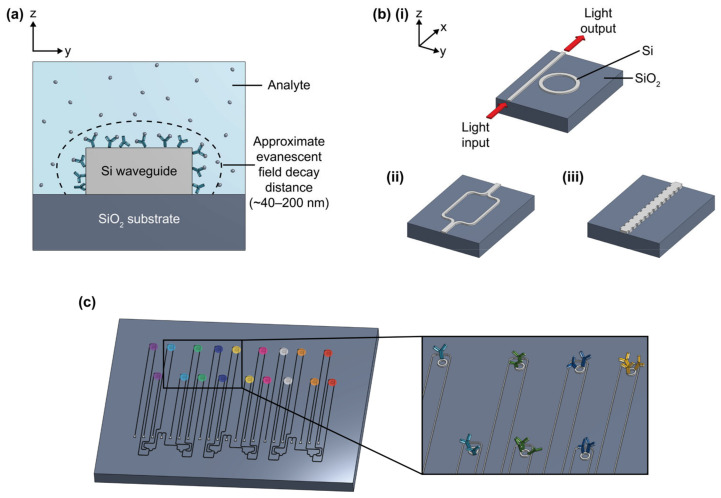
(**a**) Illustration of cross-section of silicon photonic (SiP) sensor, showing the SiO_2_ substrate, Si strip waveguide (height: 220 nm, width: 500 nm), and approximate evanescent field decay distance (~40–200 nm, depending on waveguide geometry and light polarization). (**b**) Illustration of four different SiP sensing architectures, including (i) microring resonator (MRR), (ii) Mach-Zehnder interferometer (MZI), and (iii) Bragg grating sensor. (**c**) Visual depiction of a multiplexed SiP MRR sensor chip, showing different rings functionalized with different antibodies (different antibodies are represented by different colors). Antibodies in (**a**) and (**c**) are not to scale.

**Figure 2 biosensors-13-00053-f002:**
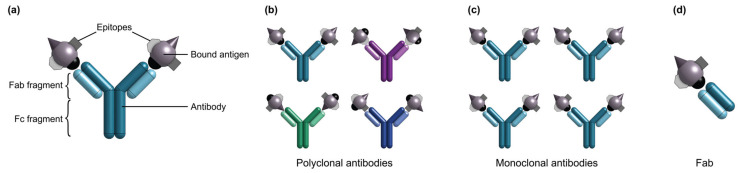
(**a**) Illustration of an antibody and bound antigens. Illustrations of different antibody subtypes, including (**b**) polyclonal antibodies, (**c**) monoclonal antibodies, and (**d**) a Fab fragment. Note that polyclonal antibodies are produced as heterogeneous mixtures in which different antibodies may bind to different epitopes of the same antigen. Monoclonal antibodies are produced as homogeneous samples in which all antibodies bind to the same epitope.

**Figure 3 biosensors-13-00053-f003:**
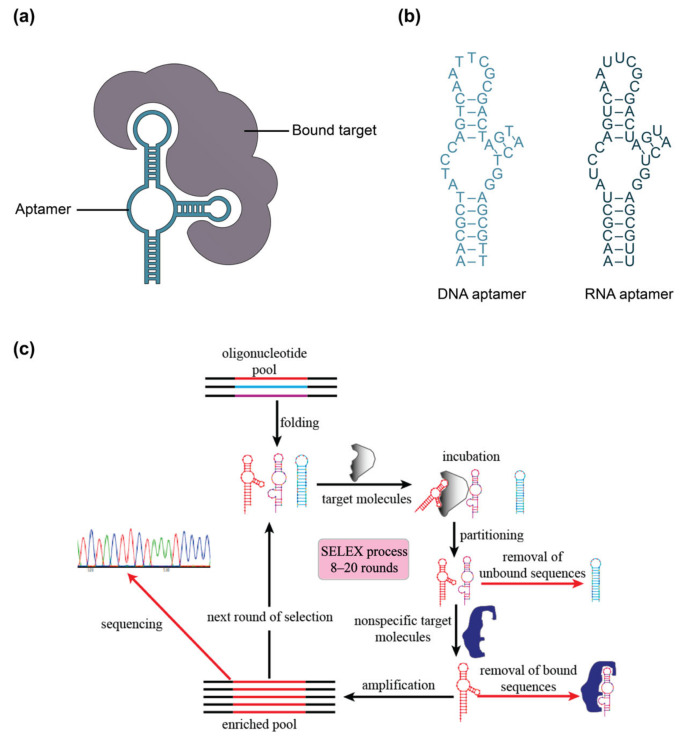
(**a**) Illustration of aptamer and bound target. (**b**) Visual representation of aptamer subtypes: DNA and RNA aptamers. (**c**) Illustration of SELEX (systematic evolution of ligands by exponential enrichment) process to design aptamers against a target. In (**c**), different colors in the oligonucleotide pool represent different nucleic acid sequences, while different colors in the sequencing step represent different nucleic acid bases identified by Sanger sequencing or high-throughput sequencing methods. Part (**c**) is reprinted from Ref. [101] in accordance with the Creative Commons Attribution 4.0 International (CC BY 4.0) license.

**Figure 4 biosensors-13-00053-f004:**
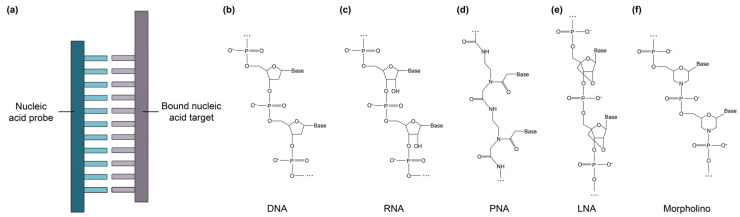
(**a**) Illustration of nucleic acid bioreceptor and bound nucleic acid target. Comparisons of the chemical structures of different nucleic acid subtypes, including (**b**) DNA, (**c**) RNA, (**d**) PNA, (**e**) LNA, and (**f**) morpholino, shown as line structures informed by Refs. [73,191].

**Figure 5 biosensors-13-00053-f005:**
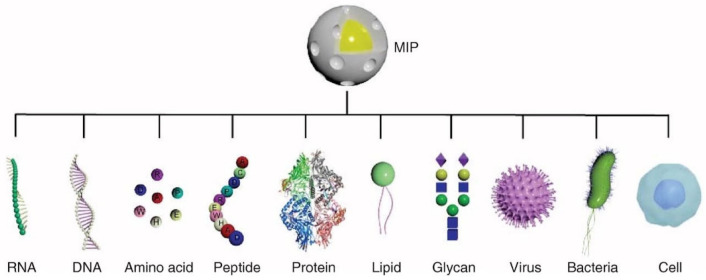
MIPs can be templated with an array of targets including: RNA, DNA, amino acids, peptides, proteins, lipids, glycans, viruses, and bacterial or cell epitopes. Reproduced from Ref. [207] in accordance with the Creative Commons Attribution 4.0 International license (CC BY 4.0).

**Figure 6 biosensors-13-00053-f006:**
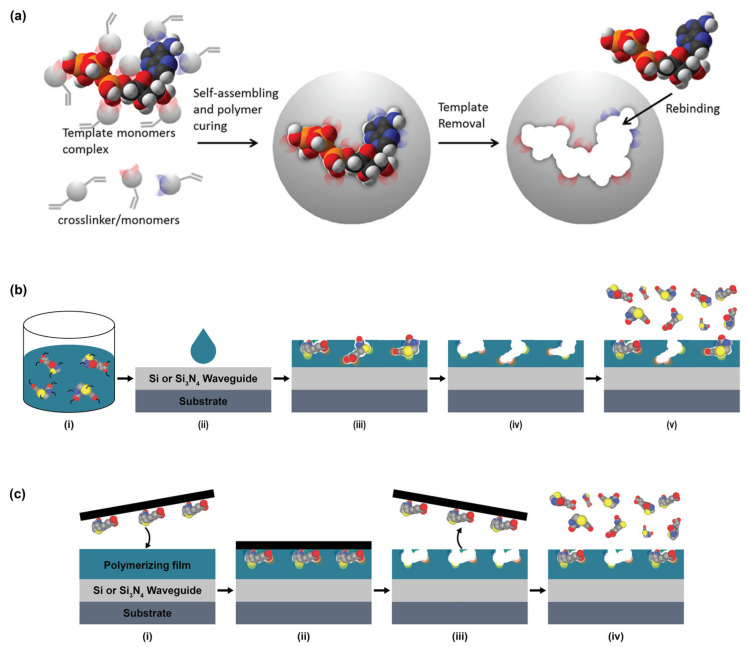
Illustration of MIP templating approaches. (**a**) MIP templating begins with a template mixed with polymer precursors followed by curing and the template removal. (**b**,**c**) Illustration of molecularly imprinted polymer (MIP) showing the random and oriented nature of template orientation on the surface of solution based (**b**) and stamped (**c**) MIPs, respectively. In solution-based MIP preparation (**b**): templates are first solvated in organic solvents with precursors, initiators, and functional monomers (i), followed by deposition on the sensor surface (ii), curing (iii), and template extraction (iv), after which the MIP can be used for target capture (v). Note that the small pieces of color left behind in the binding sites after template extraction, as seen in (iv,v), represent sites where the functional monomers formed non-covalent or covalent bonds with the template. In surface stamping based MIP preparation (**c**): templates are immobilized on a surface mold (i) and pressed into a polymer film on the sensor surface (i,ii) prior to curing. After curing, the surface mold is removed (iii), leaving imprinted binding sites on the sensor surface, which can be subsequently used for target capture (iv). Part (**a**) is reproduced with permission from Ref. [210]. Copyright 2016, American Chemical Society.

**Figure 7 biosensors-13-00053-f007:**
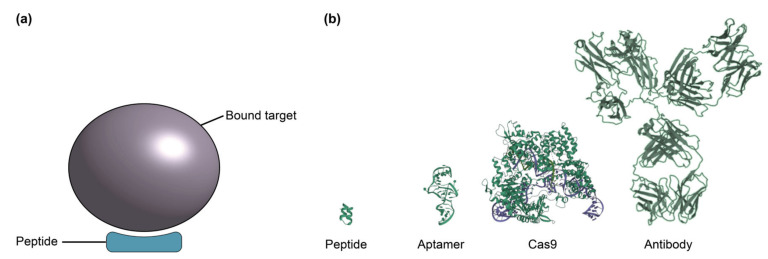
(**a**) Illustration of peptide bound target. (**b**) Comparison of peptide, aptamer, Cas9 enzyme and antibody relative sizes, informed by protein data bank crystal structures 2AU4, 4OO8 and 1IGY [226,227,228]. Peptides are smaller than aptamers, antibodies, and many other bioreceptor classes discussed here, offering potential improvement in SiP biosensor sensitivity by bringing the binding interaction into a region of the evanescent field with higher field intensity.

**Figure 8 biosensors-13-00053-f008:**
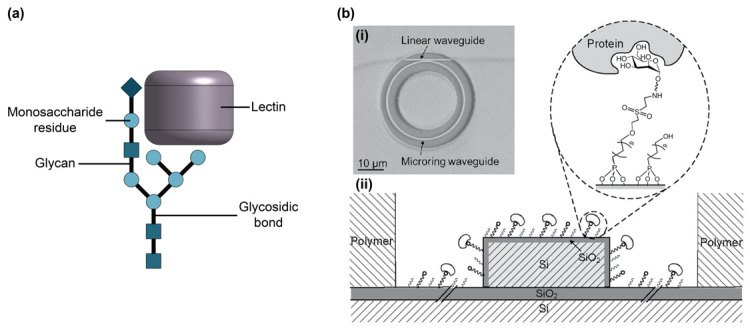
(**a**) Illustration of a glycan and bound lectin. (**b**) (i) SEM image of a microring resonator and (ii) cross-section of microring resonator waveguide using glycans as bioreceptors. The glycans are immobilized using an organophosphonate linking strategy and used for lectin (protein) capture. Part (**b**) is adapted with permission from Ref. [126]. Copyright 2012 American Chemical Society.

**Figure 9 biosensors-13-00053-f009:**
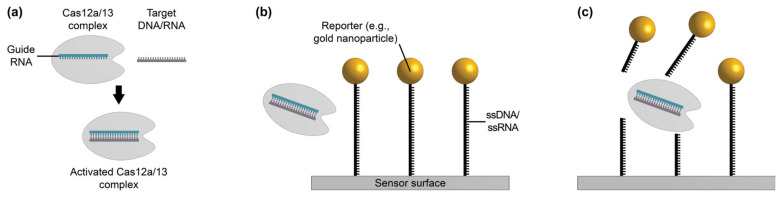
Illustration of HCCD, showing (**a**) activation of the CRISPR-Cas12a/13 effector by the target nucleic acid sample, (**b**) high index contrast reporters (e.g., gold nanoparticles) tethered to the sensor surface by single-stranded DNA or RNA prior to cleavage by the activated CRISPR-Cas12a/13 complex, and (**c**) non-specific collateral cleavage of single-stranded DNA or RNA by the activated CRISPR-Cas12a/13 complex, leading to the removal of reporters from the sensor surface.

**Figure 10 biosensors-13-00053-f010:**
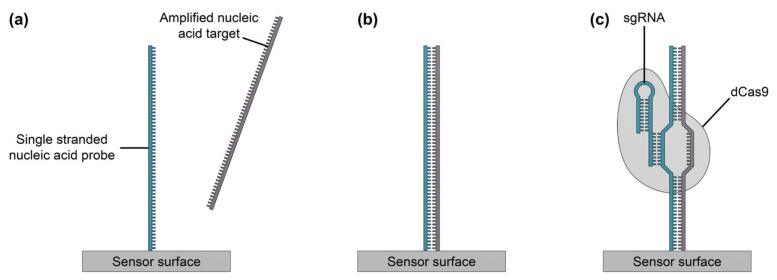
Illustration of CRISPR-dCas9-mediated sensing. (**a**) Single-stranded nucleic acid probes are immobilized on the sensor surface and the nucleic acid targets (amplified by recombinase polymerase amplification) are introduced to the sensor surface. (**b**) The nucleic acid targets hybridize to the surface-bound probes. (**c**) Deactivated Cas9 (dCas9), guided by single guide RNA (sgRNA) specifically binds to the nucleic acid duplex to amplify the signal, without cleaving the nucleic acid duplex.

**Figure 11 biosensors-13-00053-f011:**
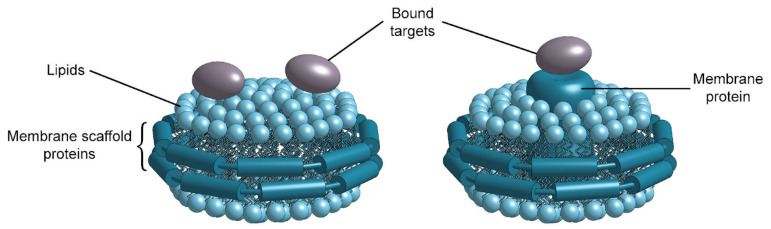
Illustrations of lipid nanodiscs with bound targets. The nanodiscs consist of lipid bilayers, held together by two encircling membrane scaffold proteins. The nanodiscs may be prepared without (**left**) or with (**right**) embedded membrane proteins.

**Figure 12 biosensors-13-00053-f012:**
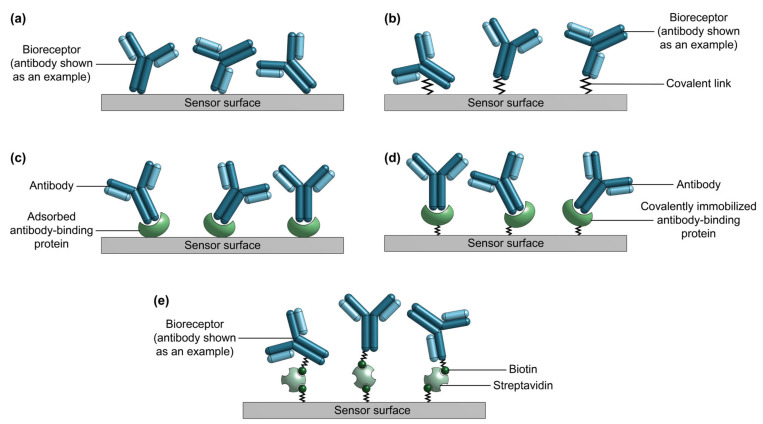
Illustrations of different strategies for immobilizing bioreceptors (antibodies are shown as an example) on SiP sensor surfaces. The depicted immobilization strategies include (**a**) non-covalent passive adsorption, (**b**) covalent attachment, (**c**) bioaffinity-based oriented immobilization using antibody-binding proteins adsorbed to the surface, (**d**) bioaffinity-based oriented immobilization using antibody-binding proteins covalently linked to the surface, and (**e**) bioaffinity-based immobilization in which the surface and bioreceptor are covalently conjugated with biotin and streptavidin is used as a linking molecule.

**Figure 13 biosensors-13-00053-f013:**
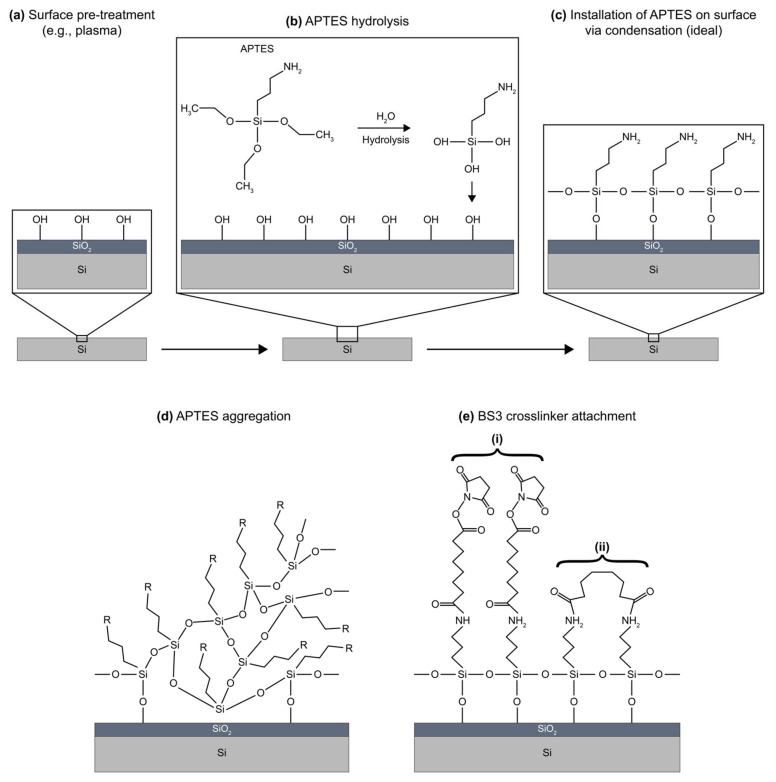
Silanization of SiP surface using 3-aminopropyltriethoxysilane (APTES). (**a**) The native oxide surface of the Si waveguide is pre-treated to remove organic contaminants and activate the surface hydroxyl groups, (**b**) APTES is hydrolyzed to form reactive silanols, and (**c**) adjacent APTES molecules are covalently linked together via silanol condensation and APTES is covalently bound to the surface. This yields a covalently bound APTES monolayer presenting functional amine groups for linker or bioreceptor immobilization. (**d**) Undesirable formation of large silane aggregates on the surface. (**e**) Attachment of a homobifunctional crosslinker to the aminosilane-coated surface, showing (i) ideal homobifunctional crosslinker attachment whereby one reactive group reacts with the silanized surface and the other remains available for conjugation with the bioreceptor, and (ii) undesirable crosslinker-mediated bridging whereby both ends of the homobifunctional crosslinker react with functional groups on the silanized surface, becoming unavailable for bioreceptor immobilization. BS^3^ is used here as an example of a homobifunctional crosslinker.

**Figure 14 biosensors-13-00053-f014:**
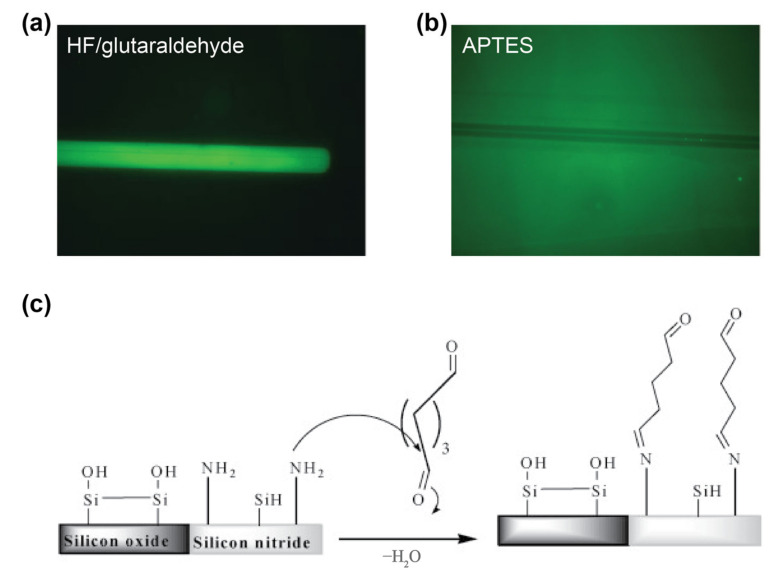
Silicon nitride waveguides from a MZI sensing window with fluorescently tagged (Alexa Fluor 488) antibodies (goat anti-mouse IgG) attached by (**a**) covalent HF/glutaraldehyde-based immobilization and (**b**) APTES functionalization followed by passive adsorption [238]. Parts (**a**,**b**) are adapted with permission from Ref. [238]. Copyright 2022 Elsevier. (**c**) Chemical reaction mechanism for selective silicon nitride functionalization by HF and glutaraldehyde crosslinking [299]. In (**c**), the subscript of “3” on the glutaraldehyde structure indicates that only one of the three carbons between the formyl end groups has been drawn for brevity. Part (**c**) is adapted with permission from Ref. [299]. Copyright 2010 Elsevier.

**Figure 15 biosensors-13-00053-f015:**
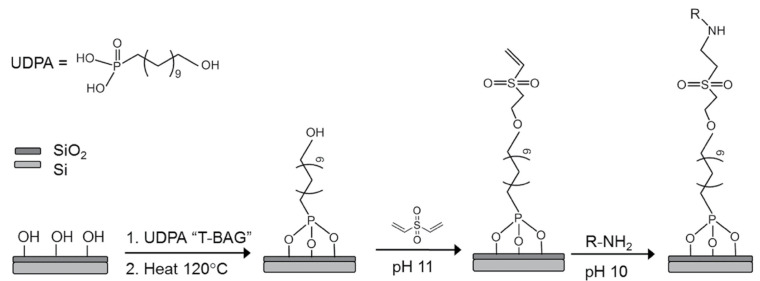
Organophosphonate-based surface functionalization scheme whereby the surface is coated with a film of UDPA using the T-BAG method and a DVS linking strategy is used for the immobilization of aminated bioreceptors. Reprinted with permission from Ref. [126]. Copyright 2012 American Chemical Society.

**Figure 16 biosensors-13-00053-f016:**
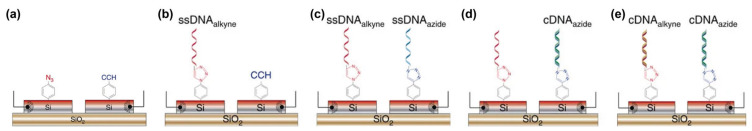
Click chemistry-mediated immobilization of nucleic acid probes on an electrophotonic ring resonator. (**a**) Two different diazonium salts (azidoaniline and 4-ethynylbenzene) are electrografted on the electrophotonic rings, which are electrically isolated. The individual microrings are then functionalized with alkyne- and azide-modified DNA probes using copper-catalyzed azide-alkyne click reaction. (**b**) First, the azide-modified sensor is functionalized with the alkyne-modified single-stranded DNA probe (ssDNA_alkyne_). (**c**) Next, the alkyne-modified sensor is functionalized with the azide-modified single-stranded DNA probe (ssDNA_azide_). The target sequences complementary to ssDNA_azide_ (**d**) ssDNA_alkyne_ (**e**) (labeled cDNA_azide_ and cDNA_alkyne_, respectively) are introduced and hybridize to the functionalized sensors. Adapted from Ref. [59] in accordance with the Creative Commons Attribution 4.0 International (CC BY 4.0) license.

**Figure 17 biosensors-13-00053-f017:**
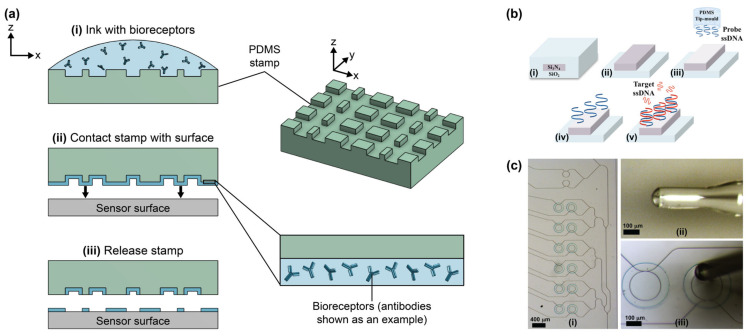
(**a**) Illustration of the process to pattern a surface using µCP. (i) First, the elastomeric stamp is inked with a bioreceptor solution whereby bioreceptors adsorb to the stamp surface. Inking may be achieved using soak, spray-on, or robotic feature-feature ink transfer methods. Subsequently, the stamp can be rinsed and dried or used wet for stamping. (ii) The stamp is contacted with the sensor surface and gentle pressure is applied to transfer the bioreceptors from the stamp to the surface at the regions of contact. (iii) The stamp is released to reveal the bioreceptor-patterned surface. (**b**) Graphical representation of the functionalization of a SiP waveguide using tip-mold microcontact printing, showing the (i) waveguide cross-section of a reference microring, (ii) waveguide cross-section of a control microring, (iii,iv) application of ssDNA probes on the waveguide surface using a PDMS tip-mold µCP tool, and (v) hybridization of ssDNA targets to the immobilized ssDNA probes on the waveguide surface. (**c**) Images of (i) the SiP MRR sensor chip functionalized by Peserico et al. [202] via tip-mold µCP, (ii) the optical fiber tip with an unpatterned PDMS cladding used as the µCP tool, and (iii) example of bioreceptor application on MRR using µCP. Parts (**b**,**c**) are adapted with permission from Ref. [202]. Copyright 2017 The Institution of Engineering and Technology.

**Figure 18 biosensors-13-00053-f018:**
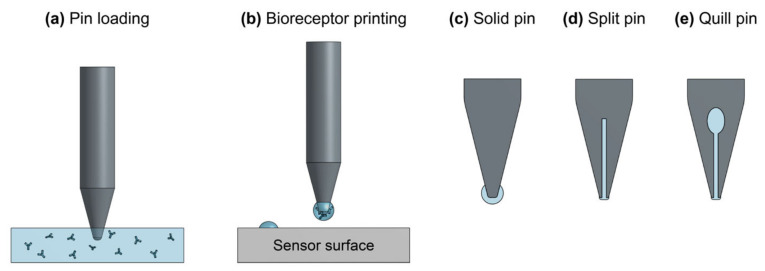
(**a**,**b**) Illustration of pin printing, showing (**a**) pin loading with the bioreceptor solution (antibody solution illustrated here as an example) and (**b**) printing of bioreceptors on a sensor surface using the loaded pin. (**c**–**e**) Different pin geometries, including (**c**) a solid pin, (**d**) split pin, and (**e**) quill pin.

**Figure 19 biosensors-13-00053-f019:**
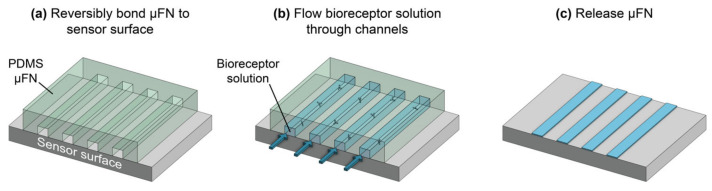
Illustration of the process of bioreceptor patterning on a sensor surface using a microfluidic network (µFN). (**a**) The PDMS µFN is reversibly bonded to the sensor surface, then (**b**) the bioreceptor solution is flowed through the microchannels via capillary or pressure-driven flow. This may be followed by rinsing and blocking steps. (**c**) Lastly, the µFN is released to reveal the bioreceptor-patterned surface.

**Figure 20 biosensors-13-00053-f020:**
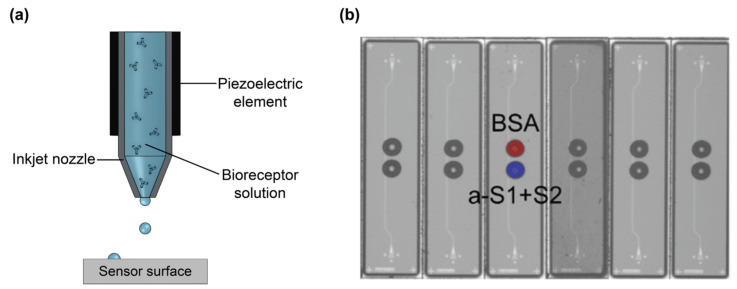
(**a**) Illustration of piezoelectric inkjet printing of bioreceptors on a sensor surface. (**b**) Image of antibody/antigen and BSA solutions spotted on silicon nitride photonic ring resonators using a Scienion SX microarrayer. The top (control) ring is spotted with BSA solution, and the bottom (test) ring is spotted with anti-(SARS-CoV-2 spike protein) polyclonal antibody solution (a-S1 + S2). Part (**b**) is reproduced from Ref. [167] in accordance with the Creative Commons Attribution 4.0 International (CC BY 4.0) license.

**Figure 21 biosensors-13-00053-f021:**
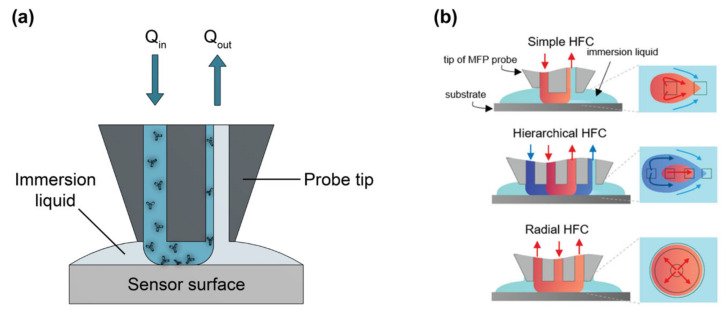
(**a**) Illustration of simple microfluidic printing (µFP) probe used to pattern a sensor surface with a bioreceptor solution (antibody solution illustrated here as an example). (**b**) Comparison of µFP using simple hydrodynamic flow confinement (HFC), hierarchical HFC, which permits recirculation of the patterning solution in the µFP head, and radial HFC, which produces circular, rather than teardrop-shaped spots. The processing (bioreceptor) solution is shown in red, the immersion liquid is shown in light blue, and the shaping liquid used for HFC is shown in dark blue. Insets on the right show the printing footprints for each µFP type. Part (**b**) is adapted with permission from Ref. [32]. Copyright 2021 American Chemical Society.

**Figure 22 biosensors-13-00053-f022:**
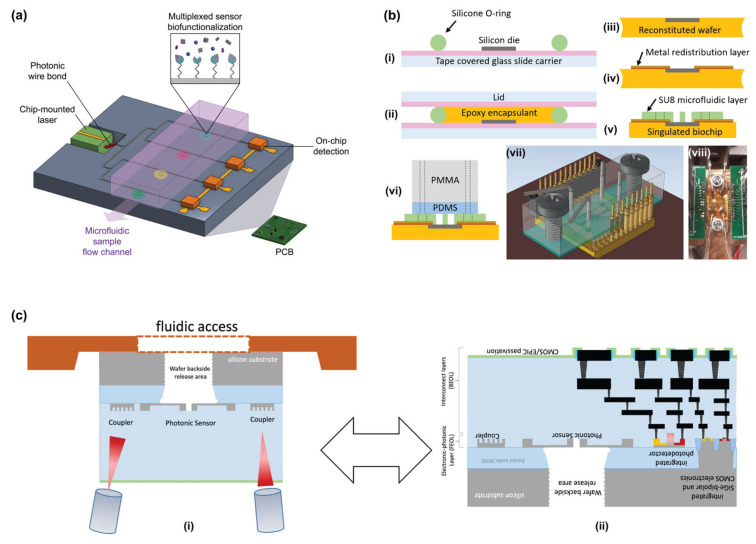
Integration approaches for SiP biosensors. (**a**) Multiplexed biofunctionalization of integrated SiP sensor system for POC use, which includes on-chip photonic inputs (chip-mounted fixed wavelength laser), outputs (on-chip detectors), photonic wire bonds, and microfluidics. See Ref. [13] for further information about this integration approach. (**b**) System-level integration of active SiP sensor by Laplatine et al. [66] using fan-out wafer-level packaging, showing (i–vi) schematics of the packaging process, (vii) 3D illustration of the packaged chip, and (viii) photograph of the experimental biochip setup. Part (**b**) is adapted with permission from Ref. [66]. Copyright 2018 Elsevier. (**c**) Photonic integrated circuit sensor chip presented by Mai et al. [354] using local backside release to enable integration with fluidics on one side of the chip and (i) optical coupling or (ii) optical coupling and electrical interconnects on the other. This allows for a more compact form factor than chips using front side integration only. Part (**c**) is adapted with permission from Ref. [354]. Copyright 2022 Elsevier.

**Table 1 biosensors-13-00053-t001:** Comparison of SiP and surface plasmon resonance (SPR) sensors, including SPR imaging (SPRi) and localized SPR (LSPR) devices.

	SiP	SPR
**Surface** [11,12,42]	Si or Si_3_N_4_, typically coated with native SiO_2_ film	Au
**Approx. evanescent field decay distance** [4,12,43,44,45,46]	~40–200 nm; depends on waveguide geometry and polarization	Typically ~200 nm; up to 600 nm when using an infrared laser or long-range surface plasmons
**Miniaturization** [12,13,47,48,49]	Very compact: chip-level integration with microfluidics, electronics, and optical inputs/outputs possible	Moderate: portable instrumentation demonstrated with dimensions ~10–20 cm
**Sensor size** [4,50,51,52]	Total sensor chip dimensions ~1–10 mm; active sensing spot dimensions ~10–100 µm	Total sensor chip dimensions ~10 mm; active sensing spot dimensions ~100 µm–1000 µm for multiplexed SPRi and LSPR devices
**Cost** [12,53]	Low (at high volume)	High
**Multiplexing** [53]	Multiplexable	Not possible with conventional SPR; multiplexable with SPRi and LSPR

**Table 2 biosensors-13-00053-t002:** Biofunctionalization needs for SiP biosensors. Please note that the performance metrics included in this table are general guidelines and designers should tailor these metrics based on their application. Interdependencies between the different columns of this table should also be considered (e.g., more expensive bioreceptors may still be suitable when combined with patterning techniques that permit very low reagent consumption).

Bioreceptor	Immobilization Chemistry	Patterning Technique
High affinity (K_D_~nM or lower) Selective (in the ideal case, signal change due to non-specific binding is less than the system limit of detection) Stable (can be stored at ambient conditions with minimal activity loss for times scales on the order of weeks; stable in biological analytes for several hours) Available as validated commercial products Scalable and reproducible production (in the ideal case, variations in target capture due to lot-to-lot variability is less than the system limit of detection) Regenerable or reversible (<5% signal loss between regeneration cycles and >10 consecutive regenerations possible) [82] Small (much smaller than evanescent field decay distance; ~10 nm or less) Low cost (~CAD 1–10/mg)	Compatible with Si or Si_3_N_4_ surfaces (or native SiO_2_) Stable (can be stored at ambient conditions for time scales on the order of weeks; stable in biological analytes for several hours) Thin (a few nm or less) Does not introduce a reduction in bioreceptor affinity due to denaturation or random orientation Replicable and uniform (<1 nm intra- and inter-chip variation in immobilization layer thickness) Compatible with system-level sensor integration (e.g., must not damage photonic wire bonds, chip-mounted lasers, or PCB materials) Scalable and simple (does not require highly skilled operators) Mild (no damage to sensor surface or bioreceptors)	Resolution ~10 µm or less Multiplexable (multiple reagents can be patterned on different regions of one surface) Uniform spots (<10% spot-to-spot variation; <10% intra-spot variation in bioreceptor loading density) [83] Reproducible (<10% run-to-run variation in spot size, shape, and bioreceptor loading density) High throughput (~10 spots per second or more) Low reagent consumption (minimal reagent waste) Simple (does not require highly skilled operators) Compatible with system-level sensor integration (e.g., must not damage photonic wire bonds, chip-mounted lasers, or PCB materials) No damage to sensor surface or bioreceptors Available as cost-effective commercial products or services

K_D_: dissociation constant, PCB: printed circuit board.

**Table 3 biosensors-13-00053-t003:** Comparison of different bioreceptor classes based on biofunctionalization needs for SiP biosensors.

Bioreceptor	Targets	Affinity	Specificity	Stability	Availability	Reproducibility of Production	Regenerability	Size	Cost *
**Antibody**	Antigens: diverse range of small molecules, complex macromolecules, viruses, and bacteria; exclude many toxins and non-immunogenic targets [29,33,84,85]	Very high; dissociation constant (K_D_) values typically in the low-nM to pM range [85,86]	Very high [87,88,89,90]	Poor [24,91]	High [92,93] Commercial products available from many vendors (e.g., Abcam plc, GenScript, Thermo Fisher Scientific Inc., etc.) ^†^ [94,95]	Poor to moderate [87,88,89]	Moderate; may experience activity loss [96]	Molecular weight (MW): ~150 kDa Diameter: ~20 nm [30,87,89]	High; ~USD 500/100 µg [89,97,98,99]
**Aptamer**	Very wide range including ions, small inorganic molecules, peptides, proteins, toxins, viral particles, and cells [85,89,100]	Very high; K_D_ values typically in the low-nM to pM range [85,100,101]	Very high [89,100]	High physical and thermal stability Long shelf life Unmodified versions susceptible to nuclease degradation in biological fluids [89,100,102]	Moderate Often require custom design and synthesis [89] Design/discovery available via companies such as Aptagen, Base Pair Biotechnologies, Aptamer Sciences (AptaSci), etc.^†^ [89] Synthesis of designed aptamers widely available via companies such as Integrated DNA Technologies (IDT), Thermo Fisher Scientific, and Twist Bioscience, etc.^†^ [103,104]	High due to chemical synthesis [89]	Good [96,105]	MW: 5–30 kDa (for aptamers consisting of nucleic acid sequences ~15–100 bases in length) Diameter: ~2 nm [85,89,102]	DNA aptamer: low; USD 0.07–5.40/mg for large scale synthesis [89] RNA aptamer: moderate; USD 20–67/mg for large scale synthesis [106]
**Nucleic acid probe**	Nucleic acids	High to very high, depending on type of nucleic acid; K_D_ values demonstrated in the low-nM range for DNA-DNA duplexes [107]	High to very high [29,73,81,108,109]	High physical and thermal stability Long shelf life Unmodified versions susceptible to nuclease degradation in biological fluids Peptide nucleic acids (PNA), locked nucleic acids (LNA), and morpholinos have improved enzymatic stability [73,81,101,108,110]	Moderate-high Easy to design once target sequence is known Usually require custom synthesis, which is widely available via companies such as IDT, Thermo Fisher Scientific, and Twist Bioscience, etc.^†^ [103,104]	High due to chemical synthesis [111,112]	Good [80]	MW: 1.5–30 kDa (for nucleic acid sequences ~5–100 bases in length) [113] Length: ~0.33 nm/base [114]	DNA: low; USD 0.07–5.40/mg for large scale synthesis [89] RNA: moderate; USD 20–67/mg for large scale synthesis [106] PNA, LNA, and morpholinos: high [115,116,117]
**Molecularly imprinted polymer (MIP)**	Various small molecules; template molecules must be able to survive in organic solvents	Wide range; K_D_ values reported in the 10^−15^–10^−5^ M range [72]	Moderate	High mechanical and chemical stability [118]	Low Custom synthesis available from some companies including Affinisep, MIP Diagnostics, MIP Technologies, etc.^†^ [72,119]	High	Good	Thin film (sub-nanometer to micrometers) [72]	Dependent on development
**Synthetic peptides** [91,120]	A range of small molecules to proteins	Very high	High	High temperature stability up to 100 °C Resistant to enzymatic degradation	High Commercially available synthesis (e.g., Custom Peptide synthesis via Thermo Fisher, Millipore Sigma, AnaSpec, GenScript) ^†^	Very high	Good	2–14 kDa; 89–204 Da per amino acid Protein-catalyzed capture agents (PCCs) are 2–4 kDa	Moderate; CAD 10–100 for custom synthesis dependent on amount and purity for <0.5 g
**Native peptides** [120]	A range of small molecules to proteins	Very high	High	Poor	High Commercially available (e.g., via Thermo Fisher, Millipore Sigma, ABclonal, RayBiotech) ^†^	Very high	Moderate	2–14 kDa; 89–204 Da per amino acid	High; CAD 100–1000; dependent on peptide, purity, and quantity
**Glycan and lectin**	Glycan: lectins, toxins, and viruses Lectin: glycans and glycoconjugates	Low; K_D_ in the µM–mM range [121,122]	Low [121,123]	Glycan: good shelf life and stability under dry and ambient conditions [124,125,126] Lectin: poor [121]	Glycan: low-moderate; some commercial products (e.g., from Sigma Aldrich, Biosynth, etc.) ^†^; custom chemical synthesis possible (e.g., via Creative Biolabs, Asparia Glycomics, Glycan Therapeutics, etc.) ^†^, but challenging [127,128,129] Lectin: moderate-high; some commercial products (e.g., from Sigma Aldrich, Vector Laboratories, Medicago AB, etc.) ^†^; custom synthesis possible [130]	Glycan: high due to chemical synthesis [127,128] Lectin: moderate [121]	Glycan: good regenerability using concentrated salt and high/low pH solutions [126] Lectin: moderate; may experience activity loss [121]	Glycan: highly variable [128] Lectin: ~10–140 kDa [131,132,133,134]	Glycan: very high; ~CAD 200–1200/10 µg for commercial products [135] Lectin: moderate; ~CAD 1–300/mg [136]
**High contrast cleavage detection (HCCD)**	Nucleic acids	High	Very high [137,138]	Nucleic acid-conjugated reporters: good thermal and physical stability; susceptible to nuclease degradation in biological fluids CRISPR-Cas enzymes: poor [33]	Moderate Most reagents commercially available (e.g., via IDT, Abcam, Sigma Aldrich, etc.) ^†^	Moderate-high	Not regenerable	Reporters: ~15–20 nm diameter Single-stranded DNA (ssDNA) anchor: ~12 kDa, ~13.2 nm long (~40 bases) [137,139]	High Guide RNA: CAD 180/10 nmol [140] Cas12a nuclease: CAD 1250/500 µg (3.2 nmol) [140] Reporters: ~CAD 400–600/100 mL for gold nanoparticle reporters [141]; ~CAD 200–900/10 mg for quantum dot reporters [142]
**CRISPR-dCas9-mediated detection**	Nucleic acids	High	Very high [143]	Nucleic acid probes: good thermal and physical stability; susceptible to nuclease degradation in biological fluids Cas9 and recombinase polymerase amplification (RPA) enzymes: poor	Moderate Most reagents commercially available (e.g., via IDT, Thermo-Fisher Scientific, TwistDX, etc.) ^†^	Moderate-high	Good	Probe MW: 1.5–30 kDa (for nucleic acid sequences ~5–100 bases in length) [113] Probe length: ~0.33 nm/base [114]	High Probe ssDNA: CAD 0.07–5.40 for large scale synthesis [89] Guide RNA: CAD 180/10 nmol [144] Cas9 nuclease: CAD 1250/500 µg [144] RPA reagents: ~CAD 5.00/reaction [145,146,147]
**Lipid nanodisc**	Proteins	Moderate; K_D_ in nM–µM range for phospholipid nanodiscs [148,149]	Variable; depends on membrane protein content [148,149,150]	Poor due to protein content	Low-moderate Few commercial products available (e.g., Cube Biotech) ^†^ Typically require custom synthesis in laboratory [151]	Moderate	Good [148,149]	8–16 nm diameter [150,151]	High ~CAD 770–1011/25 nmol (~$770–$1011/500 µg) [152] ~CAD 0.19–0.54/chip (based on chip functionalization procedure in [148])

* Prices are listed in Canadian dollars (CAD) and are based on commercial products available as of July 2022. ^†^ Vendors listed are based on an exploratory search and are not endorsed or suggested by the authors.

**Table 4 biosensors-13-00053-t004:** Comparison of antibody subtypes as bioreceptors for SiP biosensors.

	Affinity [86,87,88,89]	Specificity [87,88,89,90]	Availability [92,93,94,95,158,159]	Reproducibility of Production [87,88,89]	Size [30,87,89]	Cost * [89,97,98,99]
**Polyclonal antibody**	Very high, but with significant variability within a sample	High Greater risk of cross-reactivity than monoclonal antibodies Target multiple antigen epitopes and are less sensitive to small changes in epitope structure	High; many vendors available (key companies include Abcam plc, GenScript, Thermo Fisher Scientific Inc., etc.) ^†^	Poor	MW: ~150 kDa Diameter: ~15 nm	High ~CAD 500/100 µg
**Monoclonal antibody**	Very high with K_D_ values as low as 10 pM	Very high Target a single antigen epitope and are very sensitive to small changes in epitope structure Low risk of cross-reactivity	High; many vendors available (key companies include Abcam plc, GenScript, Thermo Fisher Scientific Inc., etc.) ^†^	Moderate	MW: ~150 kDa Diameter: ~15 nm	High ~CAD 500/100 µg
**Fab fragment**	Very high	High to very high, depending on origin	High; can be prepared from available antibodies or purchased commercially (major antibody-fragment producing companies include AbbVie, Amgen, Novartis AG, etc.) ^‡^	Poor to moderate, depending on origin	MW: ~50 kDa	High

* Prices are listed in CAD and are based on commercially available products as of July 2022. ^†^ The named vendors comprise a small subset of the major competitors in the global research antibodies market. Readers are directed elsewhere [94,95] for a more comprehensive list of major vendors. Vendors listed are not endorsed or suggested by the authors. ^‡^ The named vendors comprise a small subset of the major competitors in the global antibody fragments market. Readers are directed elsewhere [158,159] for a more comprehensive list of major vendors. Vendors listed are not endorsed or suggested by the authors.

**Table 5 biosensors-13-00053-t005:** Demonstrations of SiP biosensors using antibodies or antibody fragments as the biorecognition element and their sensing performance.

Bioreceptor Description	Sensor Type	Target	Detection Performance		Assay Format	Refs.
			Figure of Merit	Value		
Monoclonal antibody	Si MRR	Thrombin	Min. detected concentration	500 pM	Label-free	[174]
Monoclonal antibodies	Si MRR	Carcinoembryonic antigen	Min. detected concentration	10 ng/mL	Label-free	[22]
		Prostate-specific antigen				
		α-fetoprotein				
		Interleukin-8				
		Tumor necrosis factor-α				
Monoclonal antibody	Si MRR	Monocyte chemotactic protein 1	Limit of detection (LoD)	0.5 pg/mL	Amplification with secondary antibody and enzymatic enhancement	[17]
Monoclonal antibody	Si MRR	Carcinoembryonic antigen	LoD	2 ng/mL in buffer, 25 ng/mL in serum	Label-free	[161]
Monoclonal antibodies	Si MRR	Interleukin-2	LoD	100 pg/mL	Sandwich immunoassay	[18]
		Interleukin-8	LoD	100 pg/mL		
Monoclonal antibody	Si MRR	Bean pod mottle virus	LoD	10 ng/mL	Label-free	[169]
Monoclonal antibody	Si MRR	C-reactive protein	-	-	Label-free	[166]
Monoclonal antibodies	Si MRR	α-fetoprotein	Working range	0.3–20.6 ng/mL	Amplification with secondary antibody and protein-based multilayer signal enhancement	[163]
		Activated leukocyte cell adhesion molecule	Working range	1.0–43.7 ng/mL		
		Cancer antigen 15-3	Working range	2.0–91.5 units/mL		
		Cancer antigen 19-9	Working range	2.5–96.6 units/mL		
		Cancer antigen-125	Working range	2.4–95.6 units/mL		
		Carcinoembryonic antigen	Working range	0.16–20.2 ng/mL		
		Osteopontin	Working range	4.3–50.3 ng/mL		
		Prostate specific antigen	Working range	0.054–4.7 ng/mL		
Monoclonal antibodies	Si MRR	Interleukin-2	LoD	1 pg/mL	Amplification with secondary antibody and enzymatic enhancement	[162]
		Interleukin-6	LoD	1 pg/mL		
		Interleukin-8	LoD	0.5 pg/mL		
Monoclonal antibody	Hydex MRR	*E. coli* O157:H7 bacterial cells	LoD	10^5^ CFU/mL	Label-free	[172]
Monoclonal antibody	Si PhC	Human Papillomavirus virus-like particles	LoD	1.5 nM	Label-free	[170]
Antibody	Si PhC	Cardiac myoglobin	Min. detected concentration	70 ng/mL	Label-free	[165]
Monoclonal antibodies	Si_3_N_4_ planar waveguide interferometer	Hemagglutinin (H7N2 and H7N3)	Min. detected concentration	0.05 hemagglutination (HA) units/mL	Label-free	[175]
Polyclonal antibodies	Si_3_N_4_ planar waveguide interferometer	Hemagglutinin (H7N2)	Min. detected concentration	0.0005 HA units/mL		
		Hemagglutinin (H7N3)	Min. detected concentration	0.005 HA units/mL		
Polyclonal antibody	Porous Si sensor using reflectometric interference spectroscopy	Insulin	LoD	4.3 µg/mL	Label-free	[176]
Antibody	Si PhC total internal reflection	Cardiac troponin I	LoD	0.01 ng/mL	Label-free	[164]
Antibody	Si_3_N_4_/SiO_2_ slot-waveguide MRR	Bovine serum albumin	LoD	16 pg/mm^2^	Label-free	[177]
Antigen-binding fragment (Fab) from protease digestion of polyclonal IgG	SiO_x_N_y_ MRR	Aflatoxin M1	LoD	5 nM	Label-free	[25]
Single domain antibodies	Si MRR	Ricin	LoD	200 pM	Label-free	[168]

Si: silicon, LoD: limit of detection, Si_3_N_4_: silicon nitride, SiO_2_: silicon dioxide, SiO_x_N_y_: silicon oxynitride.

**Table 6 biosensors-13-00053-t006:** Advantages and limitations of antibodies as bioreceptors.

Advantages	Limitations
Diverse targets including small molecules, complex macromolecules, viruses, and bacteria [29,33,84] High affinity and specificity [87,88,89] Widely available [92,93] Regeneration possible for multiple binding cycles [18]	Poor stability [24,91] Batch-to-batch and vendor-to-vendor variability [92,156,178] Expensive [89,157] Potential activity loss from immobilization and regeneration procedures [75,96] Time-consuming and laborious discovery and production [30,89] Susceptible to sample contamination [89]

**Table 7 biosensors-13-00053-t007:** Comparison of aptamer subtypes as bioreceptors for SiP biosensors.

	Affinity [89,100]	Specificity [89,100]	Stability [89,100,102,183]	Availability [89]	Reproducibility of Production [89]	Attachment Chemistry	Size * [85,89,102]	Cost ^†^ [89,106,182]
**DNA** **aptamer**	High	High	High physical and thermal stability Unmodified versions susceptible to nuclease degradation	Moderate May require custom design via SELEX (e.g., via Aptagen, Base Pair Biotechnologies, Aptamer Sciences (AptaSci), etc.) ^‡^ Custom synthesis of already-designed aptamers widely available (e.g., via IDT, Thermo Fisher Scientific, and Twist Bioscience, etc.) ^§^	High due to chemical synthesis	Typically, covalent immobilization via terminal functional groups (e.g., 5′ amines)	MW: 5–30 kDa Diameter: ~2 nm	Low Research scale synthesis: CAD 1.40–2.40/base for 1 µmol quantities for sequences of 5–100 bases in length from IDT Large scale synthesis: USD 0.05–4/mg
**RNA** **aptamer**	Very high due to more diverse 3D conformations	Very high due to more diverse 3D conformations	High physical and thermal stability Unmodified versions more susceptible to more susceptible to nuclease degradation and hydrolysis at pH > 6 than DNA aptamers Stronger RNA-RNA intra-strand interactions than in DNA aptamers	Moderate May require custom design via SELEX (e.g., via Aptagen, Base Pair Biotechnologies, Aptamer Sciences (AptaSci), etc.) ^‡^ Custom synthesis of already-designed aptamers widely available (e.g., via IDT, Thermo Fisher Scientific, and Twist Bioscience, etc.) ^§^	High due to chemical synthesis	Typically, covalent immobilization via terminal functional groups (e.g., 5′ amines)	MW: 5–30 kDa Diameter: ~2 nm	Moderate Research scale synthesis: CAD 24.00/base at 1 µmol quantities for sequences of 5–60 bases in length and CAD 23.00/base at 80 nmol quantities for sequences of 60–120 bases in length from IDT Large scale synthesis: USD 14.50–50/mg

* Molecular weight and size for aptamers consisting of 15–100 nucleotides. ^†^ Prices are listed in CAD for products available as of July 2022, unless otherwise specified. ^‡^ Readers are directed elsewhere [89] for a more comprehensive list of key companies in the global aptamers market. Vendors listed are not endorsed or suggested by the authors. ^§^ Readers are directed elsewhere [103,104] for a more comprehensive list of key companies in the global oligonucleotide synthesis market. Vendors listed are not endorsed or suggested by the authors.

**Table 8 biosensors-13-00053-t008:** Demonstrations of SiP biosensors using DNA aptamers as the biorecognition element and their sensing performance. All demonstrations tabulated here used label-free assay formats.

Sensor Type	Target	Detection Performance		Refs.
		Figure of Merit	Value	
Si MRR	IgE	LoD	33 pM	[24]
	Thrombin	LoD	1.4 nM	
Si MRR	Thrombin	Min. detected concentration	500 pM	[174]
SiO_x_N_y_ MRR	Aflatoxin M1	LoD	5 nM	[25]
SiO_x_N_y_ MRR	Aflatoxin M1	Min. detected concentration	1.58 nM	[185]
Si PhC total internal reflection	Cardiac troponin I	LoD	0.1 ng/mL	[164]
Porous Si reflectometric interference spectroscopy	Insulin	LoD	1.9 µg/mL	[176]

**Table 9 biosensors-13-00053-t009:** Advantages and limitations of aptamers as bioreceptors.

Advantages	Limitations
High affinity and specificity [89,100] Can be designed for theoretically any target, including toxins and non-immunogenic species [85,89,100] Small size [89] Relatively low-cost and rapid discovery via SELEX process [89,102] Produced via chemical synthesis yielding low batch-to-batch variability and allowing for the introduction of chemical modifications for improved functionality [85,89] Good stability and long shelf life [89,105] Good regenerability: regeneration can be achieved using temperature or concentrated salt, acidic, basic, chaotropic agent, surfactant, and chelating agent solutions [96,105] Low production costs for DNA aptamers [89]	Nuclease susceptibility of unmodified aptamers [100,102,180] High production costs for RNA aptamers [89] Low SELEX success rates [101,180] Structural conformation and binding are sensitive to pH, ionic strength, and temperature [180] Less widely available than antibodies and usually require custom synthesis [89]

**Table 10 biosensors-13-00053-t010:** Comparison of nucleic acid subtypes as bioreceptors for SiP biosensors.

	Affinity [29,30,81,108]	Specificity [29,73,81,108,109]	Stability [73,81,101,108,110,183,189]	Availability	Reproducibility of Production [89]	Attachment Chemistry	Cost * [115,116,117,182]
**DNA**	High	High	High physical and thermal stability Susceptible to nuclease degradation	Moderate Custom synthesis required Many vendors available (e.g., IDT, Thermo Fisher Scientific, Twist Bioscience, etc.) ^‡^	High due to chemical synthesis	Typically, covalent immobilization via terminal functional groups (e.g., 5′ amines)	Low CAD 2.40/base for 5–100 bases at 1 µmol production scale from IDT 25-base custom DNA oligo: CAD 60 at 1 µmol production scale
**RNA**	High	High	High physical and thermal stability More susceptible to nuclease degradation and hydrolysis at pH > 6 than DNA aptamers RNA-RNA duplexes have higher thermal stability than DNA-DNA duplexes of the same sequence	Moderate Custom synthesis required Many vendors available (e.g., IDT, Thermo Fisher Scientific, Twist Bioscience, etc.) ^‡^	High due to chemical synthesis	Typically, covalent immobilization via terminal functional groups (e.g., 5′ amines)	Medium CAD 24.00/base for 5–60 bases at 1 µmol production scale from IDT 25-base custom RNA oligo: CAD 600 at 1 µmol production scale
**PNA**	Higher than DNA/RNA due to neutral charge	Very high	Very high physical, thermal, and enzymatic stability	Low Custom synthesis required Few vendors available (e.g., biomers.net, PNA Bio Inc., Panagene, etc.) ^§^	High due to chemical synthesis	Typically, covalent immobilization via terminal functional groups (e.g., 5′ amines)	High 25-base custom PNA oligo: ~CAD 8100 at 1 µmol production scale from biomers.net
**LNA**	Higher than DNA/RNA due to decreased configurational flexibility	Very high	Very high physical, thermal, and enzymatic stability	Low Custom synthesis required Few vendors available (e.g., IDT Affinity Plus™ products, Qiagen, LGC Biosearch Technologies, etc.) ^§^	High due to chemical synthesis	Typically, covalent immobilization via terminal functional groups (e.g., 5′ amines)	High CAD 52.00/base for 5–100 bases at 1 µmol production scale from IDT ^†^ 25-base custom LNA-containing oligo: CAD 1300 at 1 µmol production scale
**Morpholino**	Higher than DNA/RNA due to neutral charge, but lower affinity than PNA	Very high	Very high physical, thermal, and enzymatic stability Unstable at low pH	Low Custom synthesis required Few vendors available (e.g., Gene Tools, E-nnovation Life Sciences) ^§^	High due to chemical synthesis	Typically, covalent immobilization via terminal functional groups (e.g., 5′ amines)	High 25-base custom morpholino oligo: ~CAD 1300 at 1 µmol production scale from Gene Tools

* All prices are listed in CAD for oligonucleotides with no chemical modifications. Prices are provided for µmol-scale production for comparative purposes. Note that lower prices are available for large scale synthesis. ^†^ Price provided for DNA oligonucleotides containing 1–20 LNA bases. ^‡^ Readers are directed elsewhere [103,104] for a more comprehensive list of key companies in the global DNA and RNA oligonucleotide synthesis market. Vendors listed are not endorsed or suggested by the authors. ^§^ These are not exhaustive lists of PNA, LNA, and morpholino vendors. Vendors listed are based on an exploratory search and are not endorsed or suggested by the authors.

**Table 11 biosensors-13-00053-t011:** Demonstrations of SiP biosensors using nucleic acid probes as the biorecognition element and their sensing performance.

Bioreceptor	Sensor Type	Target	Detection Performance		Assay Format *	Refs.
			Figure of Merit	Value		
ssDNA	Si MRR	microRNA	LoD	150 fmol (i.e., 75 µL of 2 nM microRNA solution)	Label-free	[194]
ssDNA	Si MRR	Complementary DNA generated from targeted microRNAs	-	-	Label-free	[195]
ssDNA	Si MRR	Full-length mRNA transcripts	LoD	32 fmol for label-free detection; 512 amol with bead-based amplification	Label-free and with streptavidin-coated bead-based amplification	[196]
ssDNA	Si MRR	microRNA	LoD	10 pM (i.e., 350 amol in a 35 µL sample)	Amplification with anti-DNA:RNA antibodies	[109]
ssDNA	Si MRR	ssDNA	LoD	400 fmol (i.e., 16 µL of 25 nM ssDNA solution)	Label-free	[197]
ssDNA	Si MRR	Bacterial transfer-messenger RNA (tmRNA)	LoD	52.4 fmol (i.e., 100 µL of 524 pM tmRNA solution)	Label-free	[198]
ssDNA	Si MRR	Methylated DNA	-	-	Label-free	[23]
ssDNA	Cascaded Si MRRs	IS6110 ssDNA biomarker	LoD	1 fg (corresponds to 10 µL of 0.1 pg/mL ssDNA solution)	Label-free	[199]
		IS1081 ssDNA biomarker	LoD	10 fg (i.e., 10 µL of 1 pg/mL ssDNA solution)		
ssDNA	Si_3_N_4_ MRR	ssDNA	-	-	Label-free	[202]
ssDNA	N-doped Si MRR electrophotonic sensor	ssDNA	-	-	label-free	[59]
ssDNA	Si_3_N_4_ MZI	ssDNA	LoD	300 pM	Label-free	[200]
ssDNA	Planar Si PhC waveguide	ssDNA	LoD	19.8 nM	Label-free	[203]
ssDNA (directly conjugated)	Si MRR and Si PhC	ssDNA	LoD	50 nM	Label-free	[201]
		ssPNA	-	-		
ssDNA (synthesized in situ)	Si MRR and Silicon PhC	ssDNA	LoD	10 nM		
		ssPNA	-	-		
Methylated ssDNA	Si_3_N_4_ slot waveguide MZI	Methylated ssDNA	Min. detected concentration	1 fmol/µL (1nM)	Label-free	[192]
Morpholino	Suspended Si MRR	ssDNA	Min. detected concentration	250 pM	Label-free	[110]

* Does not include PCR amplification prior to introduction to sensor surface.

**Table 12 biosensors-13-00053-t012:** Advantages and limitations of nucleic acid probes as bioreceptors.

Advantages	Limitations
Simple to design once target sequence is known [30] Reproducible chemical synthesis [111,112] Chemical modifications can be introduced [81,111,112,190] Amenable to thermal or chemical regeneration [80] Synthetic DNA analogs (PNA, LNA, and morpholinos) available to enhance affinity, specificity, and stability [30,73,81,110]	Limited to nucleic acid targets [30] Challenging to capture long targets due to secondary structures [70,198] Targets often require significant sample preparation [188] DNA and RNA susceptible to non-specific interactions due to negative charge [30] Steric hindrance effects may limit probe immobilization density and binding capacity [71,201] Low molecular weight nucleic acid targets are challenging to detect without amplification [138]

**Table 13 biosensors-13-00053-t013:** Strategies demonstrated for the preparation of MIPs on SiP sensors and representative surfaces.

MIP Type	Film Constituents	Template	Film Deposition Technique	Template Extraction	Sensor Type	Refs.
Polymer synthesis	Methacrylic acid (functional monomer) and ethylene glycol dimethacrylate (crosslinking agent)	Testosterone	Casting, followed by thermopolymerization	Acetic acid and ethanol-based chemical extraction	Si MRR	[212]
Polymer synthesis	Methacrylic acid (functional monomer) and ethylene glycol dimethacrylate (crosslinking agent)	Progesterone	Coating, followed by UV photopolymerization	-	Cascaded Si MRRs	[219]
Sol-gel	Bis(trimethoxysilylethyl)benzene and 2-(2-pyridylethyl)trimethoxysilane, prepared in tetrahydrofuran	Carbamate (used to create trinitrotoluene binding sites)	Airspray coating or electrospray ionization	HCl and chloroform-based chemical extraction	Si MRR	[213]
Sol-gel	Ethanol, methyltrimethoxysilane, aminopropyltriethoxysilane, and HCl, prepared in dimethyl sulfoxide	Fluorescein isothiocyanate	Dip coating	Oxygen plasma degradation or chemical extraction with solutions of ethanol, acetic acid, and chloroform or acetonitrile.	SiO_2_ microsphere whispering gallery mode resonators	[211]
Sol-gel	Tetraethoxysilane (TEOS), water, ethanol, and HCl or methyltriethoxysilane (C1-TriEOS), ethanol, HCl, and MPTMS.	Cortisol	Spin coating	Ethanol-based chemical extraction.	Si chips	[214,215]

**Table 14 biosensors-13-00053-t014:** Advantages and limitations of MIPs as bioreceptors.

Advantages	Limitations
Diverse targets [72] Amenable to many (>50) cycles of reversible binding [221] Excellent chemical, mechanical, and temporal stability [221] Scalable and low-cost production due to synthetic nature	Poor specificity [223] Heterogeneous binding sites Low availability and complicated optimization of formulations

**Table 15 biosensors-13-00053-t015:** Demonstrations of SiP biosensors using MIPs as the biorecognition element and their sensing performance.

Bioreceptor	Sensor Type	Target	Detection Performance		Assay Format	Refs.
			Figure of Merit	Value		
MIP film	Si MRR	Testosterone	LoD	48.7 pg/mL	Label-free	[212]
Sol-gel MIP	Si racetrack resonators	Trinitrotoluene vapor	Min. detected concentration	5 ppb	Label-free	[213]
MIP film	Cascaded Si MRRs	Progesterone	LoD	83.5 fg/mL	Label-free	[219]
MIP film	SiO_2_ microsphere whispering gallery mode resonator	Fluorescein isothiocyanate	-	Fluorescence intensity	-	[211]
MIP film	SiO_x_N_y_ dual polarization interferometer	Hemoglobin	LoD	2 µg/mL	Label-free	[225]

**Table 16 biosensors-13-00053-t016:** Demonstrations of SiP biosensors using peptides as the biorecognition element and their sensing performance. All tabulated demonstrations used a label-free assay format.

Bioreceptor	Sensor Type	Target	Detection Performance		Refs.
			Figure of Merit	Value	
Peptide	Si_3_N_4_ MZI	SARS-CoV-2 antibodies	LoD	80 ng/mL	[238]
Peptide	Planar Si and porous Si microcavity	A20 lymphoma cancer cells	Coverage efficiency after 2 h incubation with 50,000 A20 cells	~85% and ~4% for planar and porous functionalized surfaces, respectively	[239]
PCC	Porous Si microcavity	Chikungunya virus E2 protein	Resonance shift after 3 h incubation with 1 µM E2 protein	1.7 ± 0.3 nm	[91]
PCC	Porous Si microcavity	Streptavidin	Resonance redshift after 1 h exposure to 5 µM streptavidin	12.9 nm	[240]

**Table 17 biosensors-13-00053-t017:** Advantages and limitations of peptides and PCCs as bioreceptors.

Advantages	Limitations
Small size Synthetic production is available for scalable and flexible production [230] Good specificity and selectivity [235] Good temperature stability and resistance to protease degradation [91,235,243]	Limited availability Limited data available to assess performance (particularly on SiP platforms)

**Table 18 biosensors-13-00053-t018:** Comparison of glycans and lectins as bioreceptors for SiP biosensors.

	Compatible Targets [121,122]	Affinity [121,122]	Specificity [121,123]	Stability [121,124,125,126]	Availability [127,128,129,130]	Reproducibility of Production [121,127,128]	Attachment Chemistry [244]	Size [121,128,131,132,133,134]	Cost *[135,136]
**Glycan**	Lectins, toxins, and viruses	Low; K_D_ in µM–mM range	Low	Good shelf life and stable under dry and ambient conditions	Low–moderate Some commercial products (e.g., via Sigma Aldrich, Biosynth, etc.) ^†^ Custom chemical synthesis possible for small glycans, but challenging for complex structures (e.g., synthesis available via Creative Biolabs, Asparia Glycomics, Glycan Therapeutics, etc.) ^†^	High due to chemical synthesis	Site-directed immobilization via terminal amine group	Diverse (polysaccharides containing a few monosaccharide units to thousands)	Very high; ~CAD 200–CAD 1200/10 µg for commercial products
**Lectin**	Carbohydrates, including glycans and glycoconjugates	Low; K_D_ in µM–mM range	Low	Low due to susceptibility to permanent denaturation	Moderate–high Some commercial products available (e.g., via Sigma Aldrich, Vector Laboratories, Medicago AB, etc.) ^†^ Custom synthesis via recombinant techniques available	Moderate due to variability introduced by cell-based synthesis techniques	Challenging to achieve oriented immobilization due to complex structure	Diverse, but typically smaller than antibodies (~10–140 kDa)	Moderate; ~CAD 1–300/mg

* Prices are listed in CAD and are based on commercially available products as of July 2022. ^†^ These are not exhaustive lists of glycan or lectin vendors. Vendors listed are based on an exploratory search and are not endorsed or suggested by the authors.

**Table 19 biosensors-13-00053-t019:** Demonstrations of SiP biosensors using glycans or lectins as the biorecognition element and their sensing performance. All tabulated demonstrations used label-free assay formats.

Bioreceptor	Sensor Type	Target	Detection Performance		Refs.
			Figure of Merit	Value	
Lacto-N-fucopentaose III-human serum albumin (LNFPIII-HSA) glycoprotein	Si MRR	Norovirus-like particles	LoD	250 ng/mL	[126]
BSA-mannose, BSA-lactose, BSA-galactose and RNase B glycoconjugates	Si MRR	Lectins: concanavalin A, griffithsin, and ricin	-	-	[246]
GM1 ganglioside glycan	Si_3_N_4_ MRR	Cholera toxin subunit B	LoD	400 ag (corresponds to 8 pg/mm^2^)	[132]
3-fucosyl lactose glycan	Si_3_N_4_ MRR	Aleuria Aurantia Lectin	LoD	0.5 ng/mL (7 pM)	[133]
α2,6-disialylated biantennary N-glycan	Si_3_N_4_ MRR	Sambucus Nigra Lectin	LoD	12 ng/mL (86 pM)	
Concanavalin A lectin	Porous Si	*Escherichia coli*	LoD	10^3^ cells/mL	[131]
Wheat germ agglutinin lectin	Porous Si	*Staphylococcus aureus*	LoD	10^3^ cells/mL	

**Table 20 biosensors-13-00053-t020:** Advantages and limitations of glycans and lectins as bioreceptors.

	Advantages	Limitations
**Glycan**	Easy to achieve oriented immobilization [244] Highly reproducible chemical synthesis [127,128] Good stability [124,125,126] Good regenerability [126]	Poor affinity and specificity [121,122,123] Functional study of glycans is less advanced than proteins and nucleic acids [123,129] Chemical synthesis is challenging for long and branched structures [129] Expensive [247]
**Lectin**	Well-understood and relatively inexpensive synthesis by recombinant techniques [121]	Poor affinity and specificity [121,122,123] Challenging to achieve oriented immobilization [244] Poor reproducibility due to cell-based synthesis [121] Limited regenerability [121]

**Table 21 biosensors-13-00053-t021:** Demonstration of SiP biosensor using HCCD its sensing performance.

Bioreceptor	Sensor Type	Target	Detection Performance		Assay Format	Ref.
			Figure of Merit	Value		
CRISPR-Cas12a with guide RNA complementary to target	Si MRR	SARS-CoV-2 ssDNA	Resonance shift after exposure to 1 nM of target DNA	~8 nm	Labeled: gold nanoparticle reporters tethered to sensor surface by ssDNA were cleaved by activated Cas12a effector; amplification via collateral cleavage	[139]

**Table 22 biosensors-13-00053-t022:** Advantages and limitations of HCCD.

Advantages	Limitations
Very high sensitivity Signal enhancement due to high index contrast reporters [137,138] Multiplicative signal enhancement due to collateral reporter cleavage by Cas12/Cas13 effectors [138,251] Insensitive to non-specific interactions [137] Universal reporters for simple sensor functionalization [138]	Challenging to multiplex Not regenerable Limited to nucleic acid targets Poor stability of Cas enzymes [33] Limited precedent for use and not yet demonstrated for sensing in complex biological fluids

**Table 23 biosensors-13-00053-t023:** Demonstrations of SiP biosensors using CRISPR-dCas9-mediated sensing and their performance.

Bioreceptor	Sensor Type	Target	Detection Performance		Assay Format	Ref.
			Figure of Merit	Value		
ssDNA probes	Si MRR	Scrub typhus viral DNA	LoD	0.54 aM	Isothermal pre-amplification of targets; target-specific CRISPR-dCas9 signal amplification	[143]
		Severe fever with thrombocytopenia viral RNA	LoD	0.63 aM		

**Table 24 biosensors-13-00053-t024:** Advantages and limitations of CRISPR-dCas9-mediated sensing.

Advantages	Limitations
Very high sensitivity and specificity [143] Multiplexable Good nucleic acid probe stability	Limited to nucleic acid targets Requires many assay reagents Expensive reagents Limited precedent for use

**Table 25 biosensors-13-00053-t025:** Demonstrations of SiP biosensors using lipid nanodiscs as the biorecognition element and their sensing performance. All tabulated demonstrations used a label-free assay format.

Bioreceptor	Sensor Type	Target	Detection Performance		Ref.
			Figure of Merit	Value	
Lipid nanodiscs containing PC, four binary compositions of PC and PS, and two binary combinations of PS and PA	Si MRR	Blood clotting proteins: pro-thrombin, factor X, activated factor VII, and activated protein C	-	-	[149]
Lipid nanodiscs containing POPC and POPC/POPS	Si MRR	Annexin V	-	-	[150]
Lipid nanodiscs containing GM1	Si MRR	Cholera Toxin Subunit B	-	-	
Lipid nanodiscs containing biotin-DPPE	Si MRR	Streptavidin	-	-	
Lipid nanodiscs containing CYP3A4	Si MRR	Anti-CYP3A4 antibody	-	-	
Lipid nanodiscs with 9 different compositions containing PS, PE, and PC.	Si MRR	Protein clotting factors: prothrombin, activated factor VII, factor IX, factor X, activated protein C, protein S, and protein Z	-	-	[148]

PC: phosphatidylcholine, PS: phosphatidylserine, PA: phosphatidic acid, POPC: 1-palmitoyl-2-oleoyl-sn-glycero-3-phosphocholine, POPS: 1-palmitoyl-2-oleoyl-sn-glycero-3-(phospho-L-serine), GM1: monosialotetrahexosyl ganglioside, biotin-DPPE: *N*-(biotinoyl)-1,2-dipalmitoyl-sn-glycer-3-phosphoethanolamine, CYP3A4: cytochrome P450 3A4, and PE: phosphatidylethanolamine.

**Table 26 biosensors-13-00053-t026:** Advantages and limitations of lipid nanodiscs as bioreceptors.

Advantages	Limitations
Solubilize and stabilize membrane proteins for studying cell membrane interactions [151] Better consistency, production, monodispersity, production yield, and control over lipid composition than other cell membrane mimics [150,151] Easy immobilization by physisorption [150] Regenerable [148,149]	Poor selectivity, but can be improved by incorporation of membrane proteins [148,149,150] Limited availability; usually custom synthesized in lab [151] Biosensing applications are mainly limited to the study of cell membrane interactions

**Table 27 biosensors-13-00053-t027:** Comparison of different bioreceptor immobilization chemistries based on SiP biofunctionalization needs.

Immobilization Chemistry	Compatible Bioreceptors	Surface Modification	Bioreceptor Modification	Required Linkers	Stability	Thickness	Oriented Bioreceptor Immobilization?	Impact on Bioreceptor Function	Typical Replicability and Uniformity	Compatibility with System-Level Sensor Integration	Process Scalability
**Passive adsorption**											
Passive adsorption	All	Not required	Not required	None	Typically, poor [31,69,261]	No added thickness	Typically, no	Likely to reduce bioreceptor binding activity [29,69,262,263]	Poor	Good	Very good
**Bioaffinity**											
Antibody (Ab)- binding protein	Antibodies	Ab-binding protein is typically passively adsorbed on sensor	None	None	Moderate [29,75]	3–4 nm for adsorbed PrA [259,264]	Yes	Preserves bioreceptor binding activity	Depends on Ab-binding protein immobilization strategy	Depends on Ab-binding protein immobilization strategy	Depends on Ab-binding protein immobilization strategy
Biotin/(strept)avidin	Antibodies, aptamers, nucleic acid probes, peptides, PCCs, glycans/lectins	Often silanization	Biotinylation	(Strept)avidin acts as bioaffinity linker	Good [74,265]	~6–7 nm plus thickness of chemical layer used to immobilize (strept)avidin [266]	Possible	Preserves bioreceptor binding activity [263]	Depends on (strept)avidin immobilization strategy	Depends on (strept)avidin immobilization strategy	Poor due to complexity
**Covalent**											
Silane chemistry	Antibodies, aptamers, nucleic acid probes, peptides, PCCs, glycans/lectins	Silanization	Aptamers and nucleic acids require modification with terminal functional groups (e.g., amine, carboxyl, thiol). S-4FB conjugation required for SoluLink chemistry.	Often required; popular options include GA, BS^3^, and EDC/NHS	Good [31]	<1 nm for silane monolayer; multilayer films may exceed 10 nm [267]	Possible	Typically preserves bioreceptor binding activity; strategies requiring antibody disulfide bond reduction may reduce antibody binding activity [69,75]	Variable for solution-phase silanization; good for vapor-phase silanization [261,267,268,269]	Poor for solution-phase silanization; good for vapor-phase silanization	Fair-good, depending on reaction conditions and linker requirements
Organophosphonate chemistry	Antibodies, aptamers, nucleic acid probes, peptides, PCCs, glycans/lectins	UDPA deposition	Aptamers and nucleic acids require modification with terminal functional groups	Required; DVS and 3-maleimidopropionic-acid-*N*-hydroxysuccinimidester have been used [126,270]	Very good [126]	~1 nm [270]	Possible	Preserves bioreceptor binding activity [126,270]	Very good	Poor due to solution-phase UDPA deposition	Fair; multistep process
Click chemistry	Antibodies, aptamers, nucleic acid probes, peptides, PCCs, glycans/lectins	Azide/alkyne derivatization	Modification with azide/alkyne moieties, DBCO, or tetrazine	Azide, alkyne, DBCO, or tetrazine terminations required	Good [91]	<1 nm	Possible	Preserves bioreceptor binding activity	Good; insensitive to oxygen and moisture [271]	Poor due to solution-phase chemistry and aggressive surface pre-treatment strategies, which may damage sensor [59,91,240]	Fair; multistep process
UV-crosslinking	Aptamers [272] and nucleic acids [273]	None required	Modification with poly(T) or poly(TC) tags	None	Good [272,273]	Thickness added by poly(T) or poly(TC) tag; depends on tag length	Yes	Preserves bioreceptor binding activity [272,273]	Good due to process simplicity [272]	Very good	Very good

S-4FB: sulfo succinimidyl 4-formylbenzoate, GA: glutaraldehyde, BS^3^: bis(sulfosuccinimidyl)suberate, EDC/NHS: 1-ethyl-3-[3-(dimethylamino)propyl]-carbodiimide/*N*-hydroxysuccinimide, UDPA: 11-hydroxyundecylphosphonic acid, DVS: divinyl sulfone, DBCO: dibenzoazacyclooctyne or aza-dibenzocyclooctyne, poy(T)/poly(TC): poly(thymine)/poly(thymine cytosine).

**Table 28 biosensors-13-00053-t028:** Strategies demonstrated for the immobilization of antibodies on SiP sensors.

Antibody Immobilization						
Immobilization Strategy	Bioreceptor Subtype	Surface Pre-Treatment	Chemical Surface Modification	Linking Strategy	Sensor Type	Refs.
Covalent	Fab fragment	Argon plasma treatment	MPTMS silanization	Direct conjugation to silanized surface	SiO_x_N_y_ MRR	[25]
Covalent	Monoclonal antibodies	NaOH and ethanol/water cleaning	GPTMS silanization	Direct conjugation to silanized surface	Hydex MRR	[172]
Covalent	Monoclonal [18,22,109,169,174,295] and single-domain antibodies [168]	Piranha treatment	HyNic silane surface modification	Conjugation of antibody with S-4FB for hydrazone bond formation with modified surface	Si MRR	[18,22,162,168,169,174,295]
Covalent	Monoclonal antibody	Piranha treatment	APTES silanization + S-HyNic surface modification	Conjugation of antibody with S-4FB for hydrazone bond formation with modified surface	Si MRR	[161]
Covalent	Monoclonal antibodies	Acetone/isopropanol cleaning [17,27] or oxygen plasma [166]	APTES silanization	BS^3^ crosslinker	Si MRR	[17,27,166]
Covalent	Monoclonal antibody	Piranha treatment	APTES silanization	*N,N*-diisopropylethylamine and *N,N*′-disuccinimidyl carbonate linker	Si MRR	[295]
Covalent	Monoclonal antibody	Oxygen plasma	APTES silanization	EDC/NHS activation	Si MRR	[166]
Covalent	Antibody	Piranha treatment	APTES silanization	EDC/NHS activation	Si PhC	[165]
Covalent	Monoclonal antibody	Piranha treatment	APDMES silanization	Glutaraldehyde linker	Si PhC	[170]
Covalent	Polyclonal antibody	Oxidization	CTES silanization	EDC/NHS activation	Si_3_N_4_ MRR	[171]
Covalent	Polyclonal antibody	-	Thermal hydrolyzation with undecylenic acid	EDC/NHS activation	Porous Si sensor	[176]
Covalent	Antibody	Piranha treatment	Native oxide removal with HF	Glutaraldehyde linker	Si_3_N_4_/SiO_2_ MRR	[177]
Covalent	Monoclonal and polyclonal antibodies	Cleaning with Micro-90 solution and chromic acid	MPTMS silanization	*m*-maleimidobenzoyl-*N*-hydroxysuccinimide ester linker	Si_3_N_4_ planar waveguide interferometer	[175]
Bioaffinity	Antibody	-	-	Protein A adsorbed on surface	Si MRR (sub-wavelength grating, SWG)	[264]
Bioaffinity	Antibody	-	-	Protein A adsorbed on surface	Si MRR (multibox SWG)	[1]
Bioaffinity	Antibodies	-	-	Antibody-binding fusion protein consisting of Si-tag and Protein A	Si wafer	[278]
Bioaffinity	Antibodies	-	-	Antibody-binding fusion protein consisting of Si-tag and Protein A	Si MRR	[282]
Covalent + bioaffinity	Monoclonal antibodies (oligonucleotide-conjugated)	Piranha treatment	HyNic silane surface modification	Intermediate oligonucleotides conjugated with S-4FB for hydrazone bond formation with modified surface, then used as a bioaffinity linker	Si MRR	[163]
Covalent + bioaffinity	Antibody	Plasma treatment	Silane-PEG-biotin surface modification	Streptavidin used as a bioaffinity linker to immobilize biotinylated Protein G	Si PhC	[164]
Covalent + bioaffinity	Antibodies (biotinylated)	Dry thermal oxidation	APTMS silanization	Sulfo-NHS-LC-LC-biotin linker + streptavidin used as bioaffinity linkers	Porous Si sensor	[288,289]
Covalent + bioaffinity	Monoclonal and polyclonal antibodies	HF treatment	MPTMS silanization	*N*-succinimidyl-4-malemidobutyrate crosslinker + Protein A or G	Multimode optical fibers	[283]

MPTMS: (3-mercaptopropyl)trimethoxysilane, GPTMS: 3-glycidoxypropyltrimethoxysilane, HyNic silane: 3-*N*-((6-(*N*′-Isopropylidene-hydrazino))nicotinamide)propyltriethyoxysilane, S-HyNic: succinimidyl 6-hydra- zinonicotinamide acetone hydrazone, APTES: 3-aminopropyltriethoxysilane, APTMS: 3-aminopropyltrimethoxysilane, Sulfo-NHS-LC-LC-biotin: sulfosuccinimidyl-6-(biotinamido)-6-hexanamido hexanoate, HF: hydrofluoric acid.

**Table 29 biosensors-13-00053-t029:** Strategies demonstrated for the immobilization of aptamers on SiP sensors and representative surfaces. All works listed in this table used DNA aptamers.

Aptamer Immobilization						
Immobilization Strategy	Surface Pre-Treatment	Chemical Surface Modification	Linking Strategy	Targeted Aptamer Terminal Group	Sensor Type	Refs.
Covalent	Argon plasma [25] or piranha treatment [185]	GPTMS silanization	Direct conjugation to silanized surface	Amine	SiO_x_N_y_ MRR	[25,185]
Covalent	Oxygen plasma	APTES silanization	Glutaraldehyde linker	Amine	Si MRR	[24]
Covalent	Piranha treatment	HyNic silane surface modification	Conjugation of aptamer with S-4FB for hydrazone bond formation with modified surface	Amine	Si MRR	[174]
Covalent	Plasma	Silane-PEG-COOH surface modification	EDC/NHS activation	Amine	Si PhC	[164]
Covalent	-	Thermal hydrolyzation with undecylenic acid	EDC/NHS activation	Amine	Porous Si sensor	[176]
Covalent	-	-	Direct UV crosslinking on surfaces	poly(T) and poly(TC) tags	Glass slides and SiO_2_ wafers	[272]

Silane-PEG-COOH: slane-polyethylene glycol-carboxyl.

**Table 30 biosensors-13-00053-t030:** Strategies demonstrated for the immobilization of nucleic acid probes on SiP sensors and representative surfaces.

Immobilization of Nucleic Acid Probes for Hybridization Sensing							
Immobilization Strategy Type	Bioreceptor Subtype	Surface Pre-Treatment	Chemical Surface Modification	Linking Strategy	Targeted Probe Terminal Group	Sensor Type	Refs.
Covalent	DNA	Piranha treatment	ICPTS silanization	Direct conjugation to silanized surface	Amine	Si wafers and nanostructured Si	[291]
Covalent	DNA	NaOH and ethanol/water cleaning	GPTMS silanization	Direct conjugation to silanized surface	Amine	Hydex MRR	[172]
Covalent	DNA	Piranha treatment and thermal oxidation	APTES silanization	Sulfosuccinimidyl-4-(*N*-maleimidomethyl)cyclohexane-1-carboxylate (Sulfo-SMCC) linker	Thiol	Si MRR and Si PhC	[201]
Covalent	DNA	Piranha treatment and thermal oxidation	TEOS-HBA silanization	Base-by-base in situ ssDNA probe synthesis via phosphoramidite method	-	Si MRR and Si PhC	[201]
Covalent	DNA	Piranha treatment	APTES silanization + S-HyNic surface modification	Conjugation of DNA probe with S-4FB for hydrazone bond formation with modified surface	Amine	Si MRR	[194]
Covalent	DNA	Nitric acid wash	APTES silanization	Succinic anhydride and EDC linker	Amine	Si_3_N_4_ MRR	[202]
Covalent	PNA	Oxygen plasma	11-hydroxyundecylphosphonate surface modification	3-maleimidopropionic-acid-*N*-hydroxysuccinimide linker	Cysteine tag	Si nanowires	[270]
Covalent	DNA	Ethanol and water rinse	-	Direct UV crosslinking on surfaces	No tag and poly(TC) tag	Glass slides (unmodified and GAPS II™ aminosilane coated)	[273]
Covalent	DNA	Piranha treatment	APTES silanization	BS^3^ linker	Amine	Si MRR	[195]
Covalent	DNA	Piranha and oxygen plasma treatment	Electrografting of alkyne- and azide-presenting diazonium salts	Cu-catalyzed azide-alkyne click reaction	Azide and alkyne	N-doped Si MRR electrophotonic sensor	[59]
Covalent	DNA	Piranha treatment	HyNic silane surface modification	Conjugation of DNA probe with S-4FB for hydrazone bond formation with modified surface	Amine	Si MRR	[109,196,198,303]
Covalent	DNA	Oxygen plasma and nitric acid treatment	MPTMS silanization	Direct conjugation via disulphide bond formation	Thiol	Si_3_N_4_ Mach-Zehnder interferometer	[200]
Covalent	DNA	Oxygen plasma	APTES silanization	Glutaraldehyde linker	Amine	Si MRR	[23,197,199]
Covalent	Morpholino	Piranha treatment	APTMS silanization	Glutaraldehyde linker	Amine	Suspended Si photonic microring resonator	[110]
Covalent + bioaffinity	DNA (biotinylated)	-	3-isocyanatepropyl thriethoxysilane vapor silanization	Streptavidin conjugated to silanized surface and used as a bioaffinity linker	Biotin tag	Planar photonic crystal- waveguide-based optical sensor	[203]

ICPTS: 3-isocyanatepropyl triethoxysilane, TEOS-HBA: *N*-(3-triethoxysilylpropyl)-4-hydroxybutyramide.

**Table 31 biosensors-13-00053-t031:** Strategies demonstrated for the immobilization of peptides and PCCs on SiP sensors.

Peptide and PCC Immobilization							
Immobilization Strategy	Bioreceptor Subtype	Surface Pre-Treatment	Chemical Surface Modification	Linking Strategy	Sensor Type	Refs.
Covalent	Peptide	Acetone and isopropanol cleaning	HF treatment to create secondary amines	Glutaraldehyde linker	Si_3_N_4_ Mach-Zehnder interferometer	[238]	
Passive adsorption	Peptide	Acetone and isopropanol cleaning + Piranha treatment	APTES silanization	-	Si MRR	[174]	
Covalent	Peptide	Piranha treatment	APTES silanization	BS^3^	Porous Si microcavity	[239]	
Covalent	PCC	HF treatment	Thermal hydrolyzation with 1,8-nonadiyne	PCC attachment via click chemistry with copper(I)-catalyzed azide alkyne cycloaddition	Porous Si microcavity sensor	[91,240]	

**Table 32 biosensors-13-00053-t032:** Strategies demonstrated for the immobilization of glycans and lectins on SiP sensors and representative surfaces.

Glycan and Lectin Immobilization						
Immobilization Strategy	Bioreceptor Subtype	Surface Pre-Treatment	Chemical Surface Modification	Linking Strategy	Sensor Type	Refs.
Covalent	Glycan	Piranha treatment	UDPA organophosphonate surface modification	DVS activation	Si MRR	[126]
Covalent	Glycan	Piranha and UV/ozone plasma treatment	MPTMS silanization	SM(PEG)12 linker	Si_3_N_4_ MRR	[133]
Covalent	Glycan	Piranha treatment	APTES silanization	BS(PEG)9 linker	Si_3_N_4_ MRR	[132]
Covalent	Lectin	Hydrogen peroxide and thermal treatment	APTES silanization	Glutaraldehyde linker	Porous Si sensor	[131]
Non-covalent	Glycoproteins and neoglycoconjugates	-	-	Passive adsorption	Si MRR	[246]
Covalent + bioaffinity	Lectin (biotinylated)	UV/ozone clean	APTMS silanization	NHS-PEG_4_-biotin linker + avidin	Si_3_N_4_ reflectometric interference spectroscopy sensor	[285]

SM(PEG)12: succinimidyl-([*N*-maleimidopropionamido]-dodecaethyleneglycol) ester, BS(PEG)9: bis-*N*-succinimidyl-(nonaethylene glycol) ester, NHS-PEG_4_-biotin: *N*-hydroxysuccinimide ester-polyethylene glycol-biotin.

**Table 33 biosensors-13-00053-t033:** Strategies demonstrated for the immobilization of HCCD reporters on SiP sensors.

HCCD Reporter Immobilization						
Immobilization Strategy	Reporter Type	Surface Pre-Treatment	Chemical Surface Modification	Linking Strategy	Sensor Type	Refs.
Covalent + bioaffinity	Biotinylated dsDNA-quantum dot reporters	Thermal oxidation	APTES silanization	Glutaraldehyde linker + streptavidin used as a bioaffinity linker	Porous Si sensor	[137]
Covalent + bioaffinity	Biotinylated ssDNA-gold nanoparticle reporters	Plasma treatment	Silane-PEG-biotin surface modification	Streptavidin used as a bioaffinity linker	Si MRR	[139]

**Table 34 biosensors-13-00053-t034:** Strategies demonstrated for the immobilization of nucleic acid probes for CRISPR-dCas9-mediated sensing on SiP sensors.

CRISPR-Ca9-Mediated Sensing					
Immobilization Strategy	Surface Pre-Treatment	Chemical Surface Modification	Nucleic Acid Probe Linking Strategy	Sensor Type	Refs.
Covalent	Oxygen plasma	APTES silanization	Glutaraldehyde linker	Si MRR	[143]

**Table 35 biosensors-13-00053-t035:** Strategies demonstrated for the immobilization of lipid nanodiscs on SiP sensors.

Lipid Nanodisc Immobilization					
Immobilization Strategy	Surface Pre-Treatment	Chemical Surface Modification	Linking Strategy	Sensor Type	Refs.
Non-covalent	Piranha treatment	-	Passive adsorption	Si MRR	[149]
Non-covalent	Piranha treatment	-	Passive adsorption	Si MRR	[150]
Non-covalent	Acetone and isopropanol wash	-	Passive adsorption	Si MRR	[148]

**Table 36 biosensors-13-00053-t036:** Comparison of bioreceptor patterning techniques based on SiP biofunctionalization needs.

Patterning Technique	Achievable Resolution	Ease of Multiplexing	Spot quality/ Uniformity	Reproducibility	Throughput	Reagent Consumption	Simplicity	Compatibility with System-Level Integration	Cost/ Availability
**Microcontact printing (µCP)**	0.1–0.5 µm [32,77]	Poor	Poor	Moderate	High throughput for patterning with 1 bioreceptor; low throughput for multiplexing	Low	Good	Poor; risk of surface and system damage due to stamp contact	Low cost, widely available (e.g., stamp preparation via standard soft lithography techniques; commercial µCP services available via companies including ThunderNIL and BALTFAB) *
**Pin printing**	~1–100 µm [313,314,315]	Moderate	Poor	Moderate	Low-high, depending on pin design	Low	Poor	Poor; risk of surface and system damage due to pin contact	Expensive for commercial pin printers (e.g., Bioforce Nano eNabler, BioOdyssey Calligrapher miniarrayer, Arrayit microarrayers) *
**Microfluidic patterning in channels**	~1–10 µm [316,317]	Moderate	Good	Very good	Moderate; simultaneous bioreceptor deposition possible, but limited number of uniquely addressable locations	High	Good	Moderate; µFN design and placement require careful design to avoid system damage; microfluidics often required for assays too	Low cost, widely available (e.g., µFN preparation via standard soft lithography techniques; commercial microfluidics fabrication services available via companies including ThunderNIL and MicruX Technologies) *
**Inkjet** **printing**	~30–150 µm [318,319]	Good	Poor	Moderate	Very high	Low	Poor	Good	Expensive, commercial options available (e.g., Scienion sciFLEXARRAYER, Fujifilm Dimatix DMP-2831) *
**Microfluidic probe (µFP)**	10 µm [83]	Good	Good	Very good	High	Low-moderate	Poor	Moderate; non-contact, but small risk of system damage due to probe motion	No commercial products available

µFN: microfluidic network. * These are not exhaustive lists of vendors. Vendors listed are based on an exploratory search and are not endorsed or suggested by the authors.

**Table 37 biosensors-13-00053-t037:** Demonstration of bioreceptor patterning using µCP for the functionalization of SiP sensors.

Patterning Technique	Sensor Type	Printed Bioreceptors	Multiplexed Bioreceptor Patterning (i.e., 4-Plex, 8-Plex…)	Ref.
Tip-mold reactive microcontact printing	Si_3_N_4_ MRR	ssDNA	-	[202]

**Table 38 biosensors-13-00053-t038:** Demonstrations of bioreceptor patterning using pin and pipette spotting for the functionalization of SiP sensors.

Patterning Technique	Sensor Type	Printed Bioreceptors	Multiplexed Bioreceptor Patterning (i.e., 4-Plex, 8-Plex…)	Ref.
Hand spotting with micropipette (0.2 µL/drop)	Si MRR	Antibodies	2-plex, including control	[17]
Hand spotting with micropipette (1 µL/drop)	Si MRR	Antibodies	Up to 4-plex, including controls	[168]
Hand-spotting with micropipette	Si MRR	ssDNA probes	4-plex	[109]
Hand-spotting with micropipette	Si MRR	ssDNA probes	4-plex	[303]
Hand spotting with micropipette (1 µL/drop)	Si MRR	ssDNA probes	2-plex	[198]
Hand spotting with micropipette (0.1–0.2 µL/spot)	Si MRR	Lipid nanodiscs	9-plex	[148]
Hand spotting with micropipette	Si MRR	ssDNA probes	3-plex	[196]
Hand-spotting with micropipette	Si MRR	ssDNA probes	4-plex	[194]
BioForce Nano eNabler	Si_3_N_4_ MRR	Glycans	2-plex	[133]
BioForce Nano eNabler	Si_3_N_4_ MRR	Glycan	-	[132]
BioForce Nano eNabler	Si_3_N_4_ bimodal waveguide interferometric biosensor	BSA	-	[313]
BioForce Nano eNabler	Si MRR	ssDNA (subsequently used to immobilize DNA-conjugated antibodies)	8-plex	[163]
BioForce Nano eNabler	Si MRR	Lipid nanodiscs	7-plex	[149]
BioOdyssey Calligrapher miniarrayer	Si_3_N_4_ MZI	SARS-CoV-2 peptide	2-plex, 20–46 overlapping spots	[238]

**Table 39 biosensors-13-00053-t039:** Demonstrations of bioreceptor patterning using microfluidic patterning in channels for the functionalization of SiP sensors.

Patterning Technique	Sensor Type	Printed Bioreceptors	Multiplexed Bioreceptor Patterning (i.e., 4-Plex, 8-Plex…)	Ref.
Microfluidic patterning in channels using 4-channel Mylar gasket	Si MRR	Antibody, DNA aptamer, ssDNA (control sequence), and BSA (control)	4-plex, including 2 controls	[174]
Microfluidic patterning in channels using 6-channel PDMS gasket	Si MRR	Antibodies	6-plex, including one control	[22]
Microfluidic patterning in channels using 2-channel Mylar gasket	Si MRR	Antibody	Patterning used to functionalize half of the rings with antibody and leave the other half bare for temperature corrections	[161]
Microfluidic patterning in channels using 2-channel Mylar gasket	Si MRR	Antibody	Patterning used to functionalize some rings with antibody and leave the rest bare to control for nonspecific binding, temperature, and instrumental drift	[18]
Microfluidic patterning in channels using 2-channel Mylar gasket	Si MRR	Antibody	µFN used to perform functionalization in the presence of catalyst on some rings and without catalyst on others	[295]
Microfluidic patterning in channels using 1-, 2-, and 4-channel Mylar gaskets	Si MRR	Lipid nanodiscs	Up to 4-plex	[150]
Microfluidic patterning in channels using 2- and 4-channel gaskets	Si MRR	Antibodies	Up to 4-plex	[169]
Microfluidic patterning in channels using Mylar gasket	Si MRR	Antibodies	4-plex	[162]
Microfluidic patterning in channels using 4-channel PDMS gasket	Si_3_N_4_ bimodal waveguide interferometric biosensor	BSA	2-plex to compare adsorption- and covalent-based BSA immobilization	[313]

**Table 40 biosensors-13-00053-t040:** Demonstrations of bioreceptor patterning using inkjet printing for the functionalization of SiP sensors.

Patterning Technique	Sensor Type	Printed Bioreceptors	Multiplexed Bioreceptor Patterning (i.e., 4-Plex, 8-Plex…)	Ref.
Piezoelectric non-contact printing (Scienion sciFLEXARRAYER S5)	Si MRR	SS-A antigen	-	[293]
Piezoelectric non-contact printing (Scienion sciFLEXARRAYER S3)	Si MRR	Glycoconjugates and fluorescently labeled streptavidin	Up to 4-plex	[246]
Piezoelectric non-contact printing (Scienion SciFLEXARRAYER SX)	Si_3_N_4_ MRR	Antibody, antigen, BSA	2-plex (performed on 2-ring chips; one ring spotted with antibody or antigen as the receptor and another with BSA as a control)	[167]
Piezoelectric non-contact printing (Scienion sciFLEXARRAYER S12)	Si_3_N_4_ MZI	Peptides for volatile chemicals	64 MZI sensor array	[319]
Piezoelectric non-contact printing (FUJIFILM Dimatix DMP-2831)	SiP microcantilevers	Biotin	-	[318]
Piezoelectric non-contact printing (FUJIFILM Dimatix DMP-2831)	Silicon nitride MZI	Biotin-modified polyethyleneimine functional polymer and benzophenone dextran hydrogel	-	[347]

## Data Availability

Not applicable.

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
