# Peer review of "Biofunctionalization of Multiplexed Silicon Photonic Biosensors"

_biosensors, 2022, doi:10.3390/bios13010053_

Round 1

Reviewer 1 Report

In this paper, Puumala et al. comprehensively summarized the biofunctionalization strategies of silicon photonic biosensors, including the bioreceptor types, immobilization methods, and patterning methods. Overall, the paper is of high quality and clearly written. I recommend accepting it after correcting a few minor errors.

1.      In the caption of figure 6, b should be solution based and c should be surface stamping. I also suggest revising figures b and c to better illustrate the different processes of the two methods.

2.      Line 909, 103 cells/ml should be 10^3 cells/ml.

3.      In Table 20, what does the “Glycans” mean in the limitations of Glycan?

4.      In Table 25, the cells in rows 3-6 of the Ref. column can be merged. 

Reviewer 2 Report

The authors present an impressive review that is well-positioned to become a go-to resource for researchers active in the area as well as those new to it. While the field is broad, this review focuses directly on biofunctionalization, resisting the temptation to become so broad as to be less useful. For example, there is no need to discuss, in detail, wave-guide design, etc… I just have a few comments that may help improve the overall manuscript.

1) A table of contents would be helpful. The manuscript is long enough that jumping back and forth is a bit cumbersome.

2) While the authors mention another realm of near-field optical sensing, e.g. SPR, it could be useful, in addition to the information in table 1, to discuss not just the differences (e.g. surface chemistries) but overall advantages and disadvantages. For example, many SPR and LSPR sensing platforms are interrogated in the far-field, and don’t need difficult on-chip optical excitation sources like the SiP waveguide structures. Another point of comparison would be the “surface-enhanced” effects that are so prevalent in metallic nano-structures, e.g. surface enhanced Raman spectroscopy (SERS). While the SERS effect is also being considered in silicon photonics, which could allow some very compelling multi-modal sensing strategies, it still requires more research and development (see, e.g., https://doi.org/10.1021/acs.jpclett.8b00662). Other advantages and disadvantages could be useful to mention. This should be brief, however, and not detract from the overall theme of the biofunctionalization of SiP devices.

3) “LSPR” and “SPRi” need to be defined in table 1.

4) Table 2 is a good start, but is completely subjective. What does “low cost” mean? What is “small”? How “stable” should the surface be? What is a “high” enough affinity? I think having some metrics or minimum performance characteristics here would be very helpful to those reading the review. While tables further on list some specifics, having a simple range at the beginning to define "high" and other metrics like that could be helpful, especially since the other tables and the main text discussion use many different units (for example: $/mg, $/µg, CAD$, US$, $/mol, etc…).

5) The availability of commercial products is mentioned quite a few times, especially in the tables, but only a few companies are named (e.g., in the micro-printing category). Perhaps include some of the main commercial sources in the tables? Again, if this will become a go-to resource, having this information quickly accessible would be very useful.

6) Figure 22 shows the only real schematic of a “finished” device. Perhaps have a few more realizations / concepts of what a final product could look like? Anything from the literature? Not necessary, but it could provide a broader picture of the challenges that remain.

Reviewer 3 Report

In this paper, the biological functionality of silicon photonic sensors and their applications in multiplexed analysis are reviewed. The bioreceptors, immobilization strategies and patterning techniques for the sensor surface are focused. The content of this paper is very comprehensive, and the comparison in the tables is clear. However, the following points need to be further clarified:

1. For different biosensing applications, is there a basic guide for the selection of a suitable bioreceptor?

2. The introduction to the preparation of bioreceptor can be relatively simple, which is not the focus of this article. Some contents related to biosensing can be appropriately added, such as how to improve the specificity of recognition, binding strength, and more effectively transform the recognition process into detectable signals.

3. In the surface immobilization, some contents, related to surface treatment of sensing materials (e.g. Si), and how to make the surface more strongly combine with biomolecules through such treatment, can be introduced.

4. In the process of surface immobilization of bioreceptors, how to avoid the influence of immobilization process, surface properties, steric hindrance, etc. on the recognition ability of receptors?

5. Bioreceptors, immobilization strategies and patterning techniques introduced in this article are also used in almost all biosensors (e.g. SPR). What is special about these recognition materials and surface modification methods used for SiP, or what needs special attention in relevant research?

Reviewer 4 Report

The authors provided a comprehensive and in-depth review on the biofunctionalization of silicon photonic biosensors. The review is important and timely, which have significant reference and guiding values for this area. Its contents contain rich figures and tables as well as comparisons and discussions, which helps readership easily and quickly understand the advances of this area. So I recommend its publication in Biosensors as revisions as follows:

1.     The advantages of the silicon photonic biosensors are highlighted in the Introduction parts, which indicates the importance of this research area. In fact, each kind of biosensor has its own merits. So I suggest the authors should also add the descriptions and discussions of some other typical biosensors such as electrochemical biosensors compared with the silicon photonic biosensors so as to help readers know its advance.

2.     For the Conclusions part, summary and discussion is given. But throughout the manuscript, the further development directions and suggestions are seldom mentioned, which is actually very important for the further progress of this area. So I suggest the corresponding study directions in the future should be pointed out.

3.     The point-of-care sensing is mentioned frequently and obvious very important for applications. Then its super-flexible and wearable even super-foldable devices are necessarily one important development direction in the future. So its prospect should be made by referring to the closely related references: DOI: 10.1016/j.matt.2021.07.021;

Round 2

Reviewer 3 Report

The manuscript has met the requirements for publication.